# Propane wet reforming over PtSn nanoparticles on γ-Al₂O₃ for acetone synthesis

Xinlong Ma[1,2,6], Haibin Yin[2,6], Zhengtian Pu[2,6], Xinyan Zhang[2], Sunpei Hu[2], Tao Zhou[2], Weizhe Gao[3], Laihao Luo [2 ✉], Hongliang Li [2,4 ✉] & Jie Zeng [1,2,5 ✉]

Acetone serves as an important solvent and building block for the chemical industry, but the current industrial synthesis of acetone is generally accompanied by the energy-intensive and costly cumene process used for phenol production. Here we propose a sustainable route for acetone synthesis via propane wet reforming at a moderate temperature of 350 °C with the use of platinum-tin nanoparticles supported on γ-aluminium oxide (PtSn/γ-Al₂O₃) as catalyst. We achieve an acetone productivity of 858.4 μmol/g with a selectivity of 57.8% among all carbon-based products and 99.3% among all liquid products. Detailed spectroscopic and controlled experiments reveal that the acetone is formed through a tandem catalytic process involving propene and isopropanol as intermediates. We also demonstrate facile ketone synthesis via wet reforming with the use of different alkanes (*e.g.*, n-butane, n-pentane, n-hexane, n-heptane, and n-octane) as substrates, proving the wide applicability of this strategy.

Acetone serves as an important solvent and platform intermediate for the chemical industry, baring a global market of more than US$6 billion and an annual production of more than 6 million tons[1,2]. Currently, acetone is mainly produced as a byproduct during the cumene process used for phenol synthesis (Fig. 1a)[3,4]. The process is energy-intensive and generates large quantities of hazardous waste[5]. Recent years have witnessed the development of alternative methods for acetone synthesis including direct propene hydration and isopropanol oxidation, which exhibited advantages in high atomic efficiency and simple operation procedure (Fig. 1b)[6–9]. Nevertheless, all these strategies involve downstream chemicals, *e.g.* propene (C₃H₆) and isopropanol (i-C₃H₇OH), from the petrochemical industry based on unsustainable oil cracking.

The large increase in the availability of propane (C₃H₈) from the shale gas revolution offers an alternative source for multicarbon (C₃) chemical synthesis[10]. For example, over 11% of global propene is now produced via non-oxidative propane dehydrogenation processes[11–14].

Unfortunately, the intrinsic stability of $sp^3$ carbon-hydrogen (C–H) bonds and the poor electron affinity make the dehydrogenation process thermodynamically unfavorable[15,16]. Generally, high temperatures (550–750 °C) are required to achieve satisfactory performances, which demands for a large energy in-put and not suitable for local, on-site production[17]. In contrast, the oxidative dehydrogenation of propane using molecular oxygen (O₂) or carbon dioxide (CO₂) as oxidant is more thermodynamically favorable[18–22]. It is also believe that the oxidative dehydrogenation process can overcome the equilibrium limitations and avoid thermal cracking[23]. Nevertheless, it still suffers from decreasing selectivity with increasing conversion and faces other challenges such as overoxidation[24]. More importantly, despite decades of development, there are limited researches reporting propane conversion to C₃ chemicals beyond propene[25,26]. It remains a challenge to realize efficient propane activation at moderate conditions and to directly convert propane to C₃ fine chemicals such as acetone in a high selectivity.

[1]Deep Space Exploration Laboratory, Hefei, Anhui 230088, P. R. China. [2]Hefei National Research Center for Physical Sciences at the Microscale, University of Science and Technology of China, Hefei, Anhui 230026, P. R. China. [3]Department of Applied Chemistry, School of Engineering, University of Toyama, Gofuku 3190, Toyama 930-8555, Japan. [4]National Synchrotron Radiation Laboratory, University of Science and Technology of China, Hefei, Anhui 230029, China. [5]School of Chemistry & Chemical Engineering, Anhui University of Technology, Ma'anshan, Anhui 243002, P. R. China. [6]These authors contributed equally: Xinlong Ma, Haibin Yin, Zhengtian Pu. ✉e-mail: llh0214@ustc.edu.cn; lihl@ustc.edu.cn; zengj@ustc.edu.cn

a. Industrial route for acetone synthesis

b. Alternative routes for acetone synthesis

c. Acetone synthesis via propane wet reforming (this work)

**Fig. 1 | Strategies for acetone synthesis. a** Industrial cumene route. **b** Alternative routes reported recently including propene hydration and isopropanol oxidation. **c** Propane wet reforming route proposed by this work.

Herein, we report an atom-economic catalytic process for acetone synthesis by directly converting propane and water into acetone and hydrogen ($C_3H_8 + H_2O \rightarrow C_3H_6O + 2H_2$) (Fig. 1c). This process can be conducted at a moderate temperature of 350°C with the use of platinum-tin nanoparticles supported on γ-aluminum oxide (PtSn/γ-Al$_2$O$_3$) as catalyst. We achieve an acetone productivity of 858.4 μmol/g with a selectivity of 57.8% among all carbon-based products and 99.3% among all liquid products. Mechanistic studies reveal that three tandem steps are involved to realize acetone production over different catalytic sites. Specifically, propane is dehydrogenated into propene over PtSn nanoparticles. Then, the generated propene is hydrated over γ-Al$_2$O$_3$ and isopropanol forms. Lastly, the isopropanol is dehydrogenated into acetone over PtSn nanoparticles. Further numeric simulations reveal that the propane wet reforming process exceeds the thermodynamic limit for non-oxidative propane dehydrogenation at the same temperature and partial pressure conditions. We also showcase the capability of facile ketone synthesis via wet reforming with the use of different alkanes (*e.g.* n-butane, n-pentane, n-hexane, n-heptane, and n-octane) as substrates, proving the wide extensibility of this strategy.

## Results and discussion
### Synthesis and characterization of catalysts
We prepared PtSn/γ-Al$_2$O$_3$ via an impregnation-quench method (see Method section for details). After loading Pt- and Sn-based precursors on the γ-Al$_2$O$_3$ support by incipient wetness impregnation, the obtained solid mixtures were dried and reduced under a H$_2$ gas flow at 750 °C for 2 hours. A modified tube furnace was designed to realize a fast quench of the catalyst after annealing (Supplementary Fig. 1). Based on inductively coupled plasma-atomic emission spectroscopy (ICP-AES) analysis, the mass loadings of Pt and Sn were measured as 2.71 and 1.73 wt%, respectively, corresponding to a Pt:Sn molar ratio of 1:1 (Supplementary Table 1). Figure 2a shows the high-angle annular dark-field scanning transmission electron microscopy (HAADF-STEM)

image of PtSn/γ-Al$_2$O$_3$. The metal nanoparticles were uniformly dispersed with an average size of 1.74 nm (Fig. 2b, Supplementary Figs. 2a and 3). X-ray powder diffraction (XRD) pattern only exhibited characteristic peaks for Al$_2$O$_3$ in the absence of peaks for PtSn nanocrystals (Supplementary Fig. 4). Elemental analysis of PtSn/γ-Al$_2$O$_3$ indicated the distribution of Sn on the γ-Al$_2$O$_3$ support with neighboring Pt component (Fig. 2c). For comparison, we synthesized Pt nanoparticles with an average size of 1.31 nm supported on γ-Al$_2$O$_3$ (Pt/γ-Al$_2$O$_3$) via the same method but only Pt-based precursor was adopted (Supplementary Figs. 2b, 4, and 5).

We carried out the X-ray absorption near-edge spectroscopy (XANES) and extended X-ray absorption fine structure (EXAFS) analysis to explore the electronic and coordination structures of PtSn/γ-Al$_2$O$_3$. Based on the Pt L$_3$-edge XANES spectra, we found the Pt species existed in a lower valence state in PtSn/γ-Al$_2$O$_3$ than Pt/γ-Al$_2$O$_3$ (Fig. 2d). The EXAFS fitting results indicated the existence of Pt-O bonds in PtSn nanoparticles with a coordination number (CN) of 0.3. We also identified CNs of 1.6 and 4.3 for Pt-Sn and Pt-Pt bond, respectively (Supplementary Figs. 6 and 7, Supplementary Table 2). In addition, the CNs of Pt-O and Pt-Pt bonds for Pt/γ-Al$_2$O$_3$ were 1.7 and 4.8, respectively (Supplementary Table 2). The smaller Pt-O CN in PtSn than Pt nanoparticles was presumably derived from the competitive oxidation of surface Sn atoms. The bond length of Pt-Pt (2.73 Å) in Pt/γ-Al$_2$O$_3$ was close to those of PtSn (2.70 Å) and Pt-Pt (2.74 Å) in PtSn/γ-Al$_2$O$_3$. We speculate that the enlarged size of PtSn than Pt nanoparticles was not due to the lattice expansion induced by the introduction of Sn but owing to the aggregation of more atoms (such as Sn atoms) in an individual particle. We further performed analysis on the coordination of Sn component in PtSn/γ-Al$_2$O$_3$ and only Sn-O coordination was observed (Supplementary Fig. 8 and Supplementary Table 3). Therefore, we suspect that not all the Pt and Sn atoms were uniformly mixed to form PtSn bimetallic nanoparticles. Instead, a certain proportion of Sn atoms were oxidized to form SnO$_x$ nanoparticles.

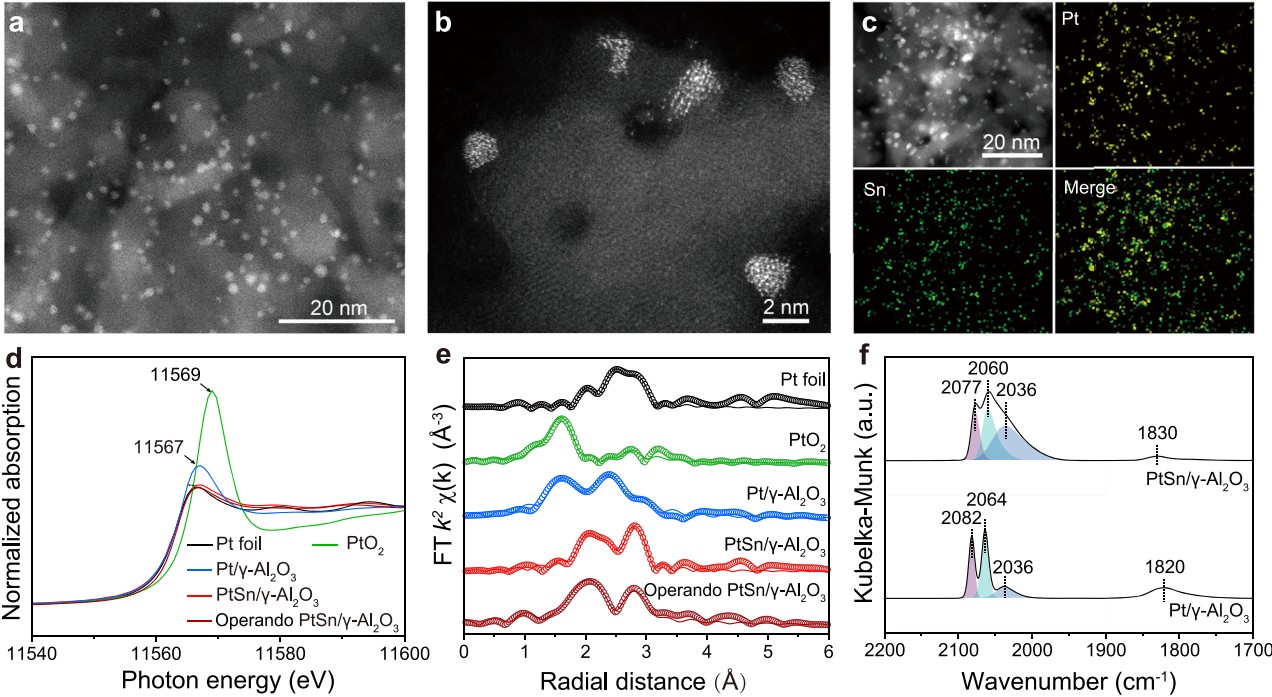

**Fig. 2 | Structural characterizations of PtSn/γ-Al₂O₃. a** HAADF-STEM image and **b** magnified HAADF-STEM image of PtSn/γ-Al₂O₃. **c** EDS elemental mapping images of PtSn/γ-Al₂O₃. **d** Pt L₃-edge XANES spectra for PtSn/γ-Al₂O₃, operando PtSn/γ-Al₂O₃, and Pt/γ-Al₂O₃. Pt foil was used as a reference. **e** The corresponding Pt L₃-edge EXAFS spectra (circles) and curve fits (lines) in R space for PtSn/γ-Al₂O₃, operando PtSn/γ-Al₂O₃, and Pt/γ-Al₂O₃. Pt foil was used as a reference. **f** CO DRIFTS spectra for PtSn/γ-Al₂O₃ and Pt/γ-Al₂O₃. Source data are provided as a Source Data file.

We further performed diffuse reflectance infrared Fourier transform spectrometry (DRIFTS) measurements using carbon monoxide (CO) as a probe molecule to explore the coordination environment of Pt in PtSn/γ-Al₂O₃. As shown in Fig. 2f, the peaks ranging from 2030 to 2100 cm⁻¹ were assigned to linear adsorption of CO molecules on the top sites of Pt atoms with different coordination structures (Supplementary Table 4). The adsorption peak with a lower wavenumber indicates a stronger adsorption strength and a higher under-coordinated Pt state[27]. The peaks at 1830 or 1820 cm⁻¹ corresponded to the adsorption of CO molecules on the bridge sites, typical characteristics for Pt-Pt ensembles[23] (Fig. 2f, Supplementary Table 4). To conclude, PtSn/γ-Al₂O₃ exhibited more coordinately unsaturated surface Pt sites and less prominent Pt-Pt ensembles compared to Pt/γ-Al₂O₃.

## Catalytic properties

PtSn/γ-Al₂O₃ was adopted for catalyzing the propane wet reforming reaction. Specifically, 25 mg of catalysts and 5 mL of water were loaded in a 15-mL slurry reactor which is charged with 6-bar gaseous mixture of propane and nitrogen (N₂). The ratio of C₃H₈ and N₂ is set as 5:1. The test was conducted at 350 °C for 2 h. After reaction, the gaseous and liquid products were quantified through corresponding standard curves by gas chromatography (GC) and nuclear magnetic resonance (NMR), respectively. Hydrogen (H₂) was the main product in the gaseous phase, and byproducts such as methane (CH₄), ethane (C₂H₆), propene (C₃H₆), and CO₂ were also identified (Supplementary Fig. 9, Supplementary Tables 5 and 6). Acetone was the main product in the liquid phase. We also observed a minor fraction of isopropanol (Supplementary Fig. 10, Supplementary Table 7). The total yield of carbon-based products was calculated to be 1484.9 μmol g⁻¹ (Fig. 3a). Acetone exhibited a unexpected selectivity of 57.8% among all carbon-based products and 99.3% among all liquid products. It was also worth noting that only isomerized C₃ oxygenates were obtained. The absence of normal C₃ oxygenates such as n-propanol or propyl aldehyde can be

rationalized by the Markovnikov rule[28]. We further calculated the turnover frequency (TOF) number of PtSn/γ-Al₂O₃ (41.3 h⁻¹) by quantifying surface Pt atoms via CO titration experiments (Supplementary Figs. 11 and 12, Supplementary Note 1, and Supplementary Table 8, see supplementary information for detailed discussion). The propane conversion reached 1.84%. Moreover, when we decreased the partial pressure of propane, the propane conversion further increased to 25.60% (Supplementary Table 9 and Supplementary Note 2). Further simulations revealed that the thermodynamic equilibrium conversion of propane into acetone increases with the decrease of propane partial pressure, which is consistent with the experimental results (Supplementary Fig. 13, Supplementary Note 3, and Supplementary Table 10).

We synthesized Sn nanoparticles supported on γ-Al₂O₃ (Sn/γ-Al₂O₃) to explore the role of Sn in acetone synthesis. Under the same catalytic conditions, negligible products were observed when Sn/γ-Al₂O₃ was adopted as catalyst (Supplementary Fig. 14). This is consistent with previous research on propane dehydrogenation, where Sn primarily acts to dilute Pt and modulate the electronic structure[29]. We also tuned the feeding ratio of Pt:Sn ranging from 5:1 to 0.4:1 and found that highest acetone yield was achieved at the ratio of 1:1 (Supplementary Fig. 15). Meanwhile, we found that with increased Sn loading, the selectivity for total C₃ products including acetone, isopropanol, and C₃H₆ increased. Pt/γ-Al₂O₃ exhibited a higher total yield of 5272.3 μmol g⁻¹ and conversion of 6.53%. However, the products were either over-cracked or over-oxidized (Fig. 3a). To make a fair comparison on the selectivity at a comparable conversion, we shortened the reaction time to 10 min over Pt/γ-Al₂O₃. The propane conversion decreased to 1.25% and the TOF number was calculated to be 70.7 h⁻¹. Despite the decrease of the conversion, the major product was still ethane with a selectivity of 68.7%. The selectivity of CO₂, methane, and acetone were 22.4%, 8.0%, and 0.9%, respectively (Supplementary Fig. 16). Meanwhile, the catalytic results by physically mixing Pt/γ-Al₂O₃ and Sn/γ-Al₂O₃ catalyst resembles that of using Pt/γ-Al₂O₃ solely, indicating the

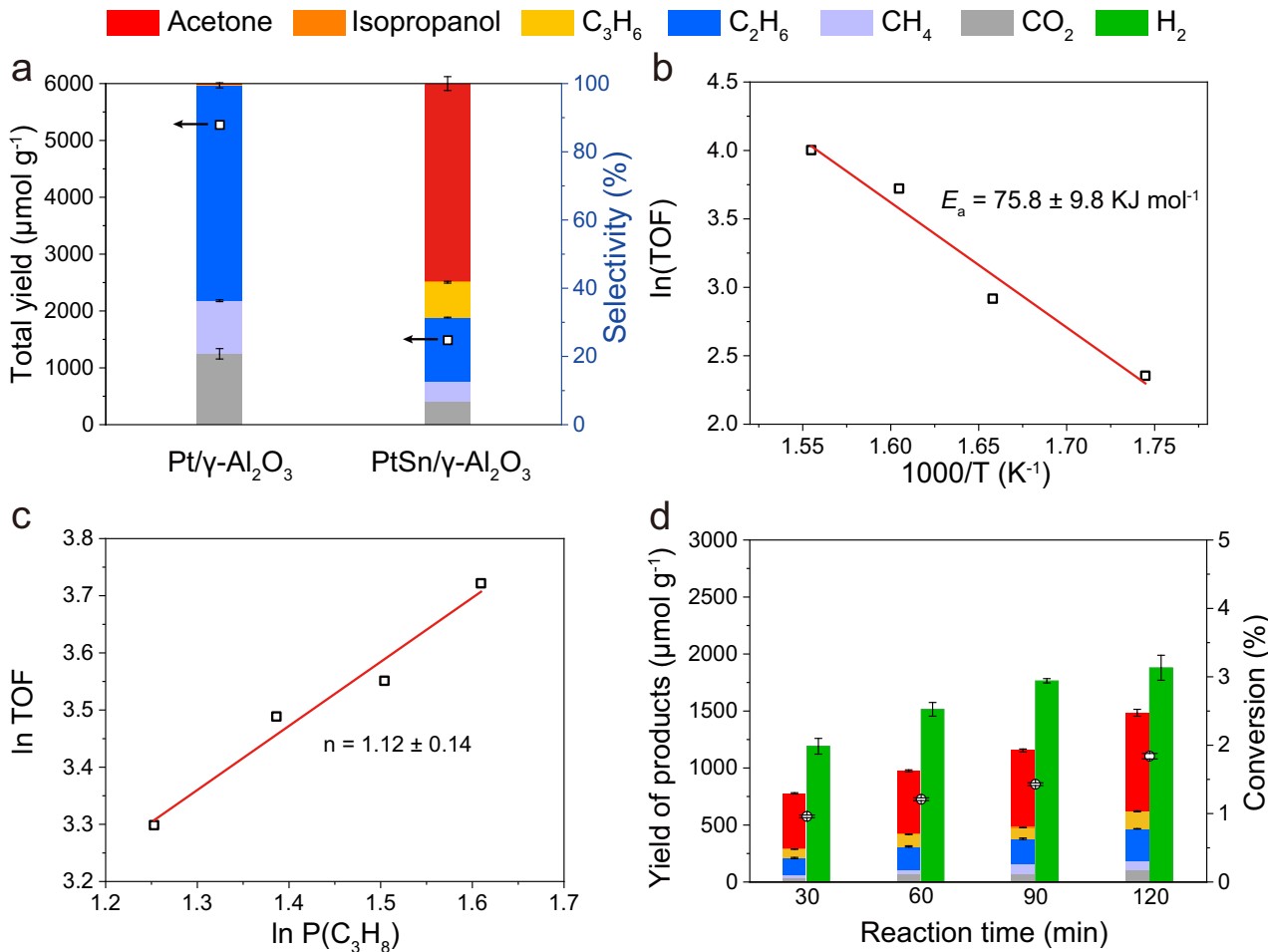

**Fig. 3 | Catalytic performance for propane wet reforming reaction. a** Total yield and product selectivity over different catalysts. Reaction conditions: 5 mL of water, 6 bar of gaseous mixture ($C_3H_8:N_2 = 5:1$), 350 °C, 2 h. **b** Arrhenius plot over PtSn/γ-$Al_2O_3$. **c** Kinetic plot of ln TOF versus ln $P(C_3H_8)$ over PtSn/γ-$Al_2O_3$. **d** Time course of the product evolution over PtSn/γ-$Al_2O_3$. **a**, **d** error bars represent the standard deviation from three independent measurements. Source data are provided as a Source Data file.

importance of a close contact between Pt and Sn components (Supplementary Fig. 17). As such, PtSn/γ-$Al_2O_3$ efficiently restrained the cracking of C-C bonds compared with Pt/γ-$Al_2O_3$, which was consistent with previous reports[30–32]. This point was further verified by Raman characterizations of the catalysts after the reaction (denoted as spent PtSn/γ-$Al_2O_3$ and spent Pt/γ-$Al_2O_3$). As shown in Supplementary Fig. 18, the coke signals were negligible for spent PtSn/γ-$Al_2O_3$. The Raman spectrum of spent Pt/γ-$Al_2O_3$ showed prominent peaks at 1340 and 1583 $cm^{-1}$, which we assigned to disordered carbon (D band) and graphite (G band), respectively[12]. The coking process led to the extra generation of $H_2$, resulting in a non-stoichiometric hydrogen yield based on the calculation of detected products. We also performed quantitative analysis on the amount of coke via thermogravimetric measurements and detailed analysis on the carbon balance (Supplementary Fig. 19, Supplementary Table 11, and Supplementary Note 4, see supplementary information for detailed discussions). The unclosed carbon balance together with the excess hydrogen production indicate possible existence of undetected coke species remaining in the solution phase or on the wall of the reactor. Based on the HAADF-STEM images of spent PtSn/γ-$Al_2O_3$ (Supplementary Fig. 20), we observed a uniform distribution of the PtSn nanoparticles. The average size of the nanoparticles was estimated as 1.78 nm (Supplementary Fig. 21), comparable to that (1.74 nm) of the fresh sample. Peaks located at 2082, 2065, and

2040 $cm^{-1}$ were observed in the CO DRIFTS spectrum of the spent PtSn/γ-$Al_2O_3$ (Supplementary Fig. 22). These peaks were slightly red-shifted comparing to the fresh samples (2077, 2060, and 2036 $cm^{-1}$), which implied a slightly oxidization of Pt species or decreased degree of unsaturation. Analysis based on the XANES results indicated a higher oxidation state of Pt species in spent PtSn/γ-$Al_2O_3$ than that in the fresh one (Supplementary Fig. 23) and the EXAFS results also showed a larger Pt-O coordination number ($0.8 \pm 0.1$) in the spent sample (Supplementary Fig. 23, Tables 12 and 13). These results were also verified by the operando XAS characterizations (Fig. 2d, e, Supplementary Fig. 24). Under-operando conditions, a slightly higher CN (0.5) of Pt-O bond was observed, compared to that (0.3) derived from the fresh sample. Meanwhile, the CN of Pt-Pt bond was decreased (Supplementary Tables 14 and 15). To further reveal the influence of Pt oxidation to the catalytic performance, we evaluated the stability of the catalyst in a cyclic manner. As shown in Supplementary Fig. 25, after 5 cycles, we observed a slight decrease by 14.0% of the acetone yield, from 1012.0 to 870.6 μmol $g^{-1}$. We ascribed the decay in the catalytic performance to the oxidation of surface metallic Pt species as revealed by CO DRIFTS and XAFS characterizations. We also observed a poor catalytic performance when deliberately oxidized PtSn/γ-$Al_2O_3$ was adopted as catalyst (Supplementary Fig. 26).

## Kinetic studies

We further investigated the kinetic properties of the wet reforming process over PtSn/γ-Al$_2$O$_3$. We varied the stirring speeds (200, 400, and 600 rpm) during the reaction and found the conversions and product distributions remained unchanged (Supplementary Fig. 27). As such, the reaction kinetics was not determined by external diffusion. We also found that the conversion and selectivity were insensitive to the catalyst size (20–40 mesh, 40–60 mesh, and >60 mesh), indicating the negligible influence of internal diffusion (Supplementary Fig. 28). To this end, the reaction was under kinetic control instead of diffusion control.

The dependence of performance on reaction temperatures was explored by carrying out the reaction at different temperatures under 6 bar of gaseous mixture (C$_3$H$_8$:N$_2$ = 5:1) for 2 h. With the increase in temperature, the total product yield increased. Meanwhile, the selectivity for cracking products (e.g., ethane and methane) also increased since these reactions were highly endothermal and were favored at higher temperatures (Supplementary Fig. 29, Supplementary Table 16). The activation energy of PtSn/γ-Al$_2$O$_3$ was determined to be 75.8 kJ/mol based on the Arrhenius plot analysis (Fig. 3b). We further measured the reaction orders. TOF numbers were evaluated under different propane partial pressures with N$_2$ as the balance gas and the temperature was fixed to be 350 °C (Supplementary Fig. 30). As shown in Fig. 3c, we derived a reaction order of 1.12 with respect to propane from the fitting result. The pseudo-first-order characteristic results in an inapparent increase of the conversion with increasing the partial pressure of propane (Supplementary Note 5). The equilibrium conversion of propane also varies slightly with the change of propane partial pressure at a low conversion level (Supplementary Fig. 13).

We changed the volume of water to explore the kinetic dependence on water. As shown in Supplementary Fig. 31, the slight variation in water volume from 5.00 mL to 4.90 mL induced a minor increase in the conversion from 1.84% to 2.15%. We suspect that the minor increase in acetone yield was ascribed to the increased amount of propane molecules, considering the fixed total volume of the reactor and total pressure. The decrease in the water volume leads to the increase in the feeding amount of propane in the reactor. When the water volume was further changed from 4.90 mL to 4.80 mL, the conversion slightly decreased to 2.05%, indicating a possible decreasing trend in the conversion of propane. Considering that the H$_2$O:C$_3$H$_8$ ratio (135:1) used in the reaction was much higher than the stoichiometric ratio, we further conducted the reaction by maintaining a H$_2$O:C$_3$H$_8$ molar ratio of 1:1. As shown in Supplementary Fig. 32, we observed the production of a large proportion of propene at significantly lower H$_2$O:C$_3$H$_8$ ratios. Due to the limited water content (approximately 50 μL), the catalyst was not fully infiltrated by water, let alone thoroughly stirred, resulting in the incomplete conversion of propene to acetone. As we gradually increased the water dosage from 50 μL to 100 μL and up to 1 mL, the yields of acetone and H$_2$ both increased. This implies the crucial role of water partial pressure, attributed to its ability to enhance mass and heat transfer processes. Notably, at a H$_2$O:C$_3$H$_8$ ratio of 1:1, the acetone concentration was approximately 22.3 μmol mL$^{-1}$, while increasing the H$_2$O:C$_3$H$_8$ ratio to 2:1 resulted in a noticeable decrease to 14.4 μmol mL$^{-1}$. When the H$_2$O:C$_3$H$_8$ ratio was approximately 135:1, the concentration of acetone further decreased to 4.3 μmol mL$^{-1}$. This underscores the pivotal role of water in efficient mass transfer, reducing the product concentration on the catalyst surface and consequently promoting the progress of the catalytic reaction. Meanwhile, we observed a significant increase in the yield of propene, indicating that the dehydrogenation of propane can be enhanced under high water partial pressure, providing more intermediates for the subsequent conversion of propene into acetone.

As the water content gradually increased from stoichiometric value to excess, the yield of acetone increased from 44.5 μmol g$^{-1}$ to 858.4 μmol g$^{-1}$, and the conversion also increased from 0.17% to 1.84%

(Supplementary Table 17). We also performed experiments with significantly decreased catalyst amount and extended reaction time to evaluate the influence of contact between the catalyst and reactants. As shown in Supplementary Fig. 33 reducing the catalyst amount and extending the reaction time increased the total yield of products and the selectivity for propene. A decreased reaction rate and the selectivity for acetone was also observed. This phenomenon can be rationalized by the decreased collision probability between the reactant and catalyst and exactly revealed that propene as an important intermediate for acetone formation. The dependence of catalytic performance and the amount of catalyst is also revealed by previously reported works[33]. Nevertheless, the observation of massive production of post-reacted products (e.g., acetone and ethane) evidently showed sufficient contact between reactant or intermediates and the catalysts, and fairly reflect the influence of water ratio (Supplementary Table 17). Of note, there are several reports that suggest a co-feeding of water steam promotes propane dehydrogenation in increasing reaction rate and decreasing apparent activation energy[34–36]. Together with the improved yield of acetone, we suspect that both propane dehydrogenation and possible sequential propene hydration process are promoted by water.

In addition, we analyzed the kinetic isotope effect (KIE) of the reaction by reacting propane with deuterium oxide (D$_2$O). The KIE value refers to the ratio of the reaction rate using H$_2$O to that using D$_2$O ($k_{H2O}/k_{D2O}$) was determined to be 4.46 (Supplementary Fig. 34 and Supplementary Note 6), indicating that the breaking or the formation of chemical bonds related to D atoms was involved in the rate-limiting step. The evolution of the products during the wet reforming process was presented in Fig. 3d. With the extension of reaction time, the yield of propene remained almost unchanged, whereas all the other products such as acetone and H$_2$ steadily increased. As such, propene is believed to be quickly consumed to generate downstream products without accumulating. This result indicated that the dehydrogenation of propane into propene was possibly the rate-limiting step.

## Mechanistic insights

Since propene has been identified as an important intermediate, we tend to decouple the propane wet reforming into tandem steps including propane dehydrogenation into propene and propene conversion into oxygen-containing products. Dry propane-pulse experiments were performed to reveal the interaction between propane and the catalysts. Specifically, the catalysts were pre-treated under 1 bar of Helium (He) at 350 °C for 0.5 h, followed by pulsing dry propane into the reactor. For PtSn/γ-Al$_2$O$_3$, the ratio of the C$_3$H$_6$/C$_3$H$_8$ peak areas was 0.77, much higher than that of 0.11 for C$_2$H$_6$/C$_3$H$_8$ and 0.03 for CH$_4$/C$_3$H$_8$ (Fig. 4a, Supplementary Table 18). As such, the cracking of propane was restrained over PtSn/γ-Al$_2$O$_3$. With regard to Pt/γ-Al$_2$O$_3$, more cracked products (e.g., CH$_4$ and C$_2$H$_6$) were generated (Fig. 4b, Supplementary Table 18). Meanwhile, γ-Al$_2$O$_3$ was inactive towards the dehydrogenation of propane as evidenced by the negligible signals observed for relevant products (Fig. 4b, Supplementary Table 18). Therefore, Pt atoms contribute to the dehydrogenation of propane, while the dilution with Sn atoms to avoid the formation of contiguous Pt ensembles mitigates the over-dehydrogenation of propene and its hydrogenolysis[30–32]. When we pulsed wet propane over PtSn/γ-Al$_2$O$_3$, we observed the product of isopropanol and negligible acetone (Supplementary Fig. 35).

The hydration of propene was explored by wet propene-pulse experiments. Specifically, the samples were pre-treated under 1 bar of He at 350 °C for 0.5 h, followed by pulsing wet propene through bubbling into the reactor. The major product was acetone over PtSn/γ-Al$_2$O$_3$ and we also observed the production of propane via the reversible hydrogenation of propene (Fig. 4d and Supplementary Fig. 36a). As for Pt/γ-Al$_2$O$_3$, most of the propene was reversibly hydrogenated into propane, along with the cracking products such as C$_2$H$_6$ and CO$_2$

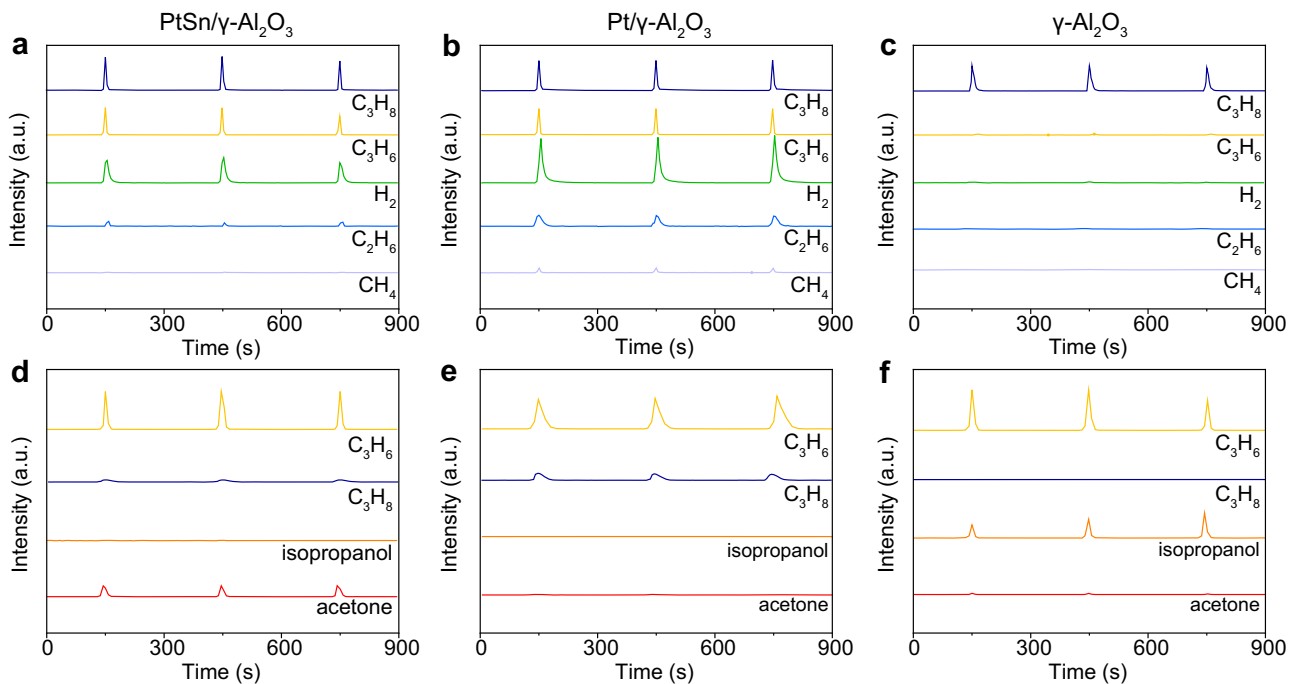

**Fig. 4 | Mass spectroscopic studies.** Transient response curves of **a** PtSn/γ-Al$_2$O$_3$, **b** Pt/γ-Al$_2$O$_3$, and **c** γ-Al$_2$O$_3$ obtained during the C$_3$H$_8$-pulse measurements. Transient response curves of **d** PtSn/γ-Al$_2$O$_3$, **e** Pt/γ-Al$_2$O$_3$, and **f** γ-Al$_2$O$_3$ obtained during the C$_3$H$_6$/H$_2$O-pulse measurements. Source data are provided as a Source Data file.

(Fig. 4e and Supplementary Fig. 36b). γ-Al$_2$O$_3$ enabled the oxidation of propene into isopropanol but was unable to catalyze the dehydrogenation of isopropanol into acetone (Fig. 4f and Supplementary Fig. 36c). Therefore, the hydration of propene happens on the γ-Al$_2$O$_3$ support, instead of the metallic components.

Then, we investigated the dehydrogenation of isopropanol via in-situ DRIFTS experiments. The catalysts were pre-treated by He at 350 °C for 1 h. Isopropanol steam was introduced to the in-situ cell via He bubbling. The time-dependent in-situ DRIFTS spectra over PtSn/γ-Al$_2$O$_3$ exhibited characteristic peaks for isopropanol and acetone (Fig. 5a and Supplementary Fig. 36d–f). Specifically, the peaks located at 3527, 2970, 2887, 1474, 1381, and 1230 cm$^{-1}$ represent the stretch of hydroxyl group (OH*), the asymmetric stretch of methyl (CH$_3$*), the symmetric stretch of CH$_3$*, the asymmetric bending vibration of CH$_3$*, the symmetric bending vibration of CH$_3$*, and the bending vibration of OH*, respectively, for the adsorbed isopropanol species (Supplementary Table 19)[8,37–39]. The typical peak for the adsorbed acetone was observed at 1732 cm$^{-1}$, representing the stretch of C=O bonds (Supplementary Table 19). With the extension of reaction time, the significantly increased peak intensity for 1732 cm$^{-1}$ indicated the continue formation of acetone (Fig. 5a). As for Pt/γ-Al$_2$O$_3$, the peaks for the symmetric stretches of OH* and CH$_3$* were weakened, whereas the peak for the asymmetric bending vibration of CH$_3$* was strengthened (Fig. 5b). We suspect that isopropanol adsorbs in a different configuration on Pt/γ-Al$_2$O$_3$. Negligible peaks can be found around 1732 cm$^{-1}$, indicating the absence of adsorbed acetone species. At the same time, a peak located at 1576 cm$^{-1}$ emerged, representing for the asymmetric stretch of carboxylate (Fig. 5b). In this case, we suspect that isopropanol was cracked and over-oxidized on Pt/γ-Al$_2$O$_3$. The in-situ DRIFTS spectra of γ-Al$_2$O$_3$ showed prominent peaks for isopropanol and weak peaks for adsorbed acetone species (Fig. 5c), revealing a much lower dehydrogenation capability than the PtSn nanoparticles. The results from in-situ DRIFTS experiments were consistent with those from the gaseous pulse tests. To conclude, PtSn nanoparticles were responsible for the dehydrogenation of isopropanol into acetone.

We further conducted controlled experiments to verify the proposed tandem mechanism. The performance of propane

dehydrogenation were evaluated over 25 mg of catalysts under 6 bar of gaseous mixture (C$_3$H$_8$:N$_2$ = 5:1) at 350 °C for 2 h without using water as solvent. As shown in Fig. 6a, γ-Al$_2$O$_3$ support was inert for propane dehydrogenation. PtSn/γ-Al$_2$O$_3$ exhibited a high selectivity (89.1%) towards propene. In comparison, Pt/γ-Al$_2$O$_3$ led to the production of cracking products. Therefore, the active sites for the propane dehydrogenation are Pt atoms, and highly dispersed Pt atoms restrict the breaking of C−C bond. The hydration of propene was explored over 25 mg of catalysts in 5 mL of water under 6 bar of gaseous mixture (C$_3$H$_6$:N$_2$ = 5:1) at 350 °C for 2 h. As shown in Fig. 6b, γ-Al$_2$O$_3$ enabled the highly selective hydration of propene into isopropanol. We also performed KIE analysis[40] on this reaction (Supplementary Fig. 37). The KIE value was determined to be 3.0, indicating that the breaking of D-O bonds was involved in the rate-limiting step (Supplementary Fig. 38). As for Pt/γ-Al$_2$O$_3$, the hydrogenation of propene overwhelmed the hydration, resulting in the generation of propane as the main product (Fig. 6b). Side reactions including dehydrogenation of isopropanol and steam reforming provided the hydrogen sources (Supplementary Table 20). With regard to PtSn/γ-Al$_2$O$_3$, acetone and propane composed the majority of the products. Negligible isopropanol was observed (Fig. 6b). To further reveal the importance of γ-Al$_2$O$_3$ in catalyzing the hydration process, we replaced γ-Al$_2$O$_3$ with SiO$_2$ to support PtSn and Pt nanoparticles. The corresponding catalysts were denoted as PtSn/SiO$_2$ and Pt/SiO$_2$, respectively. As shown in Supplementary Fig. 39, both the SiO$_2$-supported catalysts catalyzed the reverse hydrogenation of propene into propane instead of the formation of isopropanol or acetone. Therefore, we can confirm that γ-Al$_2$O$_3$ provides active sites for the hydration of propene into isopropanol.

The conversion of isopropanol into acetone was explored by loading 25 mg of PtSn/γ-Al$_2$O$_3$ and 4.95 mL of water with 0.05 mL of dissolved isopropanol in the reactor. The tests were conducted under 6 bar N$_2$ at 350 °C for 2 h. As shown in Fig. 6c, when γ-Al$_2$O$_3$ was used as catalyst, the selectivity of acetone was only 35.8%, while the propene selectivity reached 50.4%, implying that the hydration of propene into isopropanol was reversible. The use of Pt/γ-Al$_2$O$_3$ resulted in abundant cracking products including ethane and methane (Fig. 6c). In contrast,

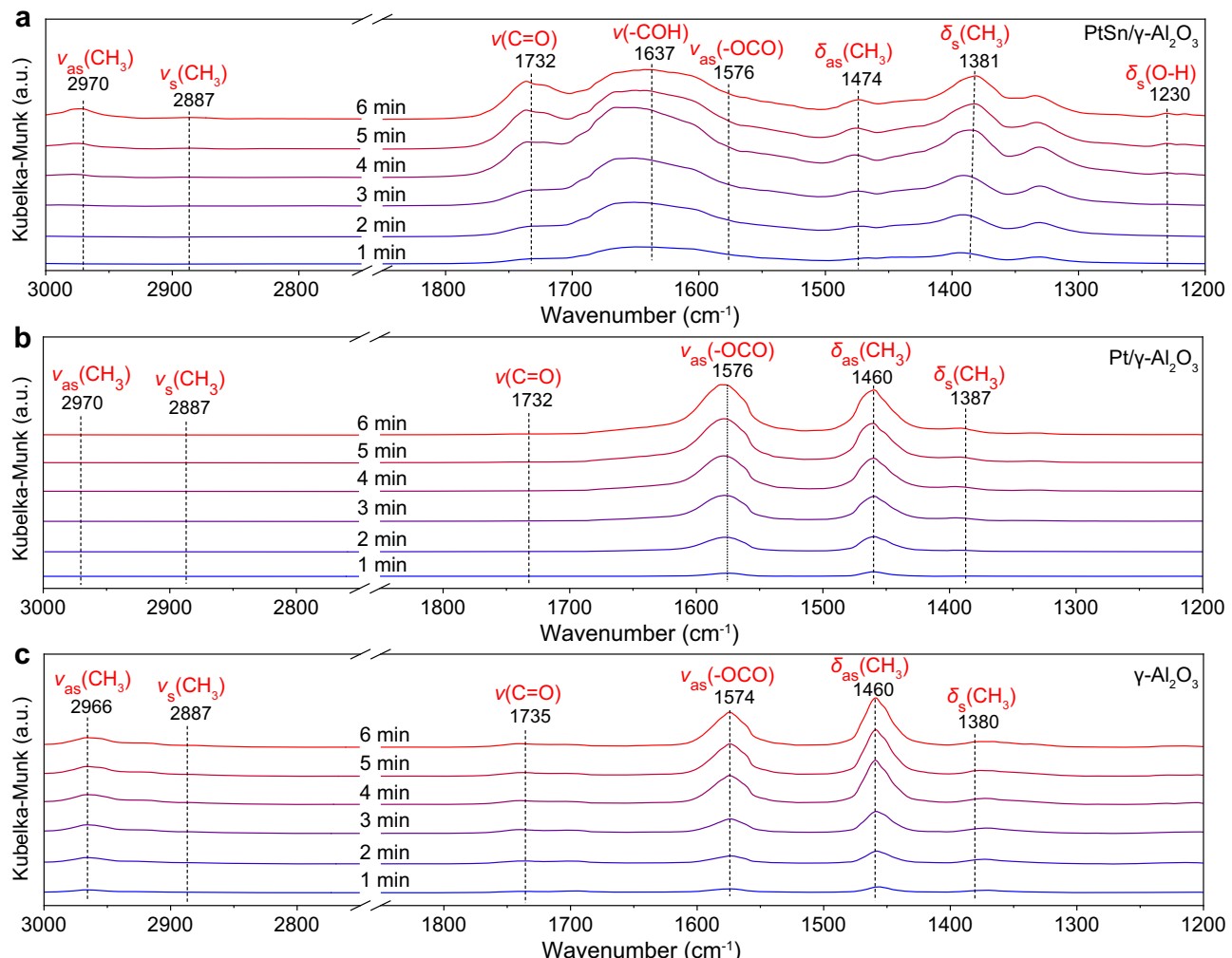

**Fig. 5 | Time-dependent in-situ DRIFTS studies.** Time-dependent in-situ DRIFTS spectra of isopropanol dehydrogenation over **a** PtSn/γ-Al$_2$O$_3$, **b** Pt/γ-Al$_2$O$_3$, and **c** γ-Al$_2$O$_3$. The spectra were obtained by purging He flow with isopropanol steam at 350 °C. Background spectra were acquired after purging 1 bar of He with a gas-flow rate of 20 mL min$^{-1}$ at 350 °C for 0.5 h. Source data are provided as a Source Data file.

PtSn/γ-Al$_2$O$_3$ achieved a high acetone selectivity of 85.8% (Fig. 6c). In this case, the dehydrogenation process over PtSn/γ-Al$_2$O$_3$ overwhelms the cracking, dehydration, and hydrogenation processes. We also found that the decomposition of acetone was more difficult over PtSn/γ-Al$_2$O$_3$ than Pt/γ-Al$_2$O$_3$, indicating that the dilution of Pt atoms with Sn atoms can also significantly mitigate the acetone cracking process (Supplementary Fig. 40).

Based on the above analysis, we derived a scheme for the mechanism of the propane wet reforming process (Fig. 6d). As illustrated, propane is initially dehydrogenated into propene over PtSn nanoparticles. The formed propene undergoes hydration over γ-Al$_2$O$_3$ and isopropanol forms. Then, isopropanol is dehydrogenated into acetone over PtSn nanoparticles. During the whole process, the highly dispersed Pt atoms due to the dilution with Sn efficiently restrict the cracking process.

**The universality of the strategy**

We demonstrated the wide applicability of the wet reforming strategy for ketone synthesis. A wide range of alkane substrates including n-butane (C$_4$H$_{10}$), n-pentane (C$_5$H$_{12}$), n-hexane (C$_6$H$_{14}$), n-heptane (C$_7$H$_{16}$), and n-octane (C$_8$H$_{18}$) were adopted as reactants. Figure 6e shows the yields of oxygenate products (Supplementary Figs. 41–43, Supplementary Tables 21–25). With the elongation of chain length, isomerization occurred (Fig. 6e). Of note, as a proof-of-concept study,

the conditions applied for the wet reforming of these alkanes are not the optimal. Nevertheless, the wet reforming strategy we proposed here provides a sustainable alternative to conventional industrial processes (e.g., dry reforming and dehydrogenation) for alkanes upgrading and ketone synthesis. Future efforts towards catalyst design by further enhancing the intrinsic reaction rate, avoiding coke formation, and facilitating mass transport process via microenvironment tuning are still needed to improve the yield and meet the requirements for industrial applications.

## Methods

### Chemicals and materials

H$_2$PtCl$_6$ · 6H$_2$O (>37.5% Pt), γ-Al$_2$O$_3$ (99.99%, 20 nm), n-hexane (C$_6$H$_{14}$, 99.5%, GC), n-heptane (C$_7$H$_{16}$, 99.0%, GC), n-octane (C$_8$H$_{18}$, 99.5%, GC), D$_2$O (99.9% D) were purchased from Shanghai Aladdin Biochemical Technology Co., Ltd. SnCl$_2$ · 2H$_2$O, n-pentane (C$_5$H$_{12}$), isopropanol, acetone and other chemicals were analytical grade and purchased from Sinopharm Chemical Reagent Co., Ltd. Propane (C$_3$H$_8$, 99.9 vol.%), propene (C$_3$H$_6$, 99.9 vol.%), n-butane (C$_4$H$_{10}$, 79.5 vol.%, balanced with Ar), CO (10% vol.%, balanced with Ar), and N$_2$ (99.99 vol.%) were purchased from Shanghai Air Liquide Co., Ltd. All materials were used as received without further purification. Deionized water was used for the preparation of all aqueous solutions and the reaction (Milli-Q).

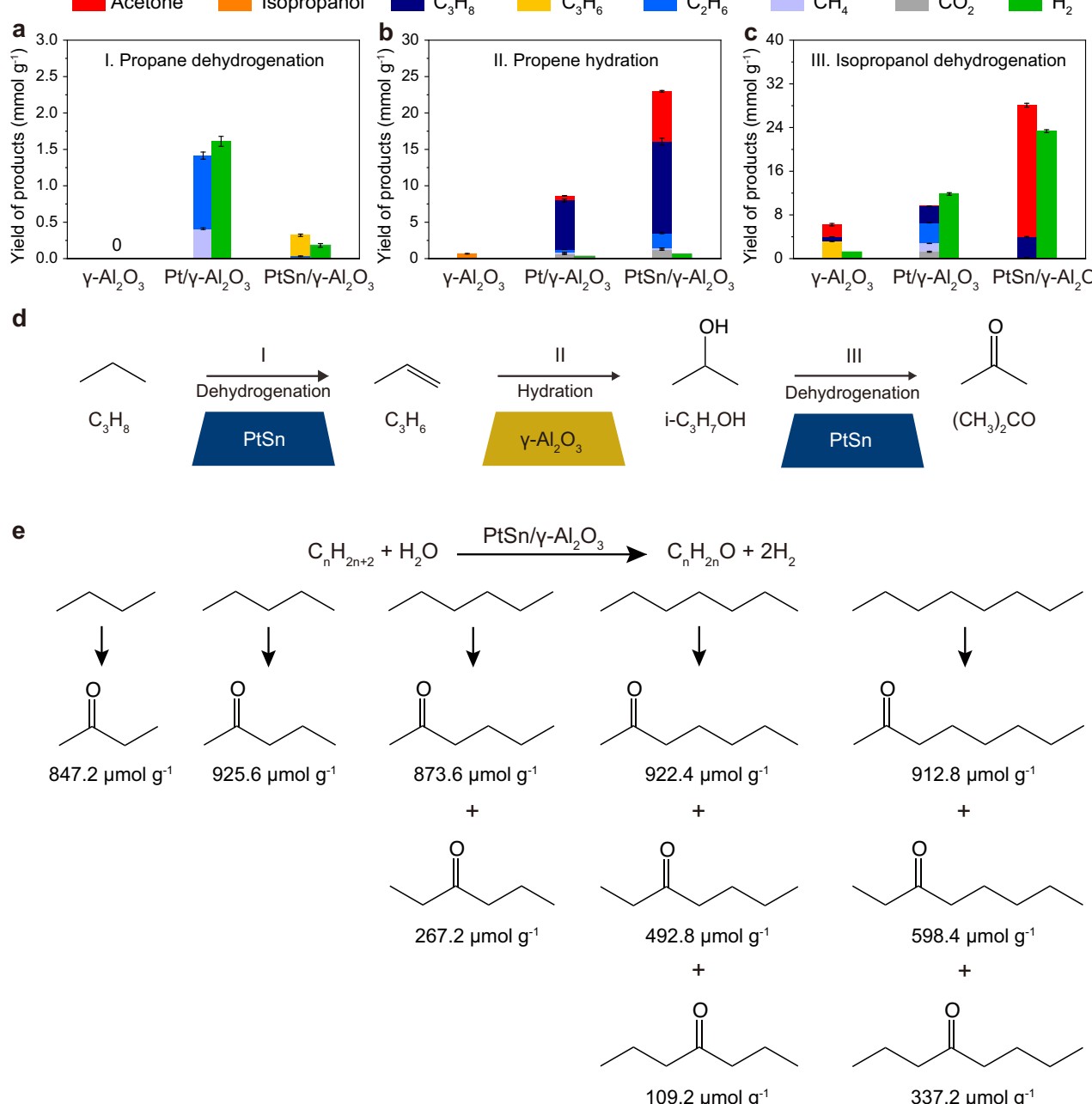

**Fig. 6 | Insights and generality of the wet reforming strategy. a** Yields of products obtained by reacting 6 bar of gaseous mixture ($C_3H_8$:$N_2$ = 5:1) at 350 °C for 2 h without water participation over $\gamma$-$Al_2O_3$, Pt/$\gamma$-$Al_2O_3$, and PtSn/$\gamma$-$Al_2O_3$. **b** Yields of products obtained by reacting 6 bar of gaseous mixture ($C_3H_6$:$N_2$ = 5:1) at 350 °C for 2 h with 5 mL of water over $\gamma$-$Al_2O_3$, Pt/$\gamma$-$Al_2O_3$ and PtSn/$\gamma$-$Al_2O_3$. **c** Yields of products obtained by reacting 4.95 mL of water with 0.05 mL of dissolved isopropanol under 6 bar $N_2$ at 350 °C for 2 h over $\gamma$-$Al_2O_3$, Pt/$\gamma$-$Al_2O_3$, and PtSn/$\gamma$-$Al_2O_3$. **d** Schematic illustration of three consecutive steps for acetone synthesis via propane wet reforming. **e** Schematic illustration of the generality of the wet reforming strategy for ketone synthesis. **a**–**c** error bars represent the standard deviation from three independent measurements. Source data are provided as a Source Data file.

## Synthesis of PtSn/$\gamma$-$Al_2O_3$

In a typical synthesis of PtSn/$\gamma$-$Al_2O_3$, 0.94 g of $H_2PtCl_6$ was dissolved in 17.50 mL of water, and 0.16 mL of HCl was added. The concentration of Pt is thus 20 $mg_{Pt}$ $mL^{-1}$. Then, 15.0 mg of $SnCl_2 \cdot 2H_2O$ was dissolved into 0.65 mL of the above solution. To this end, the concentration of Pt and Sn were 103 mmol $L^{-1}$ and 102 mmol $L^{-1}$, respectively. Finally, 0.50 g of $\gamma$-$Al_2O_3$ support was impregnated with the 0.65 mL mixture with Pt and Sn ions.

Following impregnation, the solid mixture was dried at 80 °C overnight. The dried samples were quickly pushed into the quartz tube in $H_2$ with a gas-flow rate of 10 mL $min^{-1}$. The temperature was elevated to 750 °C and the samples were annealed for 2 h. After annealing process, the tube was quickly pulled out from the furnace and cooled to room temperature. The above fast quench process was realized by our modified tube furnace as shown in Supplementary Fig. 1. When the sample holder was outside the quartz tube, we placed the sample in the end card slot and gave it a short press on multi-function button ⑩. Then the magnetically controlling device pulled the sample into the heating furnace. Afterwards, the tube system was sealed and vacuumized. $H_2$ flow was introduced, followed by programmed heating. The injection push button ⑨ was pressed. After the heating process, we pressed on the multi-function button ⑩ and rapidly pulled the sample

out to the cooling position for the quench process. After cooling down to room temperature, we closed the pipelines, shut off the vacuum system, and opened the seal ring. A long press on multi-function button ⑩ enables extraction of the sample.

### Synthesis of Pt/γ-Al₂O₃

For the synthesis of Pt/γ-Al₂O₃, 0.94 g of H₂PtCl₆ was dissolved into 17.66 mL of water. The concentration of Pt is thus 20 $mg_{Pt}$ $mL^{-1}$. Then, 0.50 g of γ-Al₂O₃ support was impregnated with 0.65 mL of the above solution. Following impregnation, the solid mixtures were dried at 80 °C overnight. The dried samples were quickly pushed into the quartz tube in H₂ with a gas-flow rate of 10 mL $min^{-1}$. The temperature was elevated to 750 °C and the samples were annealed for 2 h.

### Catalytic tests

The wet reforming of propane was conducted in a 15-mL hastelloy slurry reactor (Shanghai Yanzheng Experiment Instrument Co., LTD, China). In a typical catalytic test, the reactor was pressurized with 6 bar of gas mixture (C₃H₈:N₂ = 5:1, corresponding to 2.0 mmol of C₃H₈) at room temperature (25 °C) after the addition of 25 mg of catalysts and 5 mL of water. The reaction proceeded for 2 hours under a magnetic stirring rate of 600 rpm at 350 °C. After the reaction, the reactor was quickly cooled to room temperature. The gas-phase was determined by gas chromatograph (Shimadzu GC-2014) equipped with a thermal conductivity detector (TCD) and a flame ionization detector (FID). The liquid phase of the reaction mixture was collected by centrifugation at 10,956 × $g$ for 3 min. 600 μL of the liquid product was dissolved in 100 μL of D₂O containing 1 μmol of dimethyl sulfoxide (DMSO) as an internal standard to determine the product. Then, the deuterated mixture was analysis by ¹H NMR spectroscopy. The recyclability of the catalysts was studied for five cycles, in which catalysts were collected after being centrifuged, washed, and dried for the next cycle. The conversion, mass yields and selectivity of the products are derived by Eqs. (1), (2), and (3).

$$\text{Conversion} = \frac{\sum_i i \times n_i}{j \times n_j} \times 100\% \tag{1}$$

$$\text{Mass yield} = \frac{i \times n_i}{j \times m_{catal.}} \tag{2}$$

$$\text{Selectivity} = \frac{i \times n_i}{\sum_i i \times n_i} \times 100\% \tag{3}$$

In these equations, $i$ represents to the carbon number of the product. $n_i$ is the mole of the product with the carbon number of $i$. $j$ represents the carbon number of the reactant. $n_j$ is the mole of the reactant. $m_{catal.}$ is the mass of the catalyst. To make an intuitive and equivalent comparison, the mass yields are normalized to the carbon number of the reactant ($j = 3$ for C₃H₈). Typically, PtSn/γ-Al₂O₃ showed an acetone mass yield of 858.4 μmol $g^{-1}$ and selectivity of 57.8%. In comparison, Pt/γ-Al₂O₃ exhibited an acetone mass yield of 9.5 μmol $g^{-1}$ and selectivity of 0.9%.

### CO pulse titration

CO pulse chemisorption experiments were conducted on a VDSorb-9Xi instrument. Specifically, 50 mg of catalysts were used for the test. 10% CO/Ar was repeatedly introduced into the quantitative loop volume of 450 μL at room temperature. For PtSn/γ-Al₂O₃, according to Supplementary Fig. 11, the mole of total adsorbed CO molecules on the catalyst was calculated as 0.897 μmol (Supplementary Table 8). We assumed that one Pt atom adsorbed one CO molecule. As such, 50 mg of catalysts contained 0.897 μmol of surface Pt atoms. The TOF

number was calculated based on Eqs. (4) and (5).

$$\text{TOF} = \frac{\sum_i i \times n_i}{j \times n_{\text{Surface Pt}} \times t} \tag{4}$$

$$n_{\text{suface Pt}} = \frac{m_{catal.,exp} \times n_{CO}}{m_{catal.,CO}} \tag{5}$$

In these equations, $i$ represents to the carbon number of the product. $n_i$ is the mole of the product with the carbon number of $i$. $j$ represents the carbon number of the reactant. $t$ is the reaction time. $n_{\text{surface Pt}}$ is the moles of surface Pt atoms, which is measured according to CO pulse titration (Eq. 5). $m_{catal.\ exp}$ is the mass of the catalyst used for performance study ($m_{catal.,\ exp} = 25$ mg). $m_{catal.,CO}$ is the mass of the catalyst used for CO titration experiments ($m_{catal.,\ CO} = 50$ mg). $n_{CO}$ is the moles of adsorbed CO molecules in CO titration experiments.

For PtSn/γ-Al₂O₃, the TOF number is calculated as 41.3 $h^{-1}$. For Pt/γ-Al₂O₃, the TOF number reached 70.7 $h^{-1}$. It was worth noting that the time zero point was counted when the system temperature reached the target temperature. However, the reaction already started before the time zero point. When we calculated the TOF number, we subtracted the amount of products at the time zero point from the total yields at the reaction time.

### Catalytic tests with a trace volume of water

In a typical catalytic test, 25 mg of catalysts and different volumes of water were added into the reactor. Then, the reactor was pressurized with 6 bar of gaseous mixture (C₃H₈:N₂ = 5:1) at room temperature (25 °C). The reaction proceeded under a magnetic stirring rate of 600 rpm at 350 °C. After a 2-h reaction at 350 °C, the reactor was cooled by blowing air from the outside bottom of the reactor for facilitating the recovery of liquid-phase products in the reactor bottom. Subsequently, the bottom was washed with a quantified amount of water to recover as much liquid-phase products as possible. The gas-phase products were simultaneously analysis by gas chromatograph.

### Catalytic tests over other alkane substrates

The operation procedure for the wet reforming of n-butane (C₄H₁₀), n-pentane (C₅H₁₂), n-hexane (C₆H₁₄), n-heptane (C₇H₁₆), and n-octane (C₈H₁₈) resembles that for propane described above. Specifically, 25 mg of PtSn/γ-Al₂O₃, 2.5 mL of water, and 2.5 mL of liquid alkanes, corresponding to 21.7 mmol of C₅H₁₂, 19.1 mmol of C₆H₁₄, 17.0 mmol of C₇H₁₆, and 15.4 mmol of C₈H₁₈, were loaded in a 15-mL slurry reactor. After that, 1 bar of N₂ were charged. The test was performed at 350 °C for 2 h. For gaseous C₄H₁₀, the reactor was pressurized with 3 bar of gaseous mixture (C₄H₁₀:N₂ = 2:1, corresponding to 0.8 mmol of C₄H₁₀) and reacted at 350 °C for 2 h. The gaseous products were quantified through corresponding standard curves by gas chromatography (GC). The liquid products were quantified through corresponding standard curves by gas chromatography-mass spectrum (GC-MS). The liquid products for the wet reforming of C₄H₁₀ was determined by ¹H NMR. The mass yield for 2-butanone, 2-pentanones, 2-hexanone, 3-hexanone, 2-heptanone, 3-heptanone, 4-heptanone, 2-octanone, 3-octanone, 4-octanone are 847.2, 925.6, 873.6, 267.2, 922.4, 492.8, 109.2, 912.8, 598.4, and 337.2 μmol $g^{-1}$, respectively (Supplementary Tables 21–25). The selectivity for 2-butanone, 2-pentanones, 2-hexanone, 3-hexanone, 2-heptanone, 3-heptanone, 4-heptanone, 2-octanone, 3-octanone, 4-octanone are 63.8%, 20.2%, 17.1%, 5.2%, 14.7%, 7.9%, 1.7%, 10.8%, 7.1%, and 4.0%, respectively.

### DRIFTS measurements using CO as a probe molecule

DRIFTS experiments were conducted in a diffuse reflectance reaction chamber (Harrick Scientific) equipped with ZnSe windows, mounted inside a Praying Mantis diffuse reflectance adapter (Harrick Scientific), and coupled to a Thermo Scientific Nicolet IS50 FTIR spectrometer

with a liquid-nitrogen-cooled HgCdTe (MCT-A) detector[41]. The background spectrum of the sample was acquired after flowing under 1 bar of He with a gas-flow rate of 20 mL min$^{-1}$ at 30 °C for 1 h. Then, 1 bar of 10% CO/Ar with a rate of 15 mL min$^{-1}$ was allowed to flow into the cell for 20 min. 1 bar of He with a gas-flow rate of 20 mL min$^{-1}$ was used to purge out the gaseous CO from the sample cell so that the chemically adsorbed CO species on the samples could be detected. Based on the peak area analysis in Fig. 2f, the ratio of CO$_{linear}$ to CO$_{bridge}$ was estimated as 18.5 for PtSn/γ-Al$_2$O$_3$ and 3.0 for Pt/γ-Al$_2$O$_3$. This indicates that PtSn/γ-Al$_2$O$_3$ has fewer Pt-Pt ensembles compared to Pt/γ-Al$_2$O$_3$, likely due to the dilution effect of Sn atoms. As CO adsorption on bridge sites is a characteristic feature of Pt-Pt ensembles, these results further support the presence of PtSn alloys in PtSn/γ-Al$_2$O$_3$. Meanwhile, the presence of isolated SnO$_x$ particles as revealed by XAFS analysis indicates that the actual atomic ratio of Pt:Sn is larger than 1:1 for PtSn nanoparticles synthesized in this work, which rationalized the observation of bridge adsorbed CO.

## C$_3$H$_8$-pulse measurements

C$_3$H$_8$-pulse measurements were performed using a Micromeritics Autochem 2920 II chemisorption analyzer with an active loop volume of 525 μL. 100 mg of the sample was loaded into a U-shape quartz tube microreactor. Prior to the test, the sample was pre-treated in He with a gas-flow rate of 20 mL min$^{-1}$ at 150 °C for 1 h. Then, the temperature was ramped to 350 °C, followed by pulsing the gas with a gas-flow rate of 20 mL min$^{-1}$. We detected the mass number ($m/z$) of 29 for C$_3$H$_8$, 41 for C$_3$H$_6$, 30 for C$_2$H$_6$, 16 for CH$_4$, and 2 for H$_2$ in the outlet gas.

## C$_3$H$_6$/H$_2$O-pulse measurements

Prior to the test, the samples were heated in He with a gas-flow rate of 20 mL min$^{-1}$ at 150 °C for 1 h. Then the temperature was ramped to 350 °C. C$_3$H$_6$ was bubbled in water with a gas-flow rate of 20 mL min$^{-1}$, followed by purging into the cell. The mixed gas (C$_3$H$_6$ and H$_2$O steam) was pulsed into the He flow. Meanwhile, mass spectral responses were collected. We chose the identified mass numbers ($m/z$) with high element abundance corresponding to each species in the outlet gas. In detail, the following signals were chosen for detection, including $m/z = 29$ for C$_3$H$_8$, $m/z = 41$ for C$_3$H$_6$, $m/z = 30$ for C$_2$H$_6$, $m/z = 16$ for CH$_4$, $m/z = 2$ for H$_2$, $m/z = 18$ for H$_2$O, $m/z = 44$ for CO$_2$, $m/z = 45$ for isopropanol, and $m/z = 43$ for acetone.

## In-situ DRIFTS measurements

In-situ DRIFTS experiments were conducted in a cell with a Fourier transform infrared spectrometer and a liquid-nitrogen-cooled detector. In all, 5 mg of samples were diluted by 100 mg of KBr. The background spectrum of the sample was acquired after flowing under 1 bar of He with a gas-flow rate of 20 mL min$^{-1}$ at 350 °C for 0.5 h. Then, 1 bar of He was bubbled into isopropanol. The formed gaseous mixture containing He and isopropanol steam was purged into the in-situ cell. The time-dependent DRIFTS signals were recorded every one minute.

## Thermogravimetric measurements

TGA measurements were carried out on Pyris Diamond TG-DTA/DSC. Before measurements, 9.18 mg of PtSn/γ-Al$_2$O$_3$ and 11.06 mg Pt/γ-Al$_2$O$_3$ were dried at 100 °C in oven for 10 h, respectively. During the measurement, oxygen was purged in with a gas-flow rate of 100 mL min$^{-1}$. The whole tests were conducted under temperature-programmed control from room temperature (25 °C) to 800 °C with a heating rate of 5 °C min$^{-1}$.

## XAFS

The Pt L$_3$-edge and Sn K-edge XAFS spectra were collected at BL11B and BL14W1 beamline in Shanghai Synchrotron Radiation Facility. Some details of the XAFS tests and data analysis were as follows. The hard X-ray was monochromatized with Si(111) double-crystal

monochromator and Si(311) double-crystal monochromator for Pt L$_3$-edge and Sn K-edge measurement, respectively. The monochromator harmonics was removed by using harmonic rejection mirror. The as-prepared sample was pelletized as disc with the diameter of 10 mm. The XAFS data were collected in fluorescence mode (Lytle detector). The acquired EXAFS data were processed according to the standard procedures using the ATHENA module implemented in the IFEFFIT software packages. The EXAFS curve-fittings were performed using the ARTEMIS module of IFEFFIT[42]. In terms of Pt L$_3$-edge, the curve fitting was done on the $k^2$-weighted EXAFS function $\chi(k)$ data in the $k$-range of 3.0–12.5 Å$^{-1}$ and in the R-range of 1.0–3.2 Å. The number of independent points for these samples are $N_{ipt} = 2\Delta k \cdot \Delta R/\pi = 2 \times (12.5-3.0) \times (3.2-1.0)/\pi \approx 13$. As for Sn foil, the curve fitting was done on the $k^2$-weighted EXAFS function $\chi(k)$ data in the $k$-range of 3.0–12.0 Å$^{-1}$ and in the R-range of 2.0–3.8 Å. The number of independent points for these samples are $N_{ipt} = 2\Delta k \cdot \Delta R/\pi = 2 \times (12.0-3.0) \times (3.8-2.0)/\pi \approx 10$. As for PtSn-Al$_2$O$_3$ at Sn K-edge, the curve fitting was done on the $k^2$-weighted EXAFS function $\chi(k)$ data in the $k$-range of 2.5–10.5 Å$^{-1}$ and in the $R$-range of 1.0–2.0 Å. The number of independent points for these samples are $N_{ipt} = 2\Delta k \cdot \Delta R/\pi = 2 \times (10.5-2.5) \times (2.0-1.0)/\pi \approx 5$. Wavelet transform analysis of Pt L$_3$-edge were performed in the $k$-range of 0–12.0 Å$^{-1}$ using the Igor pro script developed by Funke et al.[43]. The Morlet wavelet was selected as basis mother wavelet and the parameters ($\eta = 8$, $\sigma = 1$) were used for a better resolution in the wave vector $k$.

## Operando XAS

Operando XAS tests were carried out using a homemade cell, which consists of reaction chamber, gas channels, cooling water channels, sample stage, Be windows and heater, as shown in Supplementary Fig. 24a–e. 30 mg of sample was compressed into a disc with 10 mm in diameter under 2 MPa for 2 min (Supplementary Fig. 24f). Then, the disc was sandwiched and locked between sample stage and the sample stage was carefully installed into the chamber. The heater and K-type thermocouple were inserted into corresponding holes for external heating and measurement of the sample temperature, respectively (Supplementary Fig. 24g, h). The control parameters can be set on the panel of console application including the target temperature, heating rate and so on (Supplementary Fig. 24i). Subsequently, 1 bar of Ar flow with the rate of 20 mL min$^{-1}$ was allowed to flow into the reaction chamber through the gas channels for 0.5 h at room temperature to purge out the air from the inside. After that, the wet propane was introduced into the cell with the rate of 20 mL min$^{-1}$ at room temperature for 0.5 h through bubbling out of water half-full filled in a 20 ml-sized bottle. Meanwhile, the circulating water was flowed through the cooling water channels embedded in the chamber wall with the rate of 8 L min$^{-1}$ by a pump to protect the gas tightness of cell enclosure and windows from heat transfer. For security, the above equipment was checked by gas alarm device carefully to prevent gas leakage.

Before operando XAS tests, the X-ray was monochromatized by a double-crystal Si(111) and Si(311) monochromator, and the energy was calibrated using a Pt foil and a Sn foil for Pt L$_3$-edge and Sn K-edge, respectively, at the BL11B beamline of the Shanghai Synchrotron Radiation Facility, China. Next, the cell was loaded and fixed on the lifting platform at precisely the right place to ensure that the incident X-ray can pass through the Be windows and sample to facilitate X-ray transmission (Supplementary Fig. 24j). With everything prepared, the cell was heated to 350 °C at a ramping rate of 20 °C min$^{-1}$ by the heater, together with the stream of wet propane, thereby providing the operando environment. Finally, the XAS spectra were acquired in transmission mode.

## Characterizations

ICP-AES (Atomscan Advantage, Thermo Jarrell Ash, USA) was used to determine the concentration of Pt and Sn species. HAADF-STEM images were collected on a JEOL ARM-200F field-emission transmission

electron microscope operating at 200 kV accelerating voltage. The energy analyzer fixed transmission energy was 30 eV. For preparing the STEM sample, the catalyst powders were dispersed in ethanol, followed by sonification to form the suspension which was subsequently dropped onto the super-thin carbon film and dried. XRD patterns were collected on a Rigaku Smartlab automated multipurpose X-ray diffractometer. For preparing the XRD sample, the catalyst powders were placed in the glass card slot, followed by flattening the surface and leveling out at sample holder plane. Raman analysis was performed using a laser Raman analyzer LabRAM HR (Horiba/Jobin Yvon, Longjumeau) with a 532 nm Ar laser.

## Data availability

All data in this work are available in the text and Supplementary Information. The relevant raw data for each figure or table (in the text and Supplementary Information) are listed in Excel documents and provided as source or supplementary data files. Source data are provided with this paper.

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

## Acknowledgements

This work was supported by National Key Research and Development Program of China (2022YFA1505300, 2021YFA1500500), CAS Project for Young Scientists in Basic Research (YSBR-051), National Science Fund for Distinguished Young Scholars (21925204), NSFC (21902149, 22221003, 22308346, 22309171, 22250007), Fundamental Research Funds for the Central Universities, Strategic Priority Research Program of the Chinese Academy of Sciences (XDB0450000), Collaborative Innovation Program of Hefei Science Center, CAS (2022HSC-CIP004), the Joint Fund of the Yulin University and the Dalian National Laboratory for Clean Energy (YLU-DNL Fund 2022012), International Partnership Program of Chinese Academy of Sciences (123GJHZ2022101GC), Joint Funds from the Hefei National Synchrotron Radiation Laboratory (KY9990000202), and USTC Research Funds of the Double First-Class Initiative (YD9990002014). J.Z. acknowledges support from the Tencent Foundation through the XPLORER PRIZE. This work was partially carried out at the Instruments Center for Physical Science, University of Science and Technology of China. This work was also partially carried out at the USTC Center for Micro and Nanoscale Research and Fabrication.

## Author contributions

X.M., H.Y., and Z.P. equally contributed to this work. X.M., Z.P., H.L., and J.Z. designed the studies and wrote the paper. X.M. and X.Z. synthesized catalysts. X.M., H.Y., W.G., and T.Z. performed catalytic tests. X.M., Z.P., S.H., L.L., and H.L. conducted mechanistic studies. All authors discussed the results and commented on the manuscript.

## Competing interests

The authors declare no competing interests.
