## [Peer Review File · Nature Communications]

Propane wet reforming over PtSn nanoparticles on γ -Al₂O₃ for acetone synthesisREVIEWER COMMENTS

Reviewer #1 (Remarks to the Author):

This paper entitled “acetone synthesis from atom-economic catalysis of propane wet oxidation” has reported a direct route to convert propane to acetone by wet oxidation. The proposed route is new, but the depth of present research is inadequate. The exploration on the reaction mechanism is not clarified well. There is a big room for improvement in the designed experiments. Also, the language should be improved. Therefore, the present manuscript is publishable, but major changes are required prior its approval.)

Please see details listed below:

1) The author stated that “we proposed a direct conversion of propane into acetone using water as a cheap abundant source of oxidants to partially oxidize propane” in multiple places over the whole manuscript. They suggested that the conversion of propane includes three elementary steps (dehydrogenation, hydration, and dehydrogenation) over two types of active sites (PtSn, Al₂O₃). However, the participation of H₂O in the 2nd step of the conversion cannot be regarded as an oxidation process. The description made by the authors could mislead the audience, since dehydration is not related to redox reaction. The author should address this issue and clarify that role of Al₂O₃ on hydration.

2) In the Results and Discussion section, more depth analysis should be made on the obtained characterization data. For example, the particle size of Pt NPs on Pt/γ-Al₂O₃ catalyst should be given in TEM analysis and compared to that of PtSn/γ-Al₂O₃ catalyst. For EXAFS analysis, more detailed analysis should be given to explain the difference between Pt/Al₂O₃ and PtSn/Al₂O₃, such as coordination number and chemical bond length. The chemical bonds and white line changes should be labeled on Fig.1. Also, there is a lack of correlation between those data.

3) The selectivity data reported in Fig.3 (a) may not be comparable under present conditions. The authors did not provide the associated propane conversion for the reported selectivity data here...The selectivity is only comparable at similar C₃H₈ conversion rate. As stated by the authors “among gaseous products, H₂ occupied the majority, with by-products of ethane, propene, methane, and CO₂” in the section of Catalytic Properties. However, they did not provide a detailed information on the percentage of liquid and gas products either in the main body of the manuscript or SI. Also, the percentage of each product that contained in the liquid or gas product should be quantitatively specified.

4) For kinetic experiment, the authors should exclude the internal and external mass transfer before any tests. At least in the present manuscript, I did not see any experiments or calculation to taking mass transfer into consideration. Figs S8 and S9 should be placed in Fig 3, and Figures 3c and 3d should move to SI. Also, what’s the kinetic dependence on H₂O? It is an important experiment; you should add this data into the paper. Can you confirm you have run the propane kinetic experiment in a zero-order H₂O dependent regime? There is a logical problem in section “with the extension of reaction, the yield.... with the prolonged reaction time”, please re-write it.

5) A few points for mechanistic studies: (a) The testing conditions should be briefly described in the legend of Figure 4. Also, the functional groups should be labeled on Fig.4 g-i and distinguished by color. (b) the conclusion stated by the authors (“Therefore, Pt atoms contribute to the dehydrogenation of propane, while the dilution with Sn atoms to avoid the formation of contiguous Pt ensembles mitigates the over-dehydrogenation of propene and its hydrogenolysis.”) is not new, several papers should be referred here. (c) from the designed experiments here, my understanding is that the dehydration occurs on Al₂O₃ to produce isopropanol, then the dehydrogenation is induced to form acetone over PtSn sites or form cracking products on Pt sites. Could you please explain the catalytic difference between PtSn and Pt in the last step of dehydrogenation? (d) the author should stress the consistency between DRIFTS and propene wet pulse experiment in their writing. (e) just wondering, how do you run the propane dehydrogenation control experiment without any solvent or water? (f) for hydration exploration, SiO₂ supported Pt or PtSn can be used to exclude the catalytic contribution from Al₂O₃ support. (g) If the breakage of D-O bonds is the rate determine step, the author should investigate the effect of H₂O in the kinetic study. (h) the author should take the reversible conversion into consideration. (i) language issue: “The acetone selectivity of γ -Al₂O₃ was only 35.8%, while the propene selectivity reached 50.4%” is not readable...do you mean “The selectivity of acetone was only 35.8% over Al₂O₃.” More efforts should be given on language. (j) I highly suggest the author to report the conversion of propane, instead of using yield.

Reviewer #2 (Remarks to the Author):

In this work, the authors claim that the direct conversion of propane to acetone can be achieved over the γ -alumina-supported PtSn clusters at 350 °C through the wet oxidation of propane using water as the oxygen source. And they proposed a reaction mechanism for the propane-wet-oxidation process on the PtSn/ γ -alumina catalyst, which couples three steps: the dehydrogenation of propane to propylene on the PtSn clusters, the hydration of propylene to isopropanol on the γ -alumina support, and the dehydrogenation of isopropanol to acetone on the PtSn clusters. However, from the available results, the efficiency of the process of wet oxidation of propylene to acetone is relatively low (It seems that the conversion of propane in the typical performance test is about 1%). Moreover, the proposed complicated reaction mechanism has not been sufficiently justified. Therefore, this manuscript is not recommended for publication in Nature Communications. Some specific comments are as follows.

- a) The authors claim that the propane-wet-oxidation process overcomes the equilibrium limitation of non-oxidative propane dehydrogenation. To verify this, the authors are advised to calculate the thermodynamic equilibrium conversion of propane under reaction conditions.
- b) Data on propane conversion is missing throughout the manuscript. What is the conversion of propane in the relevant tests?
- c) The reaction order of propane determined by linear fitting of lnTOF and lnP(C₃H₈) is 1.14, which is higher than 1. Why?
- d) In terms of the reaction mechanism, the authors directly decouple the propane-wet-oxidation process into a two-step reaction between the dehydrogenation of propane to propylene and the

subsequent hydration of propylene to acetone. Are there other potential pathways for the formation of acetone? Is it possible that oxygen-containing functional groups are directly involved in the dehydrogenation of C3 hydrocarbon intermediates?

e) Kinetic isotope effects are very helpful in understanding reaction mechanisms. The authors measured the KIE for propylene hydration and deduced that the rate-determining step in this reaction involves the cleavage of the O-H bond. What is the KIE for wet-oxidation of propane with replacing H₂O by D₂O?

f) Line 178 : ”, followed by pulsing wet propane through bubbling into the reactor.” Perhaps the author here is talking about propylene instead of propane. Moreover, how about the results for pulsing wet propane? Furthermore, the typical performance tests were conducted in water-rich environments. The H₂O/C₃H₈ molar ratios under reaction conditions were much higher than those during pulse testing. I am afraid that the results of the pulse test do not reflect the catalytic process under reaction conditions.

Reviewer #3 (Remarks to the Author):

Comments

This manuscript presents the results of a PtSn/alumina catalyst for the oxidation of propane to acetone. While these preliminary results are interesting, the quality of the manuscript and the presentation of the data together with the lack of details makes this unacceptable in its present form. The authors must consider the following:

There are no error bars included in any of the data, from the catalytic activity to the EXAFS modeling. The authors should refer to the “Addressing Rigor and Reproducibility in Thermal, Heterogeneous Catalysis,” report (<https://doi.org/10.5281/zenodo.8029159>) as a guide to how to present data in an acceptable format.

The EXAFS data are unacceptable in the current form. How were the data collected? What was the form of the sample? The fits must be shown. Was fitting attempted without a Pt-Sn scattering path? Where is the Sn K-edge data? Are the scales of the wavelet transforms (Fig. S4) the same? Where are the raw $\chi(k)$ data? What fitting range was used? Etc. etc.

The elemental stoichiometry of the sample should be provided in addition to the wt%. Why was this value chosen?

How does “absence of peaks for PtSn nanocrystals” imply that the particles were amorphous?

The authors must describe in more detail regarding the sample preparation. What does “quickly pushed into the quartz tube under a H₂ flow” even mean and how was this accomplished, and why did it have to be done quickly?

What form of the catalyst was analyzed by XRD, STEM and XAS? This is never described.

I strongly disagree with the statement “Elemental analysis of PtSn/ γ -Al₂O₃ indicated the homogeneous distribution of Pt and Sn without obvious phase segregation (Fig. 2c).” First, what does this sentence mean? I assume it means that there is a uniform distribution of Pt and Sn and that the particles are thus PtSn bimetallic. The EDS map does not support this conclusion. It appears to show Sn on the alumina support with some Pt close to the Sn. Indeed this would be expected from the method of sample preparation.

There are many aspects of this work that are well known and have been reported for decades. For example it is well known that the addition of Sn to Pt “restrains the cracking of C-C bonds”. The TOF values are extremely small, ca 2 per hour. This raises the question of whether what they are measuring is indeed catalytic.

Why was the reaction carried out at pressure?

Nowhere in the manuscript do the authors document the conversion of propane to acetone.

Point-by-point response to reviewers' comments

Manuscript ID: NCOMMS-23-30241

Title: Acetone synthesis from atom-economic catalysis of propane wet oxidation

Reviewer #1

“This paper entitled ‘acetone synthesis from atom-economic catalysis of propane wet oxidation’ has reported a direct route to convert propane to acetone by wet oxidation. The proposed route is new, but the depth of present research is inadequate. The exploration on the reaction mechanism is not clarified well. There is a big room for improvement in the designed experiments. Also, the language should be improved. Therefore, the present manuscript is publishable, but major changes are required prior its approval.) Please see details listed below:”

We sincerely thank this reviewer's valuable comments that help us to improve the quality of our manuscript. As suggested, we have added a lot of relevant experiments to address the concerns raised by this reviewer as follows.

“1) The author stated that “we proposed a direct conversion of propane into acetone using water as a cheap abundant source of oxidants to partially oxidize propane” in multiple places over the whole manuscript. They suggested that the conversion of propane includes three elementary steps (dehydrogenation, hydration, and dehydrogenation) over two types of active sites (PtSn, Al₂O₃). However, the participation of H₂O in the 2nd step of the conversion cannot be regarded as an oxidation process. The description made by the authors could mislead the audience, since dehydration is not related to redox reaction. The author should address this issue and clarify that role of Al₂O₃ on hydration.”

Thanks for pointing out this issue. We agree with this reviewer's opinion that the participation of water in the hydration process cannot be regarded as an oxidation process. The oxidation actually occurs during the first dehydrogenation process ($C_3H_8 = C_3H_6 + H_2$). As such, it is not proper to use the statement of “wet oxidation” or regard water as the oxidant. In the revised manuscript, we have replaced this statement with “wet reforming” and rewritten the relevant sentences. These changes in the revised manuscript have been highlighted in yellow color.

As for the role of γ -Al₂O₃, we regard it responsible for the second hydration process. In the original manuscript, we had directly used γ -Al₂O₃ as the catalyst which enabled highly selective hydration of propene into isopropanol. To further confirm this role, we have replaced γ -Al₂O₃ with SiO₂ to support PtSn and Pt nanoparticles, denoted as PtSn/SiO₂ and Pt/SiO₂, respectively, for the hydration reaction. Specifically, 25 mg of PtSn/SiO₂ or Pt/SiO₂ was operated in 5 mL of water under 6 bar (C₃H₆:N₂ = 5:1) at 350 °C for 2 h. Both the SiO₂-supported catalysts induced the reverse hydrogenation of propene into propane instead of the formation of isopropanol or acetone (Fig. R1). This result further implies that γ -Al₂O₃ plays the critical role in the hydration process. We have added relevant discussion in the revised manuscript (p. 11, lines 20-24, Supplementary Fig. 24, highlighted in yellow color).

Figure R1. Hydration of propene over PtSn/SiO₂ and Pt/SiO₂. Typically, 25 mg of the catalyst was operated in 5 mL of water under 6 bar (C₃H₆:N₂ = 5:1) at 350 °C for 2 h.

“2) In the Results and Discussion section, more depth analysis should be made on the obtained characterization data. For example, the particle size of Pt NPs on Pt/γ-Al₂O₃ catalyst should be given in TEM analysis and compared to that of PtSn/γ-Al₂O₃ catalyst. For EXAFS analysis, more detailed analysis should be given to explain the difference between Pt/Al₂O₃ and PtSn/Al₂O₃, such as coordination number and chemical bond length. The chemical bonds and white line changes should be labeled on Fig.1. Also, there is a lack of correlation between those data.”

We genuinely thank this reviewer for his/her constructive suggestions. As requested, we have added the size distribution of metal nanoparticles in PtSn/γ-Al₂O₃ and Pt/γ-Al₂O₃ based on the TEM analysis. The average size of metal nanoparticles in PtSn/γ-Al₂O₃ was 1.74 nm, which was slightly higher than that (1.31 nm) in Pt/γ-Al₂O₃ (Fig. R2). We have added relevant discussion in the revised manuscript (p. 5, lines 20 and 26-28, Supplementary Fig. 2, highlighted in yellow color).

Figure R2. Size distribution of metal nanoparticles in (a) PtSn/γ-Al₂O₃ and (b) Pt/γ-Al₂O₃.

To explain the variation of particle sizes, we further analyzed the EXAFS data as shown in Table R1. The bond lengths of Pt-Sn (2.70 Å) and Pt-Pt (2.74 Å) in PtSn/γ-Al₂O₃ were close to that of Pt-Pt (2.73 Å) in Pt/γ-Al₂O₃. The coordination number (4.3) of Pt-Pt in PtSn/γ-Al₂O₃ was slightly

lower than that (4.8) in Pt/ γ -Al₂O₃. As such, we speculate that the enlarged size was not due to the lattice expansion induced by the introduction of Sn but owing to the aggregation of more atoms (such as Sn atoms) in an individual particle. Moreover, the coordination number (0.3) of Pt-O in PtSn/ γ -Al₂O₃ was obviously lower than that (1.7) in Pt/ γ -Al₂O₃. We have supplemented the Sn K-edge analysis of PtSn/ γ -Al₂O₃ in the revised manuscript. From the result, we found that Sn-O bonds occupied the majority in the absence of Pt-Sn bonds. We can deduce that not all the Pt and Sn atoms were uniformly mixed to form PtSn bimetallic nanoparticles. Instead, a large proportion of Sn atoms were oxidized to form SnO_x nanoparticles, leaving some Sn atoms coordinated with Pt atoms. In PtSn nanoparticles, Sn atoms were more prone to be oxidized than Pt atoms. Considering the inertness of SnO_x towards the reaction of propane with water, we can still denote the star sample as PtSn/ γ -Al₂O₃ for simplicity. As requested by this reviewer, we have labeled the chemical bonds and white line changes on Figure 1. We have added relevant discussion in the revised manuscript (p. 5, lines 9-18, Supplementary Fig. 8, Supplementary Table 3, highlighted in yellow color).

Table R1 | EXAFS fitting parameters at the Pt L₃-edge and Sn K-edge for various samples. For Pt L₃-edge analysis, S_0^2 was set to 0.776, according to the experimental EXAFS fit of Pt foil reference by fixing coordination numbers as the known crystallographic value. For Sn K-edge analysis, S_0^2 was set to 0.90, according to the experimental EXAFS fit of Sn foil reference by fixing coordination numbers as the known crystallographic value.

Absorption edge	Sample	Shell	$N^{[a]}$	$R(\text{\AA})^{[b]}$	$\sigma^2(\text{\AA}^2)^{[c]}$	$\Delta E_0(\text{eV})^{[d]}$	$R\text{ factor}^{[e]}$
Pt L ₃ -edge	Pt/ γ -Al ₂ O ₃	Pt-O	1.7	2.00	0.0107	6.0	0.0056
		Pt-Pt	4.8	2.73	0.0091		
	PtSn/ γ -Al ₂ O ₃	Pt-O	0.3	1.95	0.0155	3.9	0.0020
		Pt-Sn	1.6	2.70	0.0064		
		Pt-Pt	4.3	2.74	0.0013		
Sn K-edge		Sn-O	5.0	2.01	0.0045	-0.3	0.0048

[a] N : coordination numbers.

[b] R : bond distance.

[c] σ^2 : Debye-Waller factors.

[d] ΔE_0 : the inner potential correction.

[e] R factor: goodness of fit.

“3) The selectivity data reported in Fig.3 (a) may not be comparable under present conditions. The authors did not provide the associated propane conversion for the reported selectivity data here...The selectivity is only comparable at similar C₃H₈ conversion rate. As stated by the authors “among gaseous products, H₂ occupied the majority, with by-products of ethane, propene, methane, and CO₂” in the section of Catalytic Properties. However, they did not provide a detailed information on the percentage of liquid and gas products either in the main body of the manuscript or SI. Also, the percentage of each product that contained in the liquid or gas product should be quantitatively specified.”

We appreciate the reviewer’s valuable comments. As suggested by this reviewer, we have calculated the propane conversion for all the catalytic data based on the following equation.

$$\text{Conversion} = \frac{n(\text{CO}_2) + n(\text{CH}_4) + 2n(\text{C}_2\text{H}_6) + 3n(\text{C}_3\text{H}_6) + 3n(\text{C}_3\text{H}_8\text{O}) + 3n(\text{C}_3\text{H}_6\text{O})}{3n(\text{C}_3\text{H}_8)_{\text{in}}} \times 100\% \quad (1)$$

We agree with this reviewer that it is important to compare the selectivity at similar propane conversions. When both catalysts were evaluated under the same condition (25 mg of catalysts, 5 mL of water, 6 bar ($\text{C}_3\text{H}_8:\text{N}_2 = 5:1$), 350 °C, 2 h), the propane conversion (6.53%) over Pt/ $\gamma\text{-Al}_2\text{O}_3$ was higher than that (1.84%) over PtSn/ $\gamma\text{-Al}_2\text{O}_3$. In order to achieve similar conversions, we have shortened the reaction time to 10 min over Pt/ $\gamma\text{-Al}_2\text{O}_3$ during which the propane conversion reached 1.25%. Under this condition over Pt/ $\gamma\text{-Al}_2\text{O}_3$, the major product was ethane with the selectivity of 68.67%, whereas the selectivities of CO_2 , methane, and acetone were 22.36%, 8.03%, and 0.94%, respectively (Fig. R3). As such, the products were either cracked or over-oxidized over Pt/ $\gamma\text{-Al}_2\text{O}_3$. We have added relevant discussion in the revised manuscript (p. 6, lines 20-27, Supplementary Fig. 11, highlighted in yellow color).

Figure R3. Comparison of product distribution at similar conversions over Pt/ $\gamma\text{-Al}_2\text{O}_3$ and PtSn/ $\gamma\text{-Al}_2\text{O}_3$.

As reminded by this reviewer, we have added the molar percentage of liquid and gas products (Table R2) as well as the molar percentage of each product contained in the liquid or gas product (Tables R3 and R4) in the revised manuscript (Supplementary Tables 5-7, highlighted in yellow color).

Table R2. Molar percentage of liquid and gaseous products obtained after a standard operation. Typically, 25 mg of PtSn/ $\gamma\text{-Al}_2\text{O}_3$ and 5 mL of water were loaded in a 15-mL slurry reactor to operate under 6 bar ($\text{C}_3\text{H}_8:\text{N}_2 = 5:1$) at 350 °C for 2 h.

Products	Gaseous products	Liquid products
Yielded ($\mu\text{mol g}^{-1}$)	3010.2	864.9

Molar percentage (mol%)	77.7	22.3
------	------

Table R3. Molar percentage of each product contained in the gaseous product obtained after a standard operation. Typically, 25 mg of PtSn/ γ -Al₂O₃ and 5 mL of water were loaded in a 15-mL slurry reactor to operate under 6 bar (C₃H₈:N₂ = 5:1) at 350 °C for 2 h.

Gaseous products	CO ₂	CH ₄	C ₂ H ₆	C ₃ H ₆	H ₂
Molar percentage (mol%)	10.2	8.3	14.0	5.1	62.4

Table R4. Molar percentage of each product contained in the liquid product obtained after a standard operation. Typically, 25 mg of PtSn/ γ -Al₂O₃ and 5 mL of water were loaded in a 15-mL slurry reactor to operate under 6 bar (C₃H₈:N₂ = 5:1) at 350 °C for 2 h.

Liquid products	C ₃ H ₆ O	C ₃ H ₈ O
Molar percentage (mol%)	99.3	0.7

“4) For kinetic experiment, the authors should exclude the internal and external mass transfer before any tests. At least in the present manuscript, I did not see any experiments or calculation to taking mass transfer into consideration.”

As suggested by this reviewer, we have explored the influence of mass transfer over the star sample PtSn/ γ -Al₂O₃. To explore the effect of external diffusion, we have varied the stirring speeds (200, 400, and 600 rpm) to conduct the wet reforming of propane. Despite of the varied stirring rates, the conversion and product distribution still kept almost unchanged (Fig. R4). As such, the reaction kinetics was not determined by external diffusion.

Figure R4. Dependence of products and conversion on the stirring rate. Typically, 25 mg of PtSn/ γ -Al₂O₃ (>60 mesh) and 5 mL of water were loaded in a 15-mL slurry reactor with different stirring speeds to operate under 6 bar (C₃H₈:N₂ = 5:1) at 350 °C for 2 h.

To investigate the influence of internal diffusion, we have granulated the catalyst into 20-40 mesh, 40-60 mesh, and >60 mesh. As shown in Figure R5, the conversion and selectivity were not sensitive to the mesh, indicating the negligible influence of internal diffusion.

Figure R5. Dependence of products and conversion on the mesh. Typically, 25 mg of PtSn/ γ -Al₂O₃ in different meshes and 5 mL of water were loaded in a 15-mL slurry reactor with the stirring speed of 600 rpm to operate under 6 bar (C₃H₈:N₂ = 5:1) at 350 °C for 2 h.

To clarify the reaction kinetics, we have added the relevant discussion in the revised manuscript (p. 7, lines 9-16, Supplementary Figs. 14 and 15, highlighted in yellow color).

“Figs S8 and S9 should be placed in Fig 3, and Figures 3c and 3d should move to SI.”

As suggested, we have placed original Supplementary Figures S8 and S9 in Figure 3, and moved original Figures 3c and 3d to the supplementary information (Fig. 3c,d, Supplementary Figs. 16 and 17, highlighted in yellow color).

“Also, what’s the kinetic dependence on H₂O? It is an important experiment; you should add this data into the paper. Can you confirm you have run the propane kinetic experiment in a zero-order H₂O dependent regime?”

Thanks for raising this issue. During the standard condition (25 mg of PtSn/ γ -Al₂O₃, 5 mL of water, 6 bar (C₃H₈:N₂ = 5:1), 350 °C, 2 h), we added 278 mmol of water and 2 mmol of propane. As such, the amount of added water was far excessive. To investigate the kinetic dependence on water, we have changed the volume of water to 4.95 mL, 4.90 mL, and 4.80 mL. With the decrease in the volume of added water, the propane conversion slightly increased mainly due to the increased yield of ethane (Fig. R6). This phenomenon was mainly ascribed to the increased amount of propane molecules considering that the total volume of the reactor was only 15 mL and the total pressure (C₃H₈:N₂ = 5:1) was kept the same under 6 bar. No wonder this trend of

decreasing the volume of water was similar to that obtained from the experiments of elevating the propane partial pressure (Supplementary Fig. 17). Overall, we can basically confirm that the propane kinetic experiment was operated in a near zero-order H₂O dependent regime. We have added relevant discussion in the revised manuscript (p. 7, lines 30-31, p. 8, lines 1-6, Supplementary Fig. 18, highlighted in yellow color).

Figure R6. Dependence of products and conversion on the volume of added water. Typically, 25 mg of PtSn/ γ -Al₂O₃ (>60 mesh) and different volumes of water were loaded in a 15-mL slurry reactor with the stirring speed of 600 rpm to operate under 6 bar (C₃H₈:N₂ = 5:1) at 350 °C for 2 h.

“There is a logical problem in section “with the extension of reaction, the yield.... with the prolonged reaction time”, please re-write it.”

Thanks for reminding. We agree with this reviewer that the rapid consumption of propene was unable to be deduced from the unchanged yield of propene with the extension of reaction. In the revised manuscript, we have also deleted the deduction that the dehydrogenation of propane into propene was the rate-limiting step. Moreover, we have further analyzed the kinetic isotope effect (KIE) of the reaction of propane with D₂O over PtSn/ γ -Al₂O₃. As shown in Figure R7, the KIE value ($k_{\text{H}_2\text{O}}/k_{\text{D}_2\text{O}}$) was 4.46, indicating that the breakage or the formation of chemical bonds related to D atoms participated in the rate-limiting step. However, the dehydrogenation of propane into propene did not involve the D atoms from water, so the first dehydrogenation step cannot be the rate-limiting step. We have added relevant discussion in the revised manuscript (p. 8, lines 6-10, Supplementary Fig. 19, highlighted in yellow color).

Figure R7. KIE test of propane wet reforming over PtSn/ γ -Al₂O₃.

“5) A few points for mechanistic studies:

(a) The testing conditions should be briefly described in the legend of Figure 4. Also, the functional groups should be labeled on Fig.4 g-i and distinguished by color.”

Thanks for the helpful suggestions. As requested by this reviewer, we have briefly described the testing condition in the legend of Figure 4. Moreover, we have also labeled the functional groups on Figure 4g-i.

“(b) the conclusion stated by the authors (“Therefore, Pt atoms contribute to the dehydrogenation of propane, while the dilution with Sn atoms to avoid the formation of contiguous Pt ensembles mitigates the over-dehydrogenation of propene and its hydrogenolysis.”) is not new, several papers should be referred here.”

Our choice of PtSn nanoparticles was just originated from the relatively mature catalyst for propane dehydrogenation in the UOP Oleflex process. As requested, we have cited several relevant papers in the revised manuscript (refs. 30-32, highlighted in yellow color).

“(c) from the designed experiments here, my understanding is that the dehydration occurs on Al₂O₃ to produce isopropanol, then the dehydrogenation is induced to form acetone over PtSn sites or form cracking products on Pt sites. Could you please explain the catalytic difference between PtSn and Pt in the last step of dehydrogenation?”

For Pt/ γ -Al₂O₃, most of the products were cracked species including ethane and methane, whereas PtSn/ γ -Al₂O₃ achieved a high acetone selectivity of 85.8%. This difference was because contiguous Pt ensembles enabled the cracking of C-C bonds. The dilution of Pt atoms with Sn atoms could significantly mitigate the cracking process. This explanation has been widely adopted for the propane dehydrogenation as raised by this reviewer in the previous question. We have also referred related papers here (p. 11, lines 5-7, refs. 30-32, highlighted in yellow color).

“(d) the author should stress the consistency between DRIFTS and propene wet pulse experiment in their writing.”

Thanks for the reminding. We have stressed the consistency between DRIFTS and propene wet pulse experiment in the revised manuscript (p. 9, lines 29-30, highlighted in yellow color).

“(e) just wondering, how do you run the propane dehydrogenation control experiment without any solvent or water?”

We just put the catalyst powders into the slurry reactor without adding any solvent or water. It should be noted that the temperature probe should be near the bottom of the reactor. The inflation process should be cautious and slow in case that the catalyst powders are purged out of the reactor.

“(f) for hydration exploration, SiO₂ supported Pt or PtSn can be used to exclude the catalytic contribution from Al₂O₃ support.”

Thanks for the helpful suggestion. As suggested, we have replaced γ -Al₂O₃ with SiO₂ to support PtSn and Pt nanoparticles, denoted as PtSn/SiO₂ and Pt/SiO₂, respectively, for the hydration reaction. Specifically, 25 mg of PtSn/SiO₂ or Pt/SiO₂ was operated in 5 mL of water under 6 bar (C₃H₆:N₂ = 5:1) at 350 °C for 2 h. Both the SiO₂-supported catalysts induced the reverse hydrogenation of propene into propane instead of the formation of isopropanol or acetone (Fig. R1). This result confirms the catalytic contribution of γ -Al₂O₃ in the hydration process. We have added relevant discussion in the revised manuscript (p. 11, lines 20-24, Supplementary Fig. 24, highlighted in yellow color).

“(g) If the breakage of D-O bonds is the rate determine step, the author should investigate the effect of H₂O in the kinetic study.”

In the original KIE tests towards the hydration of propene, we had assumed the breakage of D-O bonds in the rate-limiting step over γ -Al₂O₃ because isopropanol was the dominant product. As for PtSn/ γ -Al₂O₃, we observed the formation of other products without isopropanol. In other words, the breakage of D-O bonds in the rate-limiting step cannot be simply applied to the case of PtSn/ γ -Al₂O₃. To explore the rate-limiting step over PtSn/ γ -Al₂O₃, we have further analyzed the KIE of the reaction of propane with D₂O. As shown in Figure R7, the KIE value ($k_{\text{H}_2\text{O}}/k_{\text{D}_2\text{O}}$) was 4.46, indicating that the breakage or the formation of chemical bonds related to D atoms participated in the rate-limiting step.

Actually, the amount (278 mmol) of added water was far excessive relative to the amount (2 mmol) of propane. To investigate the kinetic dependence on water, we have changed the volume of water to 4.95 mL, 4.90 mL, and 4.80 mL over PtSn/ γ -Al₂O₃. With the decrease in the volume of added water, the propane conversion slightly increased mainly due to the increased yield of ethane (Fig. R6). This phenomenon was mainly ascribed to the increased amount of propane molecules considering that the total volume of the reactor was only 15 mL and the total pressure (C₃H₈:N₂ = 5:1) was kept the same under 6 bar. No wonder this trend of decreasing the volume of water was similar to that obtained from the experiments of elevating the propane partial pressure (Supplementary Fig. 17). Overall, we can basically confirm that the propane kinetic experiment was operated in a near zero-order H₂O dependent regime.

“(h) the author should take the reversible conversion into consideration.”

Thanks for reminding! We have taken the reversible conversion into consideration. The catalytic data just showed the net outcome. When we conducted the experiments of propene hydration and isopropanol dehydrogenation, we observed the reverse hydrogenation featured by the formation of alkane products. We have added the discussion in the revised manuscript.

“(i) language issue: “The acetone selectivity of γ -Al₂O₃ was only 35.8%, while the propene selectivity reached 50.4%” is not readable...do you mean “The selectivity of acetone was only 35.8% over Al₂O₃.” More efforts should be given on language.”

As raised by this reviewer, we just mean that “The selectivity of acetone was only 35.8% over Al₂O₃”. We have carefully checked the grammar and improved the language in the revised manuscript.

“(j) I highly suggest the author to report the conversion of propane, instead of using yield.”

As suggested, we have reported the conversion of propane in the revised manuscript (Fig. 3d, Supplementary Figs. 11, 13-18, highlighted in yellow color). In this work, we claim that the propane-wet-oxidation process overcomes the equilibrium limitation of non-oxidative propane dehydrogenation. To this end, we have calculated the thermodynamic equilibrium conversion of propane during non-oxidative propane dehydrogenation and compared it with the propane conversion during wet reforming (our work). As shown in Table R5, the propane conversion over PtSn/ γ -Al₂O₃ during wet reforming exceeded the thermodynamic limit. Based on the Le Chatelier’s principle, lowering the pressure will increase the conversion of propane for both reactions (eqs. 2 and 3). When the partial pressure of propane was lowered during wet reforming, the propane conversion indeed increased and reached 10.85% under 0.2 bar of C₃H₈ and 0.8 bar of N₂ at 350 °C. We have added relevant discussion in the revised manuscript (p. 6, lines 13-19, Supplementary Table 8, highlighted in yellow color).

Table R5. Comparison between the propane conversion during wet reforming and the thermodynamic equilibrium conversion of propane during non-oxidative propane dehydrogenation. Equilibrium calculations were performed through HSC Chemistry 6 software, which utilizes a Gibbs free energy minimization algorithm.

Reaction conditions	Propane conversion	Thermodynamic equilibrium conversion of propane
5 bar of C ₃ H ₈ and 1 bar of N ₂ , 350 °C	1.84% (2 h)	0.8%
	2.33% (4 h)	
	2.81% (6 h)	
3 bar of C ₃ H ₈ and 1 bar of N ₂ , 350 °C	2.69% (2 h)	1.0%

0.2 bar of C ₃ H ₈ and 0.8 bar of N ₂ , 350 °C	25.60% (2 h)	3.5%
---	--------------	------

Reviewer #2

“In this work, the authors claim that the direct conversion of propane to acetone can be achieved over the γ -alumina-supported PtSn clusters at 350 °C through the wet oxidation of propane using water as the oxygen source. And they proposed a reaction mechanism for the propane-wet-oxidation process on the PtSn/ γ -alumina catalyst, which couples three steps: the dehydrogenation of propane to propylene on the PtSn clusters, the hydration of propylene to isopropanol on the γ -alumina support, and the dehydrogenation of isopropanol to acetone on the PtSn clusters. However, from the available results, the efficiency of the process of wet oxidation of propylene to acetone is relatively low (It seems that the conversion of propane in the typical performance test is about 1%). Moreover, the proposed complicated reaction mechanism has not been sufficiently justified. Therefore, this manuscript is not recommended for publication in Nature Communications. Some specific comments are as follows.”

We genuinely appreciate this reviewer’s valuable comments that help us to improve the quality of our manuscript. Especially for the conversion issue, we have calculated the propane conversion based on the following equation (eq. 1) and compared it with the thermodynamic limits. We found that the propane conversion over PtSn/ γ -Al₂O₃ during wet reforming exceeded the thermodynamic equilibrium conversion of propane during non-oxidative propane dehydrogenation. Detailed analysis was specified in the following response.

$$\text{Conversion} = \frac{n(\text{CO}_2) + n(\text{CH}_4) + 2n(\text{C}_2\text{H}_6) + 3n(\text{C}_3\text{H}_6) + 3n(\text{C}_3\text{H}_8\text{O}) + 3n(\text{C}_3\text{H}_6\text{O})}{3n(\text{C}_3\text{H}_8)_{\text{in}}} \times 100\% \quad (1)$$

“a) The authors claim that the propane-wet-oxidation process overcomes the equilibrium limitation of non-oxidative propane dehydrogenation. To verify this, the authors are advised to calculate the thermodynamic equilibrium conversion of propane under reaction conditions.”

We sincerely thank this reviewer for raising this constructive suggestion. As suggested, we have calculated the thermodynamic equilibrium conversion of propane during non-oxidative propane dehydrogenation and compared it with the propane conversion during wet reforming (our work). As shown in Table R1, the propane conversion over PtSn/ γ -Al₂O₃ during wet reforming exceeded the thermodynamic limit. Based on the Le Chatelier’s principle, lowering the pressure will increase the conversion of propane for both reactions (eqs. 2 and 3). When the partial pressure of propane was lowered during wet reforming, the propane conversion indeed increased and reached 25.60% under 0.2 bar of C₃H₈ and 0.8 bar of N₂ at 350 °C. We have added relevant discussion in the revised manuscript (p. 6, lines 13-19, Supplementary Table 8, highlighted in yellow color).

Table R1. Comparison between the propane conversion during wet reforming and the thermodynamic equilibrium conversion of propane during non-oxidative propane dehydrogenation. Equilibrium calculations were performed through HSC Chemistry 6 software, which utilizes a Gibbs free energy minimization algorithm.

Reaction conditions	Propane conversion	Thermodynamic equilibrium conversion of propane
5 bar of C ₃ H ₈ and 1 bar of N ₂ , 350 °C	1.84% (2 h)	0.8%
	2.33% (4 h)	
	2.81% (6 h)	
3 bar of C ₃ H ₈ and 1 bar of N ₂ , 350 °C	2.69% (2 h)	1.0%
0.2 bar of C ₃ H ₈ and 0.8 bar of N ₂ , 350 °C	25.60% (2 h)	3.5%

“b) Data on propane conversion is missing throughout the manuscript. What is the conversion of propane in the relevant tests?”

As requested, we have added the propane conversion in the revised manuscript (Fig. 3d, Supplementary Figs. 11, 13-18, highlighted in yellow color).

“c) The reaction order of propane determined by linear fitting of lnTOF and lnP(C₃H₈) is 1.14, which is higher than 1. Why?”

Thanks for raising this issue. Since we have conducted three independent measurements to add the error bars as requested by another reviewer, the reaction order with respect to propane slightly changed to 1.12 (Fig. 3c). Anyway, this value was slightly greater than one. This phenomenon possibly derives from that the reaction of propane with water is a complex reaction. For the main product, acetone, there exists three tandem steps including the dehydrogenation of propane into propene, the hydration of propene into isopropanol, and the dehydrogenation of isopropanol into acetone. Moreover, we also observed the over-cracking products such as methane and ethane as well as the over-oxidized products such as CO₂. The complex reaction routes led to the non-unit reaction order. We have added relevant discussion in the revised manuscript (p. 7, lines 27-29, Supplementary Table 8, highlighted in yellow color).

“d) In terms of the reaction mechanism, the authors directly decouple the propane-wet-oxidation process into a two-step reaction between the dehydrogenation of propane to propylene and the subsequent hydration of propylene to acetone. Are there other potential pathways for the formation of acetone? Is it possible that oxygen-containing functional groups are directly involved in the dehydrogenation of C₃ hydrocarbon intermediates?”

This is an interesting question. Considering that propane is a saturated alkane molecule, the oxygen-containing functional groups such as hydroxyl groups must firstly remove one hydrogen atom from the propane instead of adding oxygen-related groups. If the hydrogen atom on the end C atom is removed to form CH₃CH₂CH₂*, the whole reaction is likely to generate normal C₃ oxygenates such as n-propanol or propyl aldehyde. However, only isomerized C₃ oxygenates

were obtained in our work, consistent with the Markovnikov rule where the nucleophilic group such as the hydroxyl group adds to the least hydrogenated carbon. If the hydrogen atom on the middle C atom is removed to form $^*\text{CH}(\text{CH}_3)_2$, the hydroxyl group can directly add to the middle C atom, resulting in the formation of isopropanol. This route might exist. If the reaction follows this route, propene will not be generated. Anyway, we cannot exclude the possibility of this route though we insist on our originally proposed mechanism considering the existence of propene in the products.

“e) Kinetic isotope effects are very helpful in understanding reaction mechanisms. The authors measured the KIE for propylene hydration and deduced that the rate-determining step in this reaction involves the cleavage of the O-H bond. What is the KIE for wet-oxidation of propane with replacing H₂O by D₂O?”

As suggested by this reviewer, we have further analyzed the kinetic isotope effect (KIE) of the reaction of propane with D₂O over PtSn/ γ -Al₂O₃. As shown in Figure R1, the KIE value ($k_{\text{H}_2\text{O}}/k_{\text{D}_2\text{O}}$) was 4.46, indicating that the breakage or the formation of chemical bonds related to D atoms participated in the rate-limiting step. We have added relevant discussion in the revised manuscript (p. 8, lines 6-10, Supplementary Fig. 19, highlighted in yellow color).

Figure R1. KIE test of propane wet reforming over PtSn/ γ -Al₂O₃.

“f) Line 178: “, followed by pulsing wet propane through bubbling into the reactor.” Perhaps the author here is talking about propylene instead of propane. Moreover, how about the results for pulsing wet propane? Furthermore, the typical performance tests were conducted in water-rich environments. The H₂O/C₃H₈ molar ratios under reaction conditions were much higher than those during pulse testing. I am afraid that the results of the pulse test do not reflect the catalytic process under reaction conditions.”

Thanks for pointing out this mistake. As raised by this reviewer, this sentence should be “, followed by pulsing wet propene through bubbling into the reactor.” As suggested by this

reviewer, we have added the experiments of pulsing wet propane over PtSn/ γ -Al₂O₃. As shown in Figure R2, we observed the signals at $m/z = 29$, 43, and 45. For assignment, the MS profile of pure propane also exhibited the signals at these positions. Considering that the signal at $m/z = 43$ from acetone was likely overshadowed by that from propane, we turned to the signal at $m/z = 58$ which was the secondly strongest peak for a typical MS profile of acetone. However, we did not observe this signal during the experiment of pulsing wet propane. In addition, we have used propane pulses with D₂O bubble into He flow. The signal at $m/z = 46$ appeared, implying the formation of isopropanol. Therefore, pulsing wet propane over PtSn/ γ -Al₂O₃ only enabled the generation of isopropanol. We have added relevant discussion in the revised manuscript (p. 8, line 31, p. 9, line 1, Supplementary Fig. 10, highlighted in yellow color).

Figure R2. (a) Transient response curves of pure propane pulses for the signals at $m/z = 29$, 43, and 45. (b) Transient response curves of PtSn/ γ -Al₂O₃ obtained during wet propane pulses into He flow at 350 °C with the rate of 20 mL min⁻¹ for the signals at $m/z = 29$, 43, and 45. (c) Comparison of pure propane pulses and H₂O/C₃H₈ pulses over PtSn/ γ -Al₂O₃ for the signal at $m/z = 58$. (d) Comparison of pure propane pulses and D₂O/C₃H₈ pulses over PtSn/ γ -Al₂O₃ for the signal at $m/z = 46$.

We agree with this reviewer that the H₂O/C₃H₈ molar ratios under reaction conditions were much higher than those during pulse testing. There indeed existed the inconsistency where pulsing wet propane led to the formation of negligible acetone due to the pressure gap. Anyway, we used the pulse tests for the qualitative analysis of the intermediates. Moreover, we combined multiple approaches such as *in-situ* DRIFTS characterizations and controlled catalytic experiments to deduce the catalytic mechanism in addition to the pulse testing.

Reviewer #3

“This manuscript presents the results of a PtSn/alumina catalyst for the oxidation of propane to acetone. While these preliminary results are interesting, the quality of the manuscript and the presentation of the data together with the lack of details makes this unacceptable in its present form. The authors must consider the following:”

We sincerely thank this reviewer for his/her valuable comments that help us improve the quality of our manuscript.

“There are no error bars included in any of the data, from the catalytic activity to the EXAFS modeling. The authors should refer to the “Addressing Rigor and Reproducibility in Thermal, Heterogeneous Catalysis,” report (<https://doi.org/10.5281/zenodo.8029159>) as a guide to how to present data in an acceptable format.”

This reviewer recommended us a very helpful report. As requested, we have added the error bars in catalytic data and EXAFS modeling in the revised manuscript (Figs. 3 and 5a, Supplementary Figs. 11, 13-18, 24 and 25, Supplementary Tables 2-3, highlighted in yellow color).

“The EXAFS data are unacceptable in the current form. How were the data collected? What was the form of the sample? The fits must be shown. Was fitting attempted without a Pt-Sn scattering path? Where is the Sn K-edge data? Are the scales of the wavelet transforms (Fig. S4) the same? Where are the raw $\chi(k)$ data? What fitting range was used? Etc. etc.”

The Pt L₃-edge and Sn K-edge XAFS spectra were collected at BL14W1 beamline in Shanghai Synchrotron Radiation Facility. Some details of the XAFS tests and data analysis were as follows. The hard X-ray was monochromatized with Si(111) double-crystal monochromator and Si(311) double-crystal monochromator for Pt L₃-edge and Sn K-edge measurement, respectively. The monochromator harmonics was removed by using harmonic rejection mirror. The as-prepared sample was pelletized as disc with the diameter of 10 mm. The XAFS data were collected in fluorescence mode (Lytle detector). The acquired EXAFS data were processed according to the standard procedures using the ATHENA module implemented in the IFEFFIT software packages. The EXAFS curve-fittings were performed using the ARTEMIS module of IFEFFIT [Ravel, B. & Newville, M. ATHENA, ARTEMIS, HEPHAESTUS: data analysis for X-ray absorption spectroscopy using IFEFFIT. *J. Synchrotron Radiat.* **12**, 537-541 (2005)]. In terms of Pt L₃-edge, the curve fitting was done on the k^2 -weighted EXAFS function $\chi(k)$ data in the k -range of 3.0-12.5 Å⁻¹ and in the R -range of 1.0-3.2 Å. The number of independent points for these samples are $N_{\text{ipt}} = 2\Delta k \cdot \Delta R / \pi = 2 \times (12.5 - 3.0) \times (3.2 - 1.0) / \pi \approx 13$. As for Sn foil, the curve fitting was done on the k^2 -weighted EXAFS function $\chi(k)$ data in the k -range of 3.0-12.0 Å⁻¹ and in the R -range of 2.0-3.8 Å. The number of independent points for these samples are $N_{\text{ipt}} = 2\Delta k \cdot \Delta R / \pi = 2 \times (12.0 - 3.0) \times (3.8 - 2.0) / \pi \approx 10$. As for PtSn-Al₂O₃ at Sn K-edge, the curve fitting was done on the k^2 -weighted EXAFS function $\chi(k)$ data in the k -range of 2.5-10.5 Å⁻¹ and in the R -range of 1.0-2.0 Å. The number of independent points for these samples are $N_{\text{ipt}} = 2\Delta k \cdot \Delta R / \pi = 2 \times (10.5 - 2.5) \times (2.0 - 1.0) / \pi \approx 5$. Wavelet transform analysis of Pt L₃-edge were performed in the k -range of 0-12.0 Å⁻¹ using the Igor pro script developed by

Funke et al. [Funke, H. Scheinost, A. C. & Chukalina, M. Wavelet analysis of extended x-ray absorption fine structure data. *Phys. Rev. B* **71**, 094110 (2005)]. The Morlet wavelet was selected as basis mother wavelet and the parameters ($\eta = 8$, $\sigma = 1$) were used for a better resolution in the wave vector k .

As suggested by this reviewer, we have added Sn K-edge XAFS as shown in Figure R1a,b and Table R1. Sn element in PtSn/ γ -Al₂O₃ was determined at the oxidized state. Sn-O bonds occupied the majority with the bond length of 2.01 Å and coordination number of 5. We could not fit the Sn-Pt path possibly because a large proportion of Sn atoms were oxidized to form SnO_x nanoparticles whose signals overshadowed those of Sn-Pt bonds.

Figure R1. (a) Sn K-edge XANES spectra of Sn foil, SnO₂, and PtSn/ γ -Al₂O₃. (b) The corresponding Sn K-edge EXAFS spectra (points) and curvefits (lines). The data are k^3 -weighted and not phase-corrected. k -weighted experimental and fitting spectra with respect to

Pt element in (c) Pt foil, (d) PtO₂, (e) Pt/ γ -Al₂O₃, and (f) PtSn/ γ -Al₂O₃. *k*-weighted experimental and fitting spectra with respect to Sn element in (g) PtSn/ γ -Al₂O₃ and (h) Sn foil.

Table R1 | EXAFS fitting parameters at the Sn K-edge for various samples. S_0^2 was set to 0.90, according to the experimental EXAFS fit of Sn foil reference by fixing coordination numbers as the known crystallographic value.

Sample	Shell	$N^{[a]}$	$R(\text{\AA})^{[b]}$	$\sigma^2(\text{\AA}^2)^{[c]}$	$\Delta E_0(\text{eV})^{[d]}$	$R \text{ factor}^{[e]}$
Sn foil	Sn-O	4.0	3.00±0.01	0.0085±0.0007	9.6±0.8	0.0077
	Sn-Sn	2.0	3.08±0.01			
PtSn/ γ -Al ₂ O ₃	Sn-O	5.0	2.01±0.02	0.0045±0.0035	-0.3±3.4	0.0048

[a] N : coordination numbers.

[b] R : bond distance.

[c] σ^2 : Debye-Waller factors.

[d] ΔE_0 : the inner potential correction.

[e] R factor: goodness of fit.

As suggested by this reviewer, we have also attempted fitting without a Pt-Sn scattering path. Firstly, we have replaced the Pt-Sn path with the Pt-Pt path, in other words, we have used two Pt-Pt paths and one Pt-O path for fitting. The R factor was 0.043, implying the poor matching degree. Then, we have directly removed the Pt-Sn path, in other words, we have used one Pt-Pt path and one Pt-O path for fitting. The R factor reached 0.059, still implying the poor matching degree. Therefore, the Pt-Sn should be taken into consideration.

We have added the details in the experimental section and supplementary information (p. 15, lines 15-31, p. 16, lines 1-4, refs. 37-38, Supplementary Figs. 7-8, Supplementary Table 2, highlighted in yellow color).

“The elemental stoichiometry of the sample should be provided in addition to the wt%. Why was this value chosen?”

As suggested by the reviewer, we have added the molar ratio of Pt to Sn in PtSn/ γ -Al₂O₃. Specifically, the molar ratio of Pt to Sn in PtSn/ γ -Al₂O₃ was 1:1 (p. 4, lines 17-18, highlighted in yellow color). We chose the mass ratio in the original manuscript because the mass ratio also reflects the information of the support whereas the molar ratio of Pt to Sn only reflects the information of the metal. We were unable to accurately calculate the molar ratio of the support because the metal species were partially oxidized.

“How does “absence of peaks for PtSn nanocrystals” imply that the particles were amorphous?”

Thanks for pointing out this issue. It is not rigor to imply the amorphous nature of the particles from the absence of peaks for PtSn nanocrystals because small clusters also exhibited negligible XRD peaks. We have deleted this implication in the revised manuscript (p. 4, lines 23-24, highlighted in yellow color).

“The authors must describe in more detail regarding the sample preparation. What does “quickly pushed into the quartz tube under a H₂ flow” even mean and how was this accomplished, and why did it have to be done quickly?”

The synthesis of PtSn/ γ -Al₂O₃ followed an impregnation-quench method. The quench process was carried out in our modified tube furnace as shown in Figure R2. Besides the common heating furnace, vacuum system, and pipeline system, our modified tube furnace also comprises a magnetically controlling device for rapidly pushing and pulling the sample. When the sample holder was outside the quartz tube, we placed the sample in the end card slot and gave it a short press on multi-function button ⑩. Then the magnetically controlling device pulled the sample into the heating furnace. Afterwards, we sealed the pipeline, vacuumized the system, and purged a H₂ flow, followed by programmed heating and pressing on the injection push button ⑨. When the heating time was over, we gave it a short press on multi-function button ⑩ and rapidly pulling the sample out to the cooling position for the quick cooling. After cooling down the sample, we closed the pipelines, shut off the vacuum system, opened the seal ring, and gave it a long press on multi-function button ⑩ to exit the sample. We have added the details in the experimental section. We have added the details in the revised manuscript (p. 13, lines 13-22, Supplementary Fig. 1, highlighted in yellow color).

Figure R2. Modified tube furnace. ① Heating furnace; ② Vacuum system; ③ Electronic operating interface; ④ Quartz tube; ⑤ Gas pipelines; ⑥ Cooling system; ⑦ Magnetic lever system; ⑧ Motor control system; ⑨ Injection push button; ⑩ Multi-function button (short press for the cooling position and long press for exiting)

“What form of the catalyst was analyzed by XRD, STEM and XAS? This is never described.”

Thanks for raising this issue. The obtained catalyst sample was in the powder form. When we prepared the sample for XRD characterization, we directly placed the catalyst powders in the glass card slot, followed by flattening the surface and leveling out at sample holder plane. The preparation of the STEM sample was as follows. The catalyst powders were dispersed in ethanol, followed by sonification to form the suspension. Afterwards, we dropped the suspension onto the

super-thin carbon film. The dried sample was used for STEM characterizations. As for preparing the XAS sample, the as-prepared catalyst powders were pelletized as disc with the diameter of 10 mm. We have added these details in the experimental section (p. 15, line 20, p. 16, lines 9-11 and 13-14, Supplementary Fig. 1, highlighted in yellow color).

“I strongly disagree with the statement “Elemental analysis of PtSn/ γ -Al₂O₃ indicated the homogeneous distribution of Pt and Sn without obvious phase segregation (Fig. 2c).” First, what does this sentence mean? I assume it means that there is a uniform distribution of Pt and Sn and that the particles are thus PtSn bimetallic. The EDS map does not support this conclusion. It appears to show Sn on the alumina support with some Pt close to the Sn. Indeed this would be expected from the method of sample preparation.”

Thanks for this reviewer’s valuable comments. We agree that the EDS mapping does not support the uniform distribution of Pt and Sn. We have supplemented the Sn K-edge analysis of PtSn/ γ -Al₂O₃ in the revised manuscript. From the result, we found that Sn-O bonds occupied the majority in the absence of Pt-Sn bonds though we observed them during Pt L₃-edge analysis. We can deduce that not all the Pt and Sn atoms were uniformly mixed to form PtSn bimetallic nanoparticles. Instead, a large proportion of Sn atoms were oxidized to form SnO_x nanoparticles, leaving some Sn atoms coordinated with Pt atoms. In PtSn nanoparticles, Sn atoms were more prone to be oxidized than Pt atoms. We have added the relevant discussion in the revised manuscript (p. 4, lines 24-26, p. 5, lines 13-18, Supplementary Fig. 8 and Table 3, highlighted in yellow color).

“There are many aspects of this work that are well known and have been reported for decades. For example it is well known that the addition of Sn to Pt “restrains the cracking of C-C bonds”.”

We admitted that some aspects of our work are well known. Our choice of PtSn nanoparticles was just originated from the relatively mature catalyst for propane dehydrogenation in the UOP Oleflex process. For clarity, we have referred several papers in the relevant sentences (refs. 30-32, p. 4, lines 24-26).

“The TOF values are extremely small, ca 2 per hour. This raises the question of whether what they are measuring is indeed catalytic.”

To identify whether this reaction was stoichiometric or catalytic, we have tested the spent catalyst. Specifically, the first cycle under the standard condition (25 mg of PtSn/ γ -Al₂O₃, 5 mL of water, 6 bar (C₃H₈:N₂ = 5:1), 350 °C, 2 h) was performed multiple groups in parallel to compensate the possible loss of catalysts. The spent catalyst was collected through centrifugation and dried at 60 °C in vacuum. Afterwards, we added 25 mg of the spent catalyst back into the quartz inlet, then the autoclave was pressurized with 6 bar (C₃H₈:N₂ = 5:1) again for the next 2-h operation. As shown in Figure R3, the spent catalyst exhibited similar propane conversion and product distribution to the fresh one. Therefore, the reaction was a catalytic process. We have added the relevant discussion in the revised manuscript (p. 7, lines 6-9, Supplementary Fig. 13, highlighted in yellow color).

Figure R3. Comparison of fresh and spent PtSn/γ-Al₂O₃.

“Why was the reaction carried out at pressure?”

In the original work, we had intended to achieve a relatively high yield of product. Thus we pressurized the reaction. After being reminded the reviewer, we think that it is more important to report the conversion because we claim “*the propane-wet-oxidation process overcomes the equilibrium limitation of non-oxidative propane dehydrogenation*” in the Introduction section. To this end, we have calculated the thermodynamic equilibrium conversion of propane during non-oxidative propane dehydrogenation and compared it with the propane conversion during wet reforming (our work). As shown in Table R2, the propane conversion over PtSn/γ-Al₂O₃ during wet reforming exceeded the thermodynamic limit. Based on the Le Chatelier’s principle, lowering the pressure will increase the conversion of propane for both reactions (eqs. 2 and 3). When the partial pressure of propane was lowered during wet reforming, the propane conversion indeed increased and reached 25.60% under 0.2 bar of C₃H₈ and 0.8 bar of N₂ at 350 °C. We have added relevant discussion in the revised manuscript (p. 6, lines 13-19, Supplementary Table 8, highlighted in yellow color).

Table R2. Comparison between the propane conversion during wet reforming and the thermodynamic equilibrium conversion of propane during non-oxidative propane dehydrogenation. Equilibrium calculations were performed through HSC Chemistry 6 software, which utilizes a Gibbs free energy minimization algorithm.

Reaction conditions	Propane conversion	Thermodynamic equilibrium conversion of propane
5 bar of C ₃ H ₈ and 1 bar of N ₂ , 350 °C	1.84% (2 h)	0.8%
	2.33% (4 h)	
	2.81% (6 h)	

3 bar of C ₃ H ₈ and 1 bar of N ₂ , 350 °C	2.69% (2 h)	1.0%
0.2 bar of C ₃ H ₈ and 0.8 bar of N ₂ , 350 °C	25.60% (2 h)	3.5%

“Nowhere in the manuscript do the authors document the conversion of propane to acetone.”

As requested, we have added the conversion of propane for all the catalytic data in the revised manuscript (Fig. 3d, Supplementary Figs. 11, 13-18, highlighted in yellow color). The conversion was calculated based on the following equation.

$$\text{Conversion} = \frac{n(\text{CO}_2) + n(\text{CH}_4) + 2n(\text{C}_2\text{H}_6) + 3n(\text{C}_2\text{H}_6) + 3n(\text{C}_2\text{H}_8\text{O}) + 3n(\text{C}_2\text{H}_6\text{O})}{3n(\text{C}_3\text{H}_8)_{\text{in}}} \times 100\% \quad (1)$$

REVIEWER COMMENTS

Reviewer #1 (Remarks to the Author):

The author has revised the manuscript very carefully according to the reviewers comments, hence I would like to recommend it to be published in this journal at the present form.

Reviewer #2 (Remarks to the Author):

The efforts that have been made to improve the manuscript are greatly appreciated. The authors attempted to demonstrate the viability of the propane-wet-reforming process by the relatively high acetone selectivity (57.8% among all carbon-based products and 99.3% among liquid products). However, it is worth noting that the efficiency of the propane-wet-reforming process for the production of acetone is quite low. The reaction was carried out in a batch reactor, and for a typical performance test (25 mg PtSn/ γ -Al₂O₃, 5 mL H₂O, 350°C, 6 bar, 15-mL reactor, C₃H₈/N₂=5, 2 h), the conversion of propane was only about 1.8% (2 h) despite exceeding thermodynamic equilibrium conversion of propane (0.8% for non-oxidative propane dehydrogenation). Although the conversion of propane increased to from 1.84% to 25.6% as the partial pressure of propane decreased from 5 bar to 0.2 bar, this increase in propane conversion came at the expense of propane usage, and the rate of propane consumption did not actually increase. Moreover, the authors emphasized the atom economy of the propane-wet-reforming (C₃H₈ + H₂O → C₃H₆O + 2H₂) process with the only by-product of H₂. However, in their proof-of-concept experiments, the H₂O/C₃H₈ ratio (~125) used in the reaction was much higher than the stoichiometric ratio of 1. Therefore, the experimental results are still not fully support the concept of the atom-economic reaction. Considering the inefficiency of the process and insufficiently supportive proof-of-concept experiments, I am afraid of recommending the publication in Nature Communications. The following are some specific comments.

1. The thermodynamic calculations for the propane-wet-reforming process (C₃H₈ + H₂O → C₃H₆O + 2H₂) should be performed to gain the information on the equilibrium conversion of propane and the Gibbs free energy change under different reaction conditions.
2. The authors did not provide the equation for the calculation of TOF.
3. For the KIE test of propane-wet-reforming over PtSn/Al₂O₃, the intercept is not zero, why?
4. The authors mentioned that the propane conversion slightly increased with the decrease in the volume of added water. But as shown in the Supplementary Figure 18, the conversion of propane actually decreased slightly as the volume of added water decreased to 4.8 mL.
5. As shown in Supplementary Fig. 17, the conversion of propane did not increase with increasing the partial pressure of propane in the reaction system. Why?

Reviewer #3 (Remarks to the Author):

The authors have conducted new experiments to provide responses to the earlier reviews and

should be acknowledged for their efforts. However, there are still some improvements that should be made.

Regarding the XAS data, the authors are thanked for now including the Sn K-edge XAS data. However, this now creates a difficulty. The authors claim throughout the manuscript that it is PtSn clusters that are responsible for the catalytic selectivity and activity and yet it is clear that a significant amount of the Sn is not associated with the Pt (from the EDX mapping and XAS). What is clearly missing from this manuscript is operando XAS characterization to determine the structure of the catalyst under reaction conditions. What is unknown is the stoichiometry of the PtSn clusters that are responsible for the activity. It also brings up the question of what is the effect of the Sn/alumina on the catalytic chemistry? The authors compare the reactivity to Pt/Alumina, but they also need to check the activity of the Sn-modified alumina.

Some specific comments:

The authors cannot claim that the Pt is in the metallic state as they show that there is significant Pt-O bonding.

A lower coordination number does not imply that the structure is amorphous – just that it is present as a nanoparticle.

In Figure 2e the authors put tick marks on the peaks- these should be removed as the Pt-Pt and Pt-Sn scattering paths are not resolved from each another, so these are misleading.

The authors are thanked for now showing the XAS fits in the SI. However, they show the fits in k-space, and as all frequencies are not fit in the model it is difficult to visualize the adequacy of the fit. The fits should also be shown in R-space.

Table R1 is incorrect. Why does the fit for Sn foil include both Sn-O and Sn-Sn scattering paths? There is no intensity scale in Figure S6. Are the wavelet transforms all plotted on the same intensity scale?

In Supplementary Table 2 what are there no error bars on the CNs? Also, what is the reason for the exceptionally large sigma square values for the Pt-O scattering paths?

What is still missing from the details of the XAS measurements and analysis is the specific structure or cif file that was used to extract the Pt-O and Pt-Sn scattering paths, etc.

I previously asked “What form of the catalyst was analyzed by XRD, STEM and XAS? This is never described.” The authors replied “Thanks for raising this issue. The obtained catalyst sample was in the powder form.” However, my question was with regards to an as-synthesized catalyst, or after reduction or after reaction. This brings into question regarding what characterization was performed on spent catalyst? The authors only mention Raman.

How was the TOF calculated?

Point-by-point response to reviewers' comments

Manuscript ID: NCOMMS-23-30241A

Title: Acetone synthesis from atom-economic catalysis of propane wet oxidation

Reviewer #1

“The author has revised the manuscript very carefully according to the reviewers comments, hence I would like to recommend it to be published in this journal at the present form.”

We sincerely thank this reviewer for his/her careful reading of our manuscript.

Reviewer #2

“The efforts that have been made to improve the manuscript are greatly appreciated. The authors attempted to demonstrate the viability of the propane-wet-reforming process by the relatively high acetone selectivity (57.8% among all carbon-based products and 99.3% among liquid products). However, it is worth noting that the efficiency of the propane-wet-reforming process for the production of acetone is quite low. The reaction was carried out in a batch reactor, and for a typical performance test (25 mg PtSn/ γ -Al₂O₃, 5 mL H₂O, 350 °C, 6 bar, 15-mL reactor, C₃H₈/N₂=5, 2 h), the conversion of propane was only about 1.8% (2 h) despite exceeding thermodynamic equilibrium conversion of propane (0.8% for non-oxidative propane dehydrogenation). Although the conversion of propane increased to from 1.84% to 25.6% as the partial pressure of propane decreased from 5 bar to 0.2 bar, this increase in propane conversion came at the expense of propane usage, and the rate of propane consumption did not actually increase. Moreover, the authors emphasized the atom economy of the propane-wet-reforming (C₃H₈ + H₂O → C₃H₆O + 2H₂) process with the only by-product of H₂. However, in their proof-of-concept experiments, the H₂O/C₃H₈ ratio (~125) used in the reaction was much higher than the stoichiometric ratio of 1. Therefore, the experimental results are still not fully support the concept of the atom-economic reaction. Considering the inefficiency of the process and insufficiently supportive proof-of-concept experiments, I am afraid of recommending the publication in Nature Communications. The following are some specific comments.”

We sincerely thank this reviewer for his/her valuable comments. Actually, the concept of the atom-economic reaction for the propane-wet-reforming process is our final goal. For instance, methane oxidation into methanol (2CH₄ + O₂ → 2CH₃OH) and propylene epoxidation with molecular oxygen (2C₃H₆ + O₂ → 2C₃H₆O) are both dream reactions with 100% atom economy, but their practical conversion and yield of the desired products were still far from the ideal values [Chem 2021, 7, 2270-2276; J. Am. Chem. Soc. 2022, 144, 4260-4268]. As for the reaction rates, we calculated the TOF number by counting total Pt atoms in the previous version. However, this calculation is not strict since not all the Pt atoms participated in the reaction. In the revised manuscript, we have calculated the TOF number by only counting surface Pt atoms, because heterogeneous catalysis occurs on the surface of catalysts. Based on the revised calculation, the TOF number increased by 7.4 times from 5.6 h⁻¹ to 41.3 h⁻¹. The calculations details have been provided in the response to this reviewer's second question. Anyway, we agree with this reviewer

that it is inappropriate to highlight this concept due to the relatively low conversion of propane. Therefore, we have changed the theme from emphasizing the concept of an atom-economic reaction to highlighting a novel route to directly produce acetone from propane in the revised manuscript (p. 1, lines 3-4, p. 2, lines 5-7, p. 6, lines 11-14, p. 16, lines 11-28, p. 17, lines 1-2, highlighted in yellow color).

This reviewer also mentioned that the $\text{H}_2\text{O}/\text{C}_3\text{H}_8$ ratio (~ 125) used in the reaction was much higher than the stoichiometric ratio of 1. As suggested by this reviewer, we have conducted the reaction by maintaining a 1:1 molar ratio of water to propane, while keeping all other operations consistent in the revised manuscript. After a 2-h reaction at 350 °C, the reactor was cooled by blowing air from the outside bottom of the reactor for facilitating the recovery of liquid-phase products in the reactor bottom. Subsequently, the bottom was washed with a quantified amount of water to recover as much liquid-phase product as possible, while simultaneously detecting the gas-phase products. As shown in Figure R1, it is evident that the catalytic products still included acetone and H_2 . Different from the case of excess water, when the water content was near the stoichiometric ratio, propylene occupied a higher proportion. Due to the limited water content (approximately 50 μL), the catalyst was not fully infiltrated by water, let alone thoroughly stirred, resulting in the incomplete conversion of propylene to acetone. As we gradually increased the water dosage from 50 μL to 100 μL and up to 1 mL, the yields of acetone and H_2 both increased gradually. This implies the crucial role of water partial pressure, attributed to its ability to enhance mass and heat transfer processes. Notably, at a water/propane ratio of 1, the acetone concentration was approximately 22.3 $\mu\text{mol mL}^{-1}$, while increasing the water/propane ratio to 2 resulted in a noticeable decrease to 14.4 $\mu\text{mol mL}^{-1}$. When the water/propane ratio was approximately 135, *i.e.*, the water content used under the standard condition was 5 mL, the concentration of acetone was 4.3 $\mu\text{mol mL}^{-1}$. This underscores the pivotal role of water in efficient mass transfer, reducing the product concentration on the catalyst surface and consequently promoting the progress of the catalytic reaction. Meanwhile, we observed a significant increase in the yield of propylene, indicating that the dehydrogenation of propane can be enhanced under high water partial pressure, providing more intermediates for the subsequent conversion of propylene into acetone. From Table R1, we can see that as the water content gradually increased from stoichiometric value to excess, the yield of acetone increased from 44.53 $\mu\text{mol g}^{-1}$ to 858.42 $\mu\text{mol g}^{-1}$, while the conversion also increased from 0.17% to 1.84%. In conclusion, the use of excess water enhanced heat and mass transfer processes, while the concentration diffusion of intermediates and product drives the reaction equilibrium. It is acknowledged that the claim of strict atom economy is not entirely accurate, especially when the water/propane ratio is excessively high. Thus, we have corrected our statement and added relevant discussion in the revised manuscript (p. 9, lines 7-30, Supplementary Fig. 26, Supplementary Table 13, highlighted in yellow color).

Figure R1. Dependence of product yields on on the molar ratio of H₂O/C₃H₈. Typically, 25 mg of PtSn/ γ -Al₂O₃ (>60 mesh) and different volumes of water were loaded in a 15-mL slurry reactor with the stirring speed of 600 rpm to operate under 6 bar (C₃H₈:N₂ = 5:1) at 350 °C for 2 h.

Table R1. Dependence of yield and conversion on the molar ratio of H₂O/C₃H₈.

Entry	H ₂ O/C ₃ H ₈	V _{H₂O} (mL)	n _{Acetone} (μmol g ⁻¹)	Conversion (%)
1	1	0.05	44.53 ± 4.17	0.17 ± 0.01
2	2	0.10	57.51 ± 6.37	0.21 ± 0.00
3	6	0.30	84.83 ± 2.05	0.34 ± 0.01
4	19	1.00	431.94 ± 9.84	0.85 ± 0.03
5	135	5.00	858.42 ± 30.33	1.84 ± 0.04

“1. The thermodynamic calculations for the propane-wet-reforming process ($C_3H_8 + H_2O \rightarrow C_3H_6O + 2H_2$) should be performed to gain the information on the equilibrium conversion of propane and the Gibbs free energy change under different reaction conditions.”

We genuinely thank this reviewer for his constructive suggestions. In the previous response, we only provided the equilibrium conversion of propane for the non-oxidative propane dehydrogenation. We found that the experimental conversion of propane during the wet reforming was higher the equilibrium value during the propane dehydrogenation. After being reminded by this reviewer, we realize that it is also important to offer the equilibrium conversion of propane for the propane-wet-reforming process. As suggested, we have calculated the equilibrium conversion of propane and the Gibbs free energy change under different reaction conditions in the revised manuscript. The calculations were carried out by using HSC 6 software.

Firstly, we calculated the standard reaction Gibbs energy ($\Delta_r G^\ominus$) for the propane-wet-reforming process at 350 °C. Under ideal condition at 350 °C, the saturated vapor pressure of water is 16521 kPa, while the density of liquid water is 0.59 g mL⁻¹ under 16521 kPa at 350 °C. The volume of added water at room temperature is 5 mL, while that of the reactor is 15 mL. It was estimated that the water in the slurry reactor existed in the form of liquid at 350 °C, containing 0.47 g of gaseous water and 4.54 g of liquid water. Considering that the catalyst was immersed in

the water, we assumed the water at the liquid state during the calculation. Considering that acetone can be completely dissolved in the water and that the concentration of acetone was extremely low during the experiments, we assumed the acetone as the solute in water. Taking the above assumptions into account, we input the reaction equation of “ $C_3H_8(g) + H_2O(l) \rightarrow C_3H_6O(aq) + 2H_2(g)$ ” in the module of HSC Chemistry 6-Reaction Equation. Thus, the value of $\Delta_r G^\ominus$ at 350 °C was calculated as 44.74 kJ mol⁻¹, while the corresponding standard equilibrium constant is 1.776×10^{-4} .

Then, we calculated the equilibrium conversion of propane under different conditions by using the module of HSC Chemistry 6-Equilibrium Compositions. $C_3H_8(PPEg)$, H_2O , $C_3H_6O(PREg)$, and $H_2(g)$ were selected as the reactants. The reaction temperature was set at 350 °C. The initial compositions of reactants were listed in Table R2 and input into the calculation module. The equilibrium conversion of propane was obtained and shown in Figure R2. As shown in Figure R2, the equilibrium conversion of propane decreases with the increase of the propane partial pressure. As such, the equilibrium conversion of propane in condition 3 is much higher than those in conditions 1 and 2, corresponding to the experimental value. It is worth noting that the equilibrium conversion of propane varies slightly under the propane partial pressure above 5 bar for conditions 1 and 2 because the propane conversion is rather low. This phenomenon is consistent with the experimental results where the conversion of propane did not increase with increasing the partial pressure of propane in the reaction system (Supplementary Fig. 24) as this reviewer raised in his/her fifth question.

Table R2. Initial compositions of reactants under different conditions.

Condition	C_3H_8	H_2O	N_2	C_3H_6O	H_2
1	2.05 mmol	0.28 mol	0.41 mmol	0 mmol	0 mmol
2	1.23 mmol	0.28 mol	0.41 mmol	0 mmol	0 mmol
3	0.08 mmol	0.28 mol	0.33 mmol	0 mmol	0 mmol

Figure R2. Dependence of equilibrium conversion on equilibrium pressure for propane-wet-reforming process.

Since these three conditions are at the same reaction temperature, they have the same standard equilibrium constant (eq. R1). The corresponding variation of the compositions are listed in Table R3.

$$K^\theta = \frac{([H_2]/p^\theta)^2 [C_3H_6O]/c^\theta}{[C_3H_8]/p^\theta} \quad (R1)$$

Table R3. Equilibrium Compositions for $C_3H_8(g) + H_2O(l) \rightarrow C_3H_6O(aq) + 2H_2(g)$.

	$C_3H_8(g)/(\text{bar})$	$C_3H_6O(aq)/(\text{mol L}^{-1})$	$H_2(g)/(\text{bar})$	$N_2(g)/(\text{bar})$
Initial	p_1	0	0	pN
Change	x	$0.0183x^a$	$2x$	pN
Eq.	$p_1 - x$	$0.0183x$	$2x$	pN

^aCalculation based on the equation R2.

$$c = \frac{n}{V_1} = \frac{pV_g}{RTV_1} = \frac{100x \times 0.0073}{8.314 \times 623 \times 0.0077} \text{ mol L}^{-1} = 0.0183x \text{ mol L}^{-1} \quad (R2)$$

At 350 °C, the volume of water is 7.7 mL, while the volume of total gas is 7.3 mL. Based on the equation $PV/T = \text{constant}$, the initial partial pressures of propane under conditions 1-3 are 14.6, 8.8, and 0.58 bar, respectively, while those of N_2 are 2.92, 2.92, and 2.33 bar, respectively. We plug these values into equation R1 and obtain equation R3. After solving the equation R3, the values of x under conditions 1-3 are 0.15, 0.13, and 0.05, respectively. The corresponding conversions are 1.03%, 1.48%, and 8.62%, respectively.

$$1.776 \times 10^{-4} = \frac{0.0183x \times (2x)^2}{p_1 - x} \quad (R3)$$

Finally, we calculated non-standard reaction Gibbs energy ($\Delta_r G$) based on equations R4 and R5.

$$\Delta_r G = \Delta_r G^\ominus + nRT \ln(p/p^\ominus) \quad (\text{Gas}) \quad (R4)$$

$$\Delta_r G = \Delta_r G^\ominus + nRT \ln(c/c^\ominus) \quad (\text{Solute}) \quad (R5)$$

Since the initial partial pressure of H_2 and the initial concentration of acetone are zero, it is meaningless to calculate $\Delta_r G$. As such, we calculated the values of $\Delta_r G$ after 2-h reaction based on equation R6 to judge whether the reaction reaches the equilibrium state. In equation R6, Q is the reaction quotient defined in equation R7.

$$\Delta_r G = \Delta_r G^\ominus + nRT \ln Q \quad (R6)$$

$$Q = \frac{(p(H_2)/p^\theta)^2 c(C_3H_6O)/c^\theta}{p(C_3H_8)/p^\theta} \quad (R7)$$

We plug the compositions after 2-h reaction and the standard reaction Gibbs energy into equation

R6. The corresponding values of $\Delta_r G$ are calculated as -10.70 kJ/mol, -5.76 kJ/mol, and +2.66 kJ mol⁻¹.

In conclusion, we summarized the calculation results in Table R4.

Table R4. Comparison between the propane conversion and the thermodynamic equilibrium conversion of propane during wet reforming. Equilibrium calculations were performed through HSC Chemistry 6 software, which utilizes a Gibbs free energy minimization algorithm. Reaction condition: 5 mL of H₂O, 350 °C, 2 h.

Condition	Gas composition	Propane conversion	Acetone selectivity	Yield of Acetone	Thermodynamic equilibrium conversion	Gibbs free energy change
1	5 bar of C ₃ H ₈ , 1 bar of N ₂	1.84%	57.82%	1.06%	1.03%	-10.70 kJ/mol
2	3 bar of C ₃ H ₈ , 1 bar of N ₂	2.69%	37.53%	1.01%	1.48%	-5.76 kJ/mol
3	0.2 bar of C ₃ H ₈ , 0.8 bar of N ₂	25.60%	35.85%	9.18%	8.62%	+2.66 kJ/mol

From Table R4, the thermodynamic equilibrium conversion of propane into acetone increases with the decrease of propane partial pressure, especially for condition 3, corresponding to the experimental results. During the calculation, we adopted some simplifications such as ignoring the trace water vapor, by-product gases, and consumption during the heating process. These simplifications resulted in minor deviations. For example, the experimental yield of acetone was slightly higher than the thermodynamic equilibrium conversion for the conditions 1 and 3. We have added relevant discussion in the revised manuscript (p. 6, lines 18-22, Supplementary Fig. 12, Supplementary Table 10, highlighted in yellow color).

“2. The authors did not provide the equation for the calculation of TOF.”

As suggested, we have provided the details concerning the calculation of TOF values in the revised manuscript. In the previous manuscript, we calculated the TOF values by assuming all Pt atoms as the active sites (eqs. R8 and R9).

$$\text{TOF} = \frac{\sum_i i \times n_i}{3 \times n_{\text{Pt}} \times t} \quad (\text{R8})$$

$$n_{\text{Pt}} = \frac{m_{\text{catal.}} \times w_{\text{Pt}}}{\mu_{\text{Pt}}} \quad (\text{R9})$$

In this equation, *i* refers to the carbon number. Taking C₃H₆O as an example, the *i* value is 3. *n_i* refers to the moles of the product with the carbon number of *i*. The value “3” in the denominator means that the carbon number of propane is 3. *n_{Pt}* is the moles of Pt atoms in the catalyst. *t* is the reaction time. *m_{catal.}* refers to the mass of the catalyst. *w_{Pt}* refers to the mass loading of Pt in the catalyst. *μ_{Pt}* is the weight of one mole of Pt atoms. However, this calculation was not appropriate since not all the Pt atoms participated in the reaction. In the revised manuscript, we have calculated the TOF number by only counting the surface Pt atoms (eq. R10), because

heterogeneous catalysis occurs on the surface of catalysts.

$$\text{TOF} = \frac{\sum_i i \times n_i}{3 \times n_{\text{surface Pt}} \times t} \quad (\text{R10})$$

$$n_{\text{surface Pt}} = \frac{m_{\text{catal.}} \times n_{\text{CO}}}{50 \text{ mg}} \quad (\text{R11})$$

The moles of surface Pt atoms ($n_{\text{surface Pt}}$) were measured according to CO pulse titration (eq. R11) where n_{CO} is the moles of adsorbed CO molecules. CO pulse titration was carried out as follows. Specifically, 50 mg of catalysts were added into the cell, followed by pulsing 10%CO with the quantitative loop volume of 450 μL at room temperature. According to Figure R3, the mole of total adsorbed CO molecules on the catalyst was calculated as 0.897 μmol (Table R5). We assumed that one Pt atom adsorbed one CO molecule. As such, 50 mg of catalysts contained 0.897 μmol of surface Pt atoms. Following the equation 3, the TOF number reached 41.3 h^{-1} . It was worth noting that the time zero point was counted when the system temperature reached the target temperature. However, the reaction had already occurred before the time zero point. When we calculated the TOF number, we subtracted the amount of products at the time zero point from the total yields at the reaction time. In the revised manuscript, we have replaced all the original TOF numbers with the revised ones based on surface Pt atoms and added relevant discussion (p. 6, lines 11-14, p. 16, lines 11-28, p. 17, lines 1-2, Supplementary Fig. 11, Supplementary Table 8, highlighted in yellow color).

Figure R3. CO pulse titration.

Table R5. Result of CO pulse titration.

CO	Unsaturated peak			Saturated peak		
	1st	2nd	3rd	4th	5th	6th
A_{Integral}	17.73667	27.64333	28.16417	28.73667	28.80667	28.87083
$A_{\text{Adsorption}}$	11.06805	1.161393	0.640553	/	/	/
$n_{\text{Adsorption}} (\mu\text{mol})$	0.771	0.081	0.045	/	/	/
$\Sigma n_{\text{Adsorption}} (\mu\text{mol})$	0.897			/		

“3. For the KIE test of propane-wet-reforming over PtSn/Al₂O₃, the intercept is not zero, why?”

Thanks for pointing out this issue. The none-zero intercept was ascribed to the catalyst slurry reactor. The KIE experiments were carried out at 350 °C. The time zero point was counted when the system temperature reached 350 °C. However, the reaction had already occurred before the time zero point. As such, the intercept was not zero. Actually, the reaction rate was reflected by the slope instead of the intercept. In other words, the intercept did not influence the KIE value. We have added relevant discussion in the revised manuscript (p. S45, the note of Supplementary Fig. 27, highlighted in yellow color).

“4. The authors mentioned that the propane conversion slightly increased with the decrease in the volume of added water. But as shown in the Supplementary Figure 18, the conversion of propane actually decreased slightly as the volume of added water decreased to 4.8 mL.”

Thanks for raising this point. After being reminded by this reviewer, we found that our previous statement was somewhat imprecise. The influence of water volume was complex. The slight variation in water volume from 5.00 mL to 4.90 mL did not affect the mass and heat transfer. Thus the minor increase in acetone yield was ascribed to the increased amount of propane molecules as discussed in the previous version. When the volume water was changed from 4.90 mL to 4.80 mL, the mass and heat transfer were inhibited even though the propane volume increased. In other words, a substantial reduction in the volume of water leads to a decrease in the conversion of propane just as stated in the first response (Fig. R1). For accuracy, we have added the relevant discussion in the revised manuscript (p. 8, lines 27-31, p. 9, lines 1 and 3-6, highlighted in yellow color).

Table R6 Dependence of products and conversion on the volume of added water.

Entry	$V_{\text{H}_2\text{O}}$ (mL)	$n_{\text{Acetone}} (\mu\text{mol g}^{-1})$	Conversion (%)
1	5.00	858.42	1.84 ± 0.04
2	4.95	908.33	2.08 ± 0.05

3	4.90	917.72	2.15 ± 0.03
4	4.80	778.18	2.05 ± 0.05

“5. As shown in Supplementary Fig. 17, the conversion of propane did not increase with increasing the partial pressure of propane in the reaction system. Why?”

We genuinely thank this reviewer for pointing out this issue. The reaction order with respect to propane was 1.12. Thus, we can approximately regard it as the first-order reaction with respect to propane. The steady conversion regardless of the partial pressure of propane was attributed to the first-order reaction with respect to propane and the low conversion (~2%) of propane. The differential form of the first-order reaction is written as equation R12 and rewritten as equation R13.

$$\frac{dc(t)}{dt} = -k c(t) \quad (\text{R12})$$

$$\frac{dc(t)}{c(t)} = -k dt \quad (\text{R13})$$

In the equations R12 and R13, $c(t)$ refers to the concentration (or the partial pressure) of propane as a function of reaction time. k refers to the rate coefficient. When the conversion of propane was low, the concentration of propane approximated to the initial value (eq. R14).

$$c(t) \approx c(t = 0) \quad (\text{R14})$$

As such, the equation R13 can be written as the following (eq. R15).

$$\frac{dc(t)}{c(t = 0)} = -k dt \quad (\text{R15})$$

The left term is just the conversion of propane. In other words, the conversion of propane was only related to the reaction time and independent of the partial pressure of propane at the initial reaction stage when the conversion was low (eq. R16). Moreover, the equilibrium conversion of propane also varies slightly with the change of propane partial pressure at a low conversion level (Fig. R2). For accuracy, we have added the relevant discussion in the revised manuscript (p. 6, lines 20-22, p. 8, lines 20-25, Supplementary Figs. 12 and 24, the note of Supplementary Fig. 24, highlighted in yellow color).

$$\text{Conversion} = - \int_{t=0}^t \frac{dc(t)}{c(t=0)} = \int_{t=0}^t k dt = kt \quad (\text{R16})$$

Reviewer #3

“The authors have conducted new experiments to provide responses to the earlier reviews and should be acknowledged for their efforts. However, there are still some improvements that should

be made.

Regarding the XAS data, the authors are thanked for now including the Sn K-edge XAS data. However, this now creates a difficulty. The authors claim throughout the manuscript that it is PtSn clusters that are responsible for the catalytic selectivity and activity and yet it is clear that a significant amount of the Sn is not associated with the Pt (from the EDX mapping and XAS). What is clearly missing from this manuscript is *operando* XAS characterization to determine the structure of the catalyst under reaction conditions. What is unknown is the stoichiometry of the PtSn clusters that are responsible for the activity. It also brings up the question of what is the effect of the Sn/alumina on the catalytic chemistry? The authors compare the reactivity to Pt/Alumina, but they also need to check the activity of the Sn-modified alumina.”

We sincerely thanks this reviewer’s valuable comments. As suggested by this reviewer, we have conducted *operando* XAS characterizations to investigate the evolution of coordination structures in the revised manuscript. *Operando* XAS tests were carried out using a homemade cell, wherein the investigated catalysts were loaded as compressed discs (2 MPa, $\phi = 10$ mm) at the center of the sample stage. Wet propane was introduced into the cell through bubbling, while the cell was heated to 350 °C by the heater and protected by the cooling water, thereby monitoring the *operando* environment. Figure R1 displays a photograph of this system. The rear of the reaction cell was covered with a Be film to prevent gas leakage and facilitate X-ray transmission. XAS spectra were acquired in transmission mode. Based on the *operando* XAS data, we found that the coordination number of Pt-O bonds in the *operando* sample was slightly increased to 0.5, compared with that (0.3) in the fresh sample, meanwhile the coordination number of Pt-Pt bonds was decreased (Fig. R2 and Table R1). This trend was consistent with the comparison between the spent and the fresh catalysts (Fig. R9 and Table R4). Notably, Pt species were slightly more oxidized after the *operando* treatment, contradicting the coordination number of Pt-O and the trend of the spent sample (Fig. R9a). We presumably attributed this phenomenon to the influence of the *operando* cell and temperature. When the fresh sample was placed into the cell at 350 °C without any gas flow, the oxidation state of Pt species was almost the same as that in the *operando* sample (Fig. R3). We have added the relevant discussion in the revised manuscript (p. 11, lines 24-30, p. 19, lines 8-14, Supplementary Figs. 30-31, Supplementary Tables 16-17, highlighted in yellow color).

Figure R1. Photograph of the setup for the *operando* XAS test.

Figure R2. (a) Pt L₃-edge XANES spectra and (b) the corresponding Pt L₃-edge EXAFS spectra of fresh and *operando* PtSn/γ-Al₂O₃. (c) Sn K-edge XANES spectra and (d) the corresponding Sn K-edge EXAFS spectra of fresh and *operando* PtSn/γ-Al₂O₃. *Operando* condition: 350 °C, wet propane at 20 mL min⁻¹.

Figure R3. Pt L₃-edge XANES spectra of fresh PtSn/γ-Al₂O₃, PtSn/γ-Al₂O₃ at 350 °C, and *operando* PtSn/γ-Al₂O₃.

Table R1. EXAFS fitting parameters at the Pt L₃-edge for *operando* PtSn/ γ -Al₂O₃.

Sample	Shell	N^a	$R(\text{\AA})^b$	$\sigma^2(\text{\AA}^2)^c$	$\Delta E_0(\text{eV})^d$	R factor
Fresh sample	Pt-O	0.3±0.1	1.95±0.01	0.0155±0.0052	3.9±2.0	0.0020
	Pt-Sn	1.6±0.2	2.70±0.02	0.0064±0.0030		
	Pt-Pt	4.3±0.4	2.74±0.01	0.0013±0.0012		
Operando sample	Pt-O	0.5±0.7	2.02±0.07	0.0019±0.0011	6.2±0.8	0.0048
	Pt-Sn	1.3±0.7	2.68±0.11	0.0064 ^e		
	Pt-Pt	3.8±1.4	2.73±0.01	0.0059±0.0082		

[a] N : coordination numbers.

[b] R : bond distance.

[c] σ^2 : Debye-Waller factors.

[d] ΔE_0 : the inner potential correction.

[e] The σ^2 value of the Pt-Sn path was fixed at 0.0064 for the fitting due to the following reason. We observed a marginal decline in the signal-to-noise ratio in the spectral data of *operando* samples. To ensure the quality of fitting, we set the σ^2 of the Pt-Sn path to 0.0064, a value derived from reference to static fitting results. The rationale behind fixing the Pt-Sn path arose from our analysis of *operando* data, wherein we noticed a more pronounced peak intensity variation in the first shell of the sample's R space compared to the static condition. This variation was mainly attributed to Pt-O interaction. Furthermore, the coordination number of Pt-Pt has the highest overall proportion. As a result, rather than imposing constraints on Pt-O and Pt-Pt, we fixed the σ^2 specifically for the Pt-Sn path.

Table R2. EXAFS fitting parameters at the Sn K-edge for *operando* PtSn/ γ -Al₂O₃.

Sample	Shell	N^a	$R(\text{\AA})^b$	$\sigma^2(\text{\AA}^2)^c$	$\Delta E_0(\text{eV})^d$	R factor
Fresh sample	Sn-O	5.0±0.5	2.01±0.02	0.0045±0.0035	-0.3±0.4	0.0048
Operando sample	Sn-O	5.1±0.6	2.02±0.01	0.0065±0.0015	3.4±1.6	0.0060

[a] N : coordination numbers.

[b] R : bond distance.

[c] σ^2 : Debye-Waller factors.

[d] ΔE_0 : the inner potential correction.

To determine the catalytic role of Sn in Sn/ γ -Al₂O₃, we have compared the catalytic performance of Sn/ γ -Al₂O₃ with that of PtSn/ γ -Al₂O₃. When Sn/ γ -Al₂O₃ was used as the catalyst, the obtained products were negligible (Fig. R4). This is consistent with previous research on propane dehydrogenation, where Sn primarily acts to dilute Pt and modulate the electronic structure, enabling PtSn to exhibit excellent dehydrogenation performance, while pure Sn/ γ -Al₂O₃ was almost inert towards propane dehydrogenation. We have added the relevant discussion in the revised manuscript (p. 6, lines 22-27, Supplementary Fig. 13, highlighted in yellow color).

Figure R4. Comparison of yield and conversion for Sn/γ-Al₂O₃ and PtSn/γ-Al₂O₃. Typically, 25 mg of the catalyst was operated in 5 mL of water under 6 bar (C₃H₈:N₂ = 5:1) at 350 °C for 2 h.

“Some specific comments:

The authors cannot claim that the Pt is in the metallic state as they show that there is significant Pt-O bonding.”

Thanks for pointing out this issue. As suggested, we have removed the claim that Pt is in the metallic state in the revised manuscript (p. 4, line 31, p. 5, line 1, highlighted in yellow color).

“A lower coordination number does not imply that the structure is amorphous – just that it is present as a nanoparticle.”

We agree with this reviewer’s comment. The low coordination number does not imply the amorphous nature, since the coordination number lowers with the decreased size of nanoparticles. For accuracy, we have deleted this statement in the revised manuscript (p. 5, lines 4-5, highlighted in yellow color).

“In Figure 2e the authors put tick marks on the peaks- these should be removed as the Pt-Pt and Pt-Sn scattering paths are not resolved from each another, so these are misleading.”

Thanks for this helpful suggestion. As suggested, we have removed the tick marks on the peaks in the revised Figure 2e.

“The authors are thanked for now showing the XAS fits in the SI. However, they show the fits in k-space, and as all frequencies are not fit in the model it is difficult to visualize the adequacy of the fit. The fits should also be shown in R-space.”

We are sorry for misleading this reviewer. We have already provided the fitting in R-space in the previous manuscript where the dashed lines and solid lines represented the experimental data and fitting data, respectively. However, we had also used circles to represent the experimental data in the previous manuscript. In the revised manuscript, we have adopted circles to unify the representation of experimental data (Fig. 2e, Supplementary Figs. 7, 8, 19, 31).

“Table R1 is incorrect. Why does the fit for Sn foil include both Sn-O and Sn-Sn scattering paths?”

We are sorry for this mistake. Actually, the Sn-O scattering path should also be the Sn-Sn scattering path. Actually, there are two Sn-Sn scattering paths for fitting the Sn foil as shown in Table R3. We have corrected this mistake in the revised supplementary information (p. S12, Supplementary Table 3, highlighted in yellow color).

Table R3. EXAFS fitting parameters at the Sn K-edge for various samples.

Sample	Shell	$N^{[a]}$	$R(\text{\AA})^{[b]}$	$\sigma^2(\text{\AA}^2)^{[c]}$	$\Delta E_0(\text{eV})^{[d]}$	$R \text{ factor}^{[e]}$
Sn foil	Sn-Sn	4.0	3.00 ± 0.01	0.0085 ± 0.0007	9.6 ± 0.8	0.0077
	Sn-Sn	2.0	3.08 ± 0.01			
PtSn/ γ -Al ₂ O ₃	Sn-O	5.0 ± 0.5	2.01 ± 0.02	0.0045 ± 0.0035	-0.3 ± 3.4	0.0048

[a] N : coordination numbers.

[b] R : bond distance.

[c] σ^2 : Debye-Waller factors.

[d] ΔE_0 : the inner potential correction.

[e] R factor: goodness of fit. S_0^2 was set to 0.90, according to the experimental EXAFS fit of Sn foil reference by fixing coordination numbers as the known crystallographic value.

“There is no intensity scale in Figure S6. Are the wavelet transforms all plotted on the same intensity scale?”

As required, we have added the corresponding intensity scale as shown in Figure R5 in the revised manuscript (p. S8, Supplementary Fig. 6, highlighted in yellow color)..

Figure R5 | Wavelet transforms for the k^3 -weighted EXAFS signals. (a) PtSn/ γ -Al₂O₃, (b) Pt/ γ -Al₂O₃, (c) Pt foil, and (d) PtO₂.

“In Supplementary Table 2 what are there no error bars on the CNs? Also, what is the reason for the exceptionally large sigma square values for the Pt-O scattering paths?”

As requested, we have added the error bars on the coordination numbers in the revised manuscript. We attribute the large Debye-Waller factors (σ^2) to that the amorphous structure led to the increase in the disorder degree of the system [Phys. Rev. B 2009, 79, 195203; J. Synchrotron Rad. 2015, 22, 1242-1257]. This phenomenon is consistent with our HAADF-STEM results (Fig. 2b and Supplementary Fig. 3) which proved the amorphous nature of the sample. We have added the relevant discussion in the revised manuscript (p. S10, the note of Supplementary Table 2, highlighted in yellow color).

“What is still missing from the details of the XAS measurements and analysis is the specific structure or cif file that was used to extract the Pt-O and Pt-Sn scattering paths, etc.”

As suggested, we have uploaded .inp files for the XAS analysis.

“I previously asked “What form of the catalyst was analyzed by XRD, STEM and XAS? This is never described.” The authors replied “Thanks for raising this issue. The obtained catalyst sample was in the powder form.” However, my question was with regards to an as-synthesized catalyst, or after reduction or after reaction. This brings into question regarding what characterization was performed on spent catalyst? The authors only mention Raman.”

We sincerely apologize for misunderstanding this reviewer’s previous question. In the previous manuscript, the analyses by XRD, STEM, and XAS were carried out on the fresh sample. As required by this reviewer, we have added the characterizations of the spent catalyst by means of HAADF-STEM, CO-DRIFTS, and XAS in the revised manuscript. Based on the HAADF-STEM images of spent PtSn/ γ -Al₂O₃ (Fig. R6), we still observed that the metal clusters were amorphous without the long-range ordering of metal atoms in an individual cluster. The average size of metal clusters in spent PtSn/ γ -Al₂O₃ was estimated as 1.78 nm (Fig. R7), approximating to that (1.74 nm) of in the fresh sample. In the CO-DRIFTS spectra of spent PtSn/ γ -Al₂O₃ (Fig. R8), the peaks were located at 2082, 2065, and 2040 cm⁻¹. These wavenumbers were slightly higher than the corresponding values (2077, 2060, and 2036 cm⁻¹) in the spectra of the fresh sample, implying the slightly oxidized Pt species or the slightly decreased degree of coordinate unsaturation. XANES results also supported that the oxidation state of Pt species in spent PtSn/ γ -Al₂O₃ was higher than that in the fresh one (Fig. R9). This result was further verified by EXAFS results where the Pt-O coordination number (0.8±0.1) in the spent sample became higher than that in the fresh one (Fig. R9, Tables R4 and R5). We have added the relevant discussion in the revised manuscript (p. 7, lines 11-23, Supplementary Figs. 16-19, Supplementary Tables 11-12, highlighted in yellow color).

Figure R6. Magnified HAADF-STEM image of spent PtSn/ γ -Al₂O₃.

Figure R7. Size distribution of metal nanoparticles in spent PtSn/ γ -Al₂O₃.

Figure R8. CO-DRIFTS spectra of spent PtSn/γ-Al₂O₃.

Figure R9. (a) Pt L₃-edge XANES spectra and (b) the corresponding Pt L₃-edge EXAFS spectra of fresh and spent PtSn/γ-Al₂O₃. (c) Sn K-edge XANES spectra and (d) the corresponding Sn K-edge EXAFS spectra of fresh and spent PtSn/γ-Al₂O₃.

Table R4. EXAFS fitting parameters at the Pt L₃-edge for spent PtSn/ γ -Al₂O₃.

Sample	Shell	N^a	$R(\text{\AA})^b$	$\sigma^2(\text{\AA}^2)^c$	$\Delta E_0(\text{eV})^d$	R factor
Fresh	Pt-O	0.3±0.1	1.95±0.01	0.0155±0.0052	3.9±2.0	0.0020
	Pt-Sn	1.6±0.2	2.70±0.02	0.0064±0.0030		
	Pt-Pt	4.3±0.4	2.74±0.01	0.0013±0.0012		
Spent	Pt-O	0.8±0.1	2.00±0.01	0.0030±0.0018	8.6±1.5	0.0023
	Pt-Sn	1.6±0.5	2.71±0.03	0.0064 ^e		
	Pt-Pt	3.8±0.8	2.77±0.01	0.0072±0.0018		

[a] N : coordination numbers.

[b] R : bond distance.

[c] σ^2 : Debye-Waller factors.

[d] ΔE_0 : the inner potential correction.

[e] The σ^2 value of the Pt-Sn path was fixed at 0.0064 for the fitting due to the following reason. We observed a marginal decline in the signal-to-noise ratio in the spectral data of the spent sample. To ensure the quality of fitting, we set the σ^2 of the Pt-Sn path to 0.0064, a value derived from reference to static fitting results. The rationale behind fixing the Pt-Sn path arose from our analysis of operando data, wherein we noticed a more pronounced peak intensity variation in the first shell of the sample's R space compared to the static condition. This variation was mainly attributed to Pt-O interaction. Furthermore, the coordination number of Pt-Pt has the highest overall proportion. As a result, rather than imposing constraints on Pt-O and Pt-Pt, we fixed the σ^2 specifically for the Pt-Sn path.

Table R5. EXAFS fitting parameters at the Sn K-edge for spent PtSn/ γ -Al₂O₃.

Sample	Shell	N^a	$R(\text{\AA})^b$	$\sigma^2(\text{\AA}^2)^c$	$\Delta E_0(\text{eV})^d$	R factor
Fresh	Sn-O	5.0±0.5	2.01±0.02	0.0045±0.0035	-0.3±0.4	0.0048
Spent	Sn-O	5.1±0.5	2.02±0.01	0.0054±0.0016	2.2±1.5	0.0049

[a] N : coordination numbers.

[b] R : bond distance.

[c] σ^2 : Debye-Waller factors.

[d] ΔE_0 : the inner potential correction.

“How was the TOF calculated?”

As suggested, we have provided the details concerning the calculation of TOF values in the revised manuscript. In the previous manuscript, we calculated the TOF values by assuming all Pt atoms as the active sites (eqs. R1 and R2).

$$\text{TOF} = \frac{\sum_i i \times n_i}{3 \times n_{\text{Pt}} \times t} \quad (\text{R1})$$

$$n_{\text{Pt}} = \frac{m_{\text{catal.}} \times w_{\text{Pt}}}{\mu_{\text{Pt}}} \quad (\text{R2})$$

In this equation, i refers to the carbon number. Taking C_3H_6O as an example, the i value is 3. n_i refers to the moles of the product with the carbon number of i . The value “3” in the denominator means that the carbon number of propane is 3. n_{Pt} is the moles of Pt atoms in the catalyst. t is the reaction time. $m_{catal.}$ refers to the mass of the catalyst. w_{Pt} refers to the mass loading of Pt in the catalyst. μ_{Pt} is the weight of one mole of Pt atoms. However, this calculation was not appropriate since not all the Pt atoms participated in the reaction. In the revised manuscript, we have calculated the TOF number by only counting the surface Pt atoms (eq. R3), because heterogeneous catalysis occurs on the surface of catalysts.

$$TOF = \frac{\sum_i i \times n_i}{3 \times n_{\text{surface Pt}} \times t} \quad (R3)$$

$$n_{\text{surface Pt}} = \frac{m_{\text{catal.}} \times n_{CO}}{50 \text{ mg}} \quad (R4)$$

The moles of surface Pt atoms ($n_{\text{surface Pt}}$) were measured according to CO pulse titration (eq. R4) where n_{CO} is the moles of adsorbed CO molecules. CO pulse titration was carried out as follows. Specifically, 50 mg of catalysts were added into the cell, followed by pulsing 10%CO with the quantitative loop volume of 450 μL at room temperature. According to Figure R10, the mole of total adsorbed CO molecules on the catalyst was calculated as 0.897 μmol (Table R6). We assumed that one Pt atom adsorbed one CO molecule. As such, 50 mg of catalysts contained 0.897 μmol of surface Pt atoms. Following the equation R3, the TOF number reached 41.3 h^{-1} . It was worth noting that the time zero point was counted when the system temperature reached the target temperature. However, the reaction had already occurred before the time zero point. When we calculated the TOF number, we subtracted the amount of products at the time zero point from the total yields at the reaction time. In the revised manuscript, we have replaced all the original TOF numbers with the revised ones based on surface Pt atoms and added relevant discussion (p. 6, lines 11-14, p. 16, lines 11-28, p. 17, lines 1-2, Supplementary Fig. 11, Supplementary Table 8, highlighted in yellow color).

Figure R10. CO pulse titration.

Table R6. Result of CO pulse titration.

CO	Unsaturated peak			Saturated peak		
	1st	2nd	3rd	4th	5th	6th
A_{Integral}	17.73667	27.64333	28.16417	28.73667	28.80667	28.87083
$A_{\text{Adsorption}}$	11.06805	1.161393	0.640553	/	/	/
$n_{\text{Adsorption}} (\mu\text{mol})$	0.771	0.081	0.045	/	/	/
$\Sigma n_{\text{Adsorption}} (\mu\text{mol})$	0.897			/		

REVIEWER COMMENTS

Reviewer #3 (Remarks to the Author):

The authors are thanked for the additional experiments and resulting data in response to my earlier comments. It is surprising to me the lack of difference between the operando XAS data and the ex-situ data.

The authors still need to address the following points:

The authors need to provide far greater detail as to how the operando XAS data were collected.

Their statement of "Wet propane was introduced into the cell through bubbling, while the cell was heated to 350 oC by the heater and protected by the cooling water, thereby monitoring the operando environment" would not allow the experiment to be reproduced by others. They must start with the amount of the catalyst pressed into the wafer and describe the experimental procedure.

The fitting of the Sn foil XAS makes no sense. It is not possible to fit two scattering paths as exactly the same distance.

The authors have replied: "The low coordination number does not imply the amorphous nature, since the coordination number lowers with the decreased size of nanoparticles. For accuracy, we have deleted this statement in the revised manuscript (p. 5, lines 4-5, highlighted in yellow color)", and yet I still see statements like: "the sum of which was also much lower than 12.0 and matched the amorphous nature of Pt clusters"

The authors state: "We can deduce that not all the Pt and Sn atoms were uniformly mixed to form PtSn bimetallic nanoparticles. Instead, a large proportion of Sn atoms were oxidized to form SnOx nanoparticles, leaving some Sn atoms coordinated with Pt atoms, consistent with the elemental mapping analysis.". The elemental mapping analysis does not, and can not, show Sn atoms coordinated to Pt atoms.

All the XAS data needs to be replotted:

The XANES spectra are normalized. The y-axis is thus unitless. The units are not "a.u."

The magnitude of the FT has units associated with it. The y-axis of the FT plots must show the units.

Reviewer #4 (Remarks to the Author):

In this work, the authors reported the synthesis of acetone using the PtSn/ γ -Al₂O₃ catalyst under high pressure of propane and water. The direct synthesis of acetone from propane and water (wet reforming of propane) is simple but interesting, having a high impact in this field of catalysis. The authors have conducted a lot of experiments to investigate the reaction mechanism for acetone production. The reviewer greatly respects for their efforts. However, there are some issues that need more clarification and reconsideration. The reviewer thinks that this paper is potentially worthy for publication in Nature Communications after addressing the suggested points.

Comment 1: As shown in Supplementary Figs 2 and 3, the mean particle size of PtSn/ γ -Al₂O₃ was larger than 1.5 nm. Therefore, the authors should use the term "nanoparticles" instead of "clusters".

Why the authors used the "clusters"? The authors should calculate the Pt dispersion from the CO pulse titration (Supplementary Figure 11). The Pt dispersion helps to better understand the size of Pt.

Comment 2: The authors should note the number of significant digits.

Comment 3: The reviewer is interested in the influence of Sn loading amount for acetone production. The reviewer strongly recommends the corresponding experiments.

Comment 4: What will happen when the physical mixture of Pt/ γ -Al₂O₃ and Sn/ γ -Al₂O₃ is used.

Comment 5: How about the stability? Please perform the stability test at least 5 cycles.

Comment 6: In the response to referees letter, the authors commented as following: "From Table R1, we can see that as the water content gradually increased from stoichiometric value to excess, the yield of acetone increased from 44.53 $\mu\text{mol g}^{-1}$ to 858.42 $\mu\text{mol g}^{-1}$, while the conversion also increased from 0.17% to 1.84%. In conclusion, the use of excess water enhanced heat and mass transfer processes, while the concentration diffusion of intermediates and product drives the reaction equilibrium". However, the reviewer is skeptical of this conclusion. The reviewer suggests the authors to perform the reaction with different conditions (for instance, molar ratio of water to propane is kept 1:1, the amount of catalyst: 1/10-fold, reaction time: 10-fold). In such condition, the contact between the catalyst and reactants can be increased, helping to understand the role of excess amount of water clearly.

Comment 7: With the increase in the reaction temperatures, CO₂ selectivity increased (Supplementary Figure 23). Please explain why the selectivity of CO₂ increases.

Comment 8: As shown in Supplementary Figure 19, the Pt LIII-edge XAFS spectrum of the spent PtSn/ γ -Al₂O₃ catalyst showed that the Pt species were oxidized during the reaction. What are the true active sites for this reaction, metallic Pt or oxidized Pt species?

Reviewer #5 (Remarks to the Author):

In this manuscript, the authors have shown the conversion of propane to acetone with PtSn/Al₂O₃ catalyst through a tandem process. While the process is of interest, there are some critical issues regarding to the characterizations of the catalysts and the interpretation of the catalytic results. I think the manuscript requires further major revisions before it can be accepted for publication in Nature Communications.

1. In Fig. 2f, the authors show the CO-IR spectra of the Pt/Al₂O₃ and PtSn/Al₂O₃ samples. It is unusual to observe more CO adsorbed on PtSn sites in the bridged configuration because normally, the formation of PtSn alloys will cause the preferential adsorption of CO on Pt sites in a linear

configuration. Such phenomena have been observed in numerous supported PtSn catalysts with similar structural features as those in the present manuscript. [Journal of the Chemical Society Faraday Transactions 93(20):3715. DOI:10.1039/a702174g] The authors need to clarify this issue because these results are contradictory to the claim that the PtSn/Al₂O₃ sample contains PtSn alloys.

2. The number of exposed Pt sites in the catalysts was quantified by CO chemisorption, but the results are not given. The authors should provide the percentage of the surface Pt sites derived from the CO chemisorption measurements. As mentioned above, the formation of PtSn alloys will suppress the adsorption of CO on Pt sites, thus making it difficult to obtain an accurate number of exposed surface sites. Actually, it is mentioned by the author that Pt/Al₂O₃ sample is much more active than the PtSn/Al₂O₃ for transforming propane into other molecules. But, why the TOF of Pt/Al₂O₃ is almost identical to PtSn/Al₂O₃? It is not reasonable to the referee.

3. In Supplementary Table 9, the improved conversion in the test with a lower partial pressure of propane is associated with the shift of the reaction by Le Chatelier's principle. However, the amount of the reactant is decreased by 25 times. In other words, the number of converted propane molecules are much less in the test under high-pressure conditions. The claim by the authors seem not make sense in terms of reaction kinetics. If the authors want to comment on this point, they should calculate the forward reaction rate based on the kinetic equations of the propane-to-acetone process.

4. I suggest the authors to carry out TG analysis of the spent catalysts to determine the coke contents in the Pt/Al₂O₃ and PtSn/Al₂O₃ catalysts, because it can help the authors to close the mass balance of carbon species.

5. The authors show the influence of the volume of water on the catalytic performance. The reviewer agrees with the authors that the amount of water will affect the yields of acetone. However, by varying the water volume in such a small range can cause marked differences is quite interesting. Could these phenomena be reproduced? In particular, it is unlikely that the presence of water can promote the propane dehydrogenation reaction. A plausible explanation for the experimental observation could be that, the addition of water can favour the hydration of propylene, which shifts the propane dehydrogenation reaction. I strongly suggest the authors to carefully re-considered the discussion in this section.

6. In Fig. 5b, the authors show that, a mixture of propene and water will be transformed into propane, which is surprising to the referee. How can this occur? What's the mechanism? There are no hydrogen in the reaction feed. How does the reverse hydrogenation reaction occur?

7. In Supplementary Fig. 30, the authors carry out operando XAS measurements. I suggest they move these data to the main text to replace the ex-situ XAS data shown in Fig. 2.

8. The direct conversion of alkanes into ketones is an interesting process. However, when comparing this process with the mature industrial process for production of oxygenates, the

authors need to consider the very low reaction rates of the direct process. In other words, an interesting reaction proceeds in a very low rate is not meaningful for chemical industry. The discussion and comparison of different processes should be made in a reasonable manner in order to avoid confusion/misleading to the readers.

Point-by-point response to reviewers' comments

Manuscript ID: NCOMMS-23-30241B

Title: Acetone synthesis from propane-wet-reforming process over PtSn nanoparticles on γ -Al₂O₃

Reviewer #3

“The authors are thanked for the additional experiments and resulting data in response to my earlier comments. It is surprising to me the lack of difference between the operando XAS data and the ex-situ data. The authors still need to address the following points:

The authors need to provide far greater detail as to how the operando XAS data were collected. Their statement of “Wet propane was introduced into the cell through bubbling, while the cell was heated to 350 °C by the heater and protected by the cooling water, thereby monitoring the operando environment” would not allow the experiment to be reproduced by others. They must start with the amount of the catalyst pressed into the wafer and describe the experimental procedure.”

As suggested by this reviewer, we have described the operando XAS tests in detail as follows.

The operando experiment were carried out using a homemade cell, which consists of reaction chamber, gas channels, cooling water channels, sample stage, Be windows and heater, as shown in **Supplementary Fig. 35a-e** (copied below). Firstly, 30 mg of sample was compressed into a disc with 10 mm in diameter under 2 MPa for 2 min as shown in **Supplementary Fig. 35f**. Then, the disc was sandwiched and locked between sample stage and the sample stage was carefully installed into the chamber. As shown in **Supplementary Fig. 35g-h**, the heater and K-type thermocouple were inserted into corresponding holes for external heating and measurement of the sample temperature, respectively. The control parameters can be set on the panel of console application including the target temperature, heating rate and so on, as displayed in **Supplementary Fig. 35i**. Subsequently, 1 bar of Ar flow with the rate of 20 mL min⁻¹ was allowed to flow into the reaction chamber through the gas channels for 0.5 h at room temperature to purge out the air from the inside. After that, the wet propane was introduced into the cell with the rate of 20 mL min⁻¹ at room temperature for 0.5 h through bubbling out of water half-full filled in a 20 ml-sized bottle. Meanwhile, the circulating water was flowed through the cooling water channels embedded in the chamber wall with the rate of 8 L min⁻¹ by a pump to protect the gas tightness of cell enclosure and windows from heat transfer. For security, the above equipment was checked by gas alarm device carefully to prevent gas leakage.

Before operando XAS tests, the X-ray was monochromatized by a double-crystal Si(111) and Si(311) monochromator, and the energy was calibrated using a Pt foil and a Sn foil for Pt L₃-edge and Sn K-edge, respectively, at the BL11B beamline of the Shanghai Synchrotron Radiation Facility (SSRF), China. Next, the cell was loaded and fixed on the lifting platform at precisely the right place to ensure that the incident X-ray can pass through the Be windows and sample to facilitate X-ray transmission, as displayed in **Supplementary Fig. 35j**. With everything prepared, the cell was heated to 350 °C at a ramping rate of 20 °C min⁻¹ by the heater, together with the stream of wet propane, thereby providing the operando environment. Finally, the XAS spectra were acquired in transmission mode. We have added the above detail of operando XAS test in the **Methods** section of the revised manuscript (p. 19, lines 513-524, p. 20, lines 525-537, Supplementary Fig. 35, highlighted in yellow color).

Supplementary Fig. 35. Photograph of the setup for the operando XAS test. (a) Front view and (b) top view of cell. (c) Top view of chamber of cell. (d) Locked and (e) unlocked sample stage. (f) Sample compressed into a disc with 10 mm in diameter under 2 MPa for 2 min. (g) Top view of cell connected with the heater and K-type thermocouple. (h) Back view and (i) front view of console application for external heating and measurement of the sample temperature. (j) Front view of installed setup for the operando XAS test, consists of reaction cell, gas channels for wet propane, cooling water channels, Be windows and heater control system.

“The fitting of the Sn foil XAS makes no sense. It is not possible to fit two scattering paths as exactly the same distance.”

We thank the reviewer for pointing out this issue. As suggested, we have removed the fitting of the Sn foil XAS in the revised manuscript (p. S12, Supplementary Table 3, highlighted in yellow color).

“The authors have replied: “The low coordination number does not imply the amorphous nature, since the coordination number lowers with the decreased size of nanoparticles. For accuracy, we have deleted this statement in the revised manuscript (p. 5, lines 4-5, highlighted in yellow color)”, and yet I still see statements like: “the sum of which was also much lower than 12.0 and matched the amorphous nature of Pt clusters””

We sincerely apologize for this mistake. We have checked through the revised manuscript and removed related discussion involving amorphous structure.

“The authors state: “We can deduce that not all the Pt and Sn atoms were uniformly mixed to form PtSn bimetallic nanoparticles. Instead, a large proportion of Sn atoms were oxidized to form SnO_x nanoparticles, leaving some Sn atoms coordinated with Pt atoms, consistent with the elemental mapping analysis”. The elemental mapping analysis does not, and cannot, show Sn atoms coordinated to Pt atoms.”

We thank the reviewer for his/her valuable comment. We agree with this reviewer’s comment that the elemental mapping cannot reflect the alloy nature. We removed this statement in the revised manuscript.

“All the XAS data needs to be replotted:

*The XANES spectra are normalized. The y-axis is thus unitless. The units are not “a.u.”
The magnitude of the FT has units associated with it. The y-axis of the FT plots must show the
units.”*

We thank the reviewer for pointing out this issue. As suggested, we have replotted all the XAS data with corrected units of y-axis in the revised manuscript (Figs. 2d, 2e, Supplementary Figs. 6, 7, 8, 22, highlighted in yellow color).

Reviewer #4

“In this work, the authors reported the synthesis of acetone using the PtSn/ γ -Al₂O₃ catalyst under high pressure of propane and water. The direct synthesis of acetone from propane and water (wet reforming of propane) is simple but interesting, having a high impact in this field of catalysis. The authors have conducted a lot of experiments to investigate the reaction mechanism for acetone production. The reviewer greatly respects for their efforts. However, there are some issues that need more clarification and reconsideration. The reviewer thinks that this paper is potentially worthy for publication in Nature Communications after addressing the suggested points.”

We appreciate the reviewer's positive feedback on our work. We have performed additional experiments and provided detailed discussions to address the comments as follows.

“Comment 1: As shown in Supplementary Figs 2 and 3, the mean particle size of PtSn/ γ -Al₂O₃ was larger than 1.5 nm. Therefore, the authors should use the term "nanoparticles" instead of "clusters". Why the authors used the "clusters"? The authors should calculate the Pt dispersion from the CO pulse titration (Supplementary Figure 11). The Pt dispersion helps to better understand the size of Pt.”

We thank the reviewer for his/her valuable comments. We have to admit that it is still under debate to achieve an absolute definition of “cluster”. Generally, cluster is defined as an ensemble of the metal atoms with limited size smaller than 2 nm [*Acc. Chem. Res.* 50, 8, 1894–1901 (2017); *Acc. Chem. Res.* 47, 816–824 (2014); *Nat. Catal.* 5, 485–493 (2022); *Nat. Mater.* 11, 49–52 (2012); *Nat. Nanotechnol.* 10, 577–588 (2015); *Science* 327, 850–853 (2010)]. In this work, the mean particle size measured for Pt and PtSn are 1.31 and 1.74 nm, therefore we referred them with the “cluster” word. At the same time, clusters with size of 1–2 nm could be regarded as very small nanoparticles as well [*ACS Catal.* 2020, 10, 11011–11045; *Nat. Catal.* 1, 540–546 (2018); *Acc. Chem. Res.* 46, 1682–1691 (2013)]. To avoid unwanted controversy, we have replaced the term “clusters” with “nanoparticles” in the revised manuscript.

As requested, we calculated the dispersion of Pt in PtSn/ γ -Al₂O₃. The details are described as follows. Using 50 mg of catalysts with a Pt mass loading of 2.71 wt% for CO pulse titration, the moles of total Pt atoms were calculated as 6.949 μ mol. The moles of total adsorbed CO molecules on the catalyst were calculated as 0.897 μ mol (**Supplementary Fig. 11, Supplementary Table 8**, copied below). We assumed that one Pt atom adsorbs one CO molecule. As such, the dispersion of Pt in PtSn/ γ -Al₂O₃ catalyst was thus calculated as $0.897 \mu\text{mol} \div 6.949 \mu\text{mol} \times 100\% = 12.9\%$. Meanwhile, the dispersion of Pt in Pt/ γ -Al₂O₃ catalyst was thus calculated as $4.281 \mu\text{mol} \div 7.282 \mu\text{mol} \times 100\% = 58.8\%$.

Supplementary Figure 11 | CO pulse titration over PtSn/ γ -Al₂O₃ (a) and Pt/ γ -Al₂O₃ (b).

Supplementary Table 8 | Result of CO pulse titration over PtSn/ γ -Al₂O₃ and Pt/ γ -Al₂O₃.

CO pulse titration		Unsaturated peak				Saturated peak		
		1st	2nd	3rd	4th	5th	6th	7th
PtSn/ γ - Al ₂ O ₃	A_{Integral}	17.74	27.64	28.16	28.74	28.81	28.87	28.79
	$A_{\text{Adsorption}}$	11.07	1.16	0.64	/	/	/	/
	$n_{\text{Adsorption}} (\mu\text{mol})$	0.77	0.08	0.05	/	/	/	/
	$\Sigma n_{\text{Adsorption}} (\mu\text{mol})$	0.90						
Pt/ γ - Al ₂ O ₃	A_{Integral}	10.97	13.53	15.43	20.51	26.86	29.84	30.29
	$A_{\text{Adsorption}}$	19.32	16.75	14.86	9.78	3.43	0.45	/
	$n_{\text{Adsorption}} (\mu\text{mol})$	1.28	1.11	0.98	0.65	0.23	0.03	/
	$\Sigma n_{\text{Adsorption}} (\mu\text{mol})$	4.28						

We further took an investigation of the relation between nanoparticle size and dispersion to help understand the structural feature of the catalysts. Under the assumption of PtSn as well-defined spherical nanoparticles, we introduced a parameter, P_s , to represent the packing fraction on a surface crystalline plane (the ratio of the plane area occupied by atoms to the total plane area). For the (111) close-packed plane of a face-centered cubic (*fcc*) structure, the area 'covered' by an atom of radius r equals its cross-section (πr^2). The total surface area associated with each atom is $2\sqrt{3}r^2$. Therefore, $P_{S(111),fcc} = \pi r^2 / (2\sqrt{3}r^2) = 0.91$.

The number of surface atoms, n_s , is calculated as follows:

$$n_s = P_s \frac{4\pi R^2}{\pi r^2} = 4P_s \frac{R^2}{r^2} \quad (\text{eq. R1})$$

where R is the radius of the spherical nanoparticle, and r is the atomic radius deduced from the atomic volume ($V_a = 4\pi r^3/3$).

Next, a lattice packing fraction, P_L , is introduced, representing the ratio of the volume of the crystal

occupied by atoms to the total volume. Using the example of an *fcc* lattice, the volume of a single atom is $4\pi r^3/3$, and the total volume associated with each atom in the unit cell is $4\sqrt{2}r^3$. Therefore $P_{L, fcc} = (4\pi r^3/3)(4\sqrt{2}r^3) = 0.74$.

To calculate the total number of atoms in the nanoparticle, considering the outer surface composed of many crystalline planes, the volume of the nanoparticle occupied by atoms can be calculated as:

$$n_i V_a + \frac{1}{2} n_s V_a = P_L V_p \quad (\text{eq. R2})$$

$$n_t = n_i + n_s \quad (\text{eq. R3})$$

where V_a is the atomic volume, and V_p is the volume of the particle. Thus,

$$n_t = P_L \frac{R^3}{r^3} + 2P_s \frac{R^2}{r^2} \quad (\text{eq. R4})$$

Assuming a surface atomic ratio of Pt/Sn as 1:1 and the metals share a same atomic radius of $r_{Pt} = 0.14$ nm, the dispersion of Pt can be calculated as:

$$dispersion_{PtSn \text{ nanoparticle}} = \frac{1}{2} \times \frac{n_s}{n_t} = \frac{2P_s}{P_L \frac{R}{r} + 2P_s} = \frac{1}{1.094 \times R + 1} \quad (\text{eq. R5})$$

$$dispersion_{Pt \text{ nanoparticle}} = \frac{n_s}{n_t} = \frac{4P_s}{P_L \frac{R}{r} + 2P_s} = \frac{2}{1.094 \times R + 1} \quad (\text{eq. R6})$$

To this end, a clear relationship between dispersion and nanoparticle size can be illustrated in **Supplementary Figure N1** (copied below).

Supplementary Figure N1 | The relationship between the nanoparticle size and Pt dispersion obtained by modeling. **(a)** PtSn nanoparticle, **(b)** Pt nanoparticle.

As illustrated, the increase of the nanoparticle size induces the significant decrease of the Pt dispersion. With an average nanoparticle size (R) of 1.74 nm for PtSn nanoparticles and 1.31 nm for Pt nanoparticles (**Supplementary Fig. 2**), we can deduce a theoretical Pt dispersion of 34.4% for PtSn nanoparticles and 82.2% for Pt nanoparticles, based on **equation 5** and **6**, respectively. Of note, these values were apparently larger than the value (12.9% and 58.8%) based on CO pulse titration. The difference can be attributed to the imperfectly spherical morphology of the nanoparticles, and more importantly, the close intact of the nanoparticles to the γ -Al₂O₃ support, which hinders part of the surface atoms from adsorbing CO molecules. We also exclude the possibility of the deviation of Pt/Sn ratio to 1:1 that caused the decrease, since the existence of isolated SnO_x nanoparticles indicated a higher Pt/Sn ratio in the surface which leads to a higher Pt dispersion. For comparison, previously reported Pt/Al₂O₃ catalyst synthesized *via* traditional incipient-wetness impregnation method exhibited a Pt dispersion of 70~85% with Pt nanoparticle size of 1.2~1.4 nm, which is very close to the calculated 82.2% value based on the above model [*ACS Catal.* 2020, 10, 12932-12942; *J. Am. Chem. Soc.* 2011, 133, 4498–4517]. Therefore, we ascribed the relatively low dispersion value mainly to the close intact between nanoparticles and support, which provides abundant interfaces for successive catalytic processes and ensures a good stability.

We have added related data and discussions in the revised manuscript (p. 16, lines 422-431, Supplementary Figure 11, Supplementary Table 8, Ref. S7 and S8, highlighted in yellow color).

“Comment 2: The authors should note the number of significant digits.”

As requested, we have carefully checked the number of significant digits.

“Comment 3: The reviewer is interested in the influence of Sn loading amount for acetone production. The reviewer strongly recommends the corresponding experiments.”

We thank the reviewer for his/her helpful advice. As suggested, we have investigated the influence of Sn loading in the revised manuscript. We prepared PtSn/ γ -Al₂O₃ catalysts with different Pt:Sn ratios ranging from 5:1 to 0.4:1. As shown in **Supplementary Figure 14** (copied below), the highest acetone yield was achieved at the Pt:Sn ratio of 1:1. With increased Sn loading, the selectivity for total C₃ products including acetone, isopropanol, and C₃H₆ increased. As such, the loading of Sn element in PtSn/ γ -Al₂O₃ catalysts mitigates the cracking of C-C bonds. We also noticed that increasing the Sn loadings led to a decreased productivity, which can be ascribed to the reduced exposed Pt active sites. We have added related content to the revised manuscript (p. 6, lines 123-126, Supplementary Figure 14).

Supplementary Figure 14 | Dependence of catalytic properties of PtSn/ γ -Al₂O₃ catalysts on Pt/Sn ratios.

“Comment 4: What will happen when the physical mixture of Pt/ γ -Al₂O₃ and Sn/ γ -Al₂O₃ is used.”

We thank the reviewer for his/her insightful comment. As suggested, we explored the catalytic properties of the physical mixture of Pt/ γ -Al₂O₃ and Sn/ γ -Al₂O₃ (denoted as Pt+Sn/ γ -Al₂O₃) with the same metal amount. As shown in **Supplementary Figure 16** (copied below), we found that Pt+Sn/ γ -Al₂O₃ behaved similarly to Pt/ γ -Al₂O₃, mainly producing the cracked products (e.g., C₂H₆, CH₄, and CO₂) and much differently from PtSn/ γ -Al₂O₃. This phenomenon indicates the necessity of alloying Pt and Sn for acetone production. The dilution of Pt atoms with Sn atoms significantly mitigates the cracking process. We have added related content to the revised manuscript (p. 6, lines 133-135, Supplementary Figure 16).

Supplementary Figure 16 | Comparison in catalytic properties of Pt+Sn/ γ -Al₂O₃, Pt/ γ -Al₂O₃, and PtSn/ γ -Al₂O₃.

“Comment 5: How about the stability? Please perform the stability test at least 5 cycles.”

We thank the reviewer for his/her important question. As suggested, we performed the stability test over PtSn/ γ -Al₂O₃ catalyst for 5 cycles. After 5 cycles, we observed a slight decrease by 14.0% of the acetone yield, from 1012.0 to 870.6 $\mu\text{mol g}^{-1}$ (**Supplementary Figure 23**, copied below).

The decreased activity is possibly originated from the oxidation of metallic Pt species in the fresh sample since Pt species were oxidized in the spent sample. We also verified this point by evaluating the catalytic performance over intentionally oxidized PtSn/ γ -Al₂O₃, a more detailed discussion can be found in the response to **Comment 8**. We have added related content to the revised manuscript (p. 7, lines 157-162, Supplementary Figure 23 and 24).

Supplementary Figure 23 | Stability tests over PtSn/ γ -Al₂O₃ for different cycles.

“Comment 6: In the response to referees’ letter, the authors commented as following: “From Table R1, we can see that as the water content gradually increased from stoichiometric value to excess, the yield of acetone increased from 44.53 $\mu\text{mol g}^{-1}$ to 858.42 $\mu\text{mol g}^{-1}$, while the conversion also increased from 0.17% to 1.84%. In conclusion, the use of excess water enhanced heat and mass transfer processes, while the concentration diffusion of intermediates and product drives the reaction equilibrium”. However, the reviewer is skeptical of this conclusion. The reviewer suggests the authors to perform the reaction with different conditions (for instance, molar ratio of water to propane is kept 1:1, the amount of catalyst: 1/10-fold, reaction time: 10-fold). In such condition, the contact between the catalyst and reactants can be increased, helping to understand the role of excess amount of water clearly.”

We genuinely thank this reviewer’s constructive suggestion. As suggested by this reviewer, we decreased the amount of PtSn/ γ -Al₂O₃ to 2.5 mg (1/10 of the original amount) and extended the reaction time to 20 h (10-fold). It is expected that under this condition, the contact between the catalyst and reactants is increased. However, a decreased reaction rate (Yield divide by reaction time) and the selectivity for acetone was observed (**Supplementary Figure 31**, copied below). We suspect that this phenomenon is caused by the over-reduced amount of catalyst, which decreased collision probability between the reactant and catalyst. We are afraid that the change in reaction conditions introduces a new variate, the amount of catalyst, which would interfere with the study of water ratio in this work. Actually, the dependence of catalytic performance and the amount of catalyst is also revealed by previously reported works, and the influence can be ignored after exceeding a certain quantity [*Angew. Chem. Int. Ed.* 55, 737-741 (2016)]. We think that the observation of massive production of post-reacted products (eg. acetone and ethane) evidently showed sufficient contact between reactant or intermediates and the catalysts, thus the influence of the amount of catalyst can be ignored. The influence of water ratio is fairly reflected.

Nevertheless, the recommended experiments helped us to reveal propene as an important intermediate for acetone formation. Of note, there are also several reports that suggest a co-feeding of water steam promotes propane dehydrogenation in increasing reaction rate and decreasing

apparent activation energy [*Catal. Sci. Technol.* 2015, 5, 3991-4000; *Chem. Eng. J.* 2015, 278, 240-248; *J. Phys. Chem. C* 2021, 125, 5623-5634]. Together with the improved yield of acetone, we suspect that both propane dehydrogenation and possible sequential propene hydration process are promoted by water. To conclude, as both reactant and solvent, the excess amount of water not only facilitates the dehydrogenation and promotes the hydration process, but also helps to reduce the product concentration on the catalyst surface. We have added related content to the revised manuscript (p. 9, lines 226-228, p.10, lines 229-240, Supplementary Figure 31).

Supplementary Figure 31 | Exploration of the contact between the catalyst and reactants.

“Comment 7: With the increase in the reaction temperatures, CO₂ selectivity increased (Supplementary Figure 23). Please explain why the selectivity of CO₂ increases.”

We genuinely thank this reviewer for raising this issue. We suspect that CO₂ molecules were mainly produced from steam reforming reactions which are highly endothermic, as shown in **Supplementary Table 14** (copied below) [*Int. J. Hydrogen Energy* 31, 13-19 (2006); *Appl. Catal. A: General* 332, 310-317 (2007)]. These reactions are favored in high reaction temperatures and lead to the enhanced production of CO₂ and cracking products. We have added related content to the revised manuscript (p. 8, lines 175-179, Supplementary Table 14).

Supplementary Table 14. Typical reactions for CO₂ production and C-C cracking.

Reaction		ΔH^\ominus (kJ/mol)
$C_3H_6O + 5H_2O \rightarrow 3CO_2 + 8H_2$	steam reforming of acetone	+244
$C_3H_6 + 6H_2O \rightarrow 3CO_2 + 9H_2$	steam reforming of propene	+251
$C_3H_8 + 6H_2O \rightarrow 3CO_2 + 10H_2$	steam reforming of propane	+375
$C_2H_6 + 6H_2O \rightarrow 3CO_2 + 9H_2$	steam reforming of ethane	+251
$C_3H_8 \rightarrow C_2H_6 + H_2 + C$ (coke)	cracking	+19
$C_2H_6 \rightarrow CH_4 + H_2 + C$ (coke)	cracking	+9

“Comment 8: As shown in Supplementary Figure 19, the Pt LIII-edge XAFS spectrum of the spent

PtSn/γ-Al₂O₃ catalyst showed that the Pt species were oxidized during the reaction. What are the true active sites for this reaction, metallic Pt or oxidized Pt species?"

We genuinely thank this reviewer for his/her constructive comments. To explore the true active sites, we have intentionally prepared oxidized Pt species for this reaction in the revised manuscript. We calcined PtSn/γ-Al₂O₃ in air at 750 °C to oxidize Pt species (denoted as ox-PtSn/γ-Al₂O₃). As shown in **Supplementary Figure 24** (copied below), CO-probed IR spectrum of ox-PtSn/γ-Al₂O₃ only showed the peaks for oxidized Pt species at 2095 cm⁻¹, with no peaks for metallic Pt species. Then, ox-PtSn/γ-Al₂O₃ showed much poorer performance in terms of product yields compared with PtSn/γ-Al₂O₃. As such, metallic Pt was determined as the active sites. The presence of oxidized Pt species in the spent sample accounted for the decreased activity observed after stability tests (**Supplementary Figure 23**, comment 5). We have added related content to the revised manuscript (p. 7, lines 157-162, Supplementary Figure 23 and 24).

Supplementary Figure 24 | Comparison of the catalytic performance between PtSn/γ-Al₂O₃ and deliberately oxidized PtSn/γ-Al₂O₃ (denoted as ox-PtSn/γ-Al₂O₃). (b) CO-DRIFTS spectra of ox-PtSn/γ-Al₂O₃.

Reviewer #5

“In this manuscript, the authors have shown the conversion of propane to acetone with PtSn/Al₂O₃ catalyst through a tandem process. While the process is of interest, there are some critical issues regarding to the characterizations of the catalysts and the interpretation of the catalytic results. I think the manuscript requires further major revisions before it can be accepted for publication in Nature Communications.”

We sincerely appreciate the reviewer’s valuable comments and the approval of the novelty of our work. We have clarified this reviewer’s specific questions as follows.

“1. In Fig. 2f, the authors show the CO-IR spectra of the Pt/Al₂O₃ and PtSn/Al₂O₃ samples. It is unusual to observe more CO adsorbed on PtSn sites in the bridged configuration because normally, the formation of PtSn alloys will cause the preferential adsorption of CO on Pt sites in a linear configuration. Such phenomena have been observed in numerous supported PtSn catalysts with similar structural features as those in the present manuscript. [Journal of the Chemical Society Faraday Transactions 93(20):3715. DOI:10.1039/a702174g] The authors need to clarify this issue because these results are contradictory to the claim that the PtSn/Al₂O₃ sample contains PtSn alloys.”

We sincerely thank the review for his/her critical comment. Indeed, CO-IR spectra in **Fig. 2f** (copied below) reveals that CO is adsorbed more on the PtSn sites in the linear configuration rather than in the bridged configuration. As shown in **Fig. 2f**, for PtSn/ γ -Al₂O₃, the peaks at 2077, 2060, and 2036 cm⁻¹ were assigned to the linear adsorption of CO molecules on Pt sites (CO_{linear}), while that at 1830 cm⁻¹ corresponded to the bridged configuration (CO_{bridge}). Based on the peak area, the ratio of CO_{linear} to CO_{bridge} was estimated as 18.5 for PtSn/ γ -Al₂O₃. For Pt/ γ -Al₂O₃, the peaks at 2082, 2064, and 2036 cm⁻¹ were assigned to CO_{linear}, while that at 1820 cm⁻¹ corresponded to CO_{bridge}. Based on the peak area, the ratio of CO_{linear} to CO_{bridge} was estimated as 3.0 for Pt/ γ -Al₂O₃. According to the above analysis, the ratio (18.5) of CO_{linear} to CO_{bridge} for PtSn/ γ -Al₂O₃ was higher than that (3.0) for Pt/ γ -Al₂O₃. This indicates that PtSn/ γ -Al₂O₃ has fewer Pt-Pt ensembles compared to Pt/ γ -Al₂O₃, likely due to the dilution effect of Sn atoms, as CO adsorption on bridge sites is a characteristic feature of Pt-Pt ensembles. Therefore, these results support the presence of PtSn alloys in PtSn/ γ -Al₂O₃.

Fig. 2f CO-probe DRIFTS spectra of PtSn/ γ -Al₂O₃ and Pt/ γ -Al₂O₃.

As indicated by the reviewer, the normally observed preferential adsorption of CO on the Pt sites are generally found in PtSn alloys with a relatively high Sn content (atomic percentage $\geq 50\%$). For example, bridged adsorbed CO was observed in PtSn/ Al_2O_3 catalysts with an atomic Pt:Sn ratio of 1:1 and was negligible at ratios of 1:2, 1:4, and 1:6 [Chem. Eng. J. 2022, 443, 136393]. Bridged adsorbed CO was also observed in PtSn alloys with relatively high Pt content (atomic percentage $\geq 50\%$) [J. Catal. 2017, 356, 307-314].

Based on the XAFS analysis, we deduced that not all the Pt and Sn atoms were uniformly mixed to form PtSn bimetallic nanoparticles. Instead, a certain proportion of Sn atoms were oxidized to form SnO_x nanoparticles. Therefore, the actual atomic ratio of Pt:Sn is larger than 1:1 for PtSn nanoparticles synthesized in this work, which rationalized the observation of bridge adsorbed CO.

We have added related discussions to the revised manuscript (p. 17, lines 450-457).

“2. The number of exposed Pt sites in the catalysts was quantified by CO chemisorption, but the results are not given. The authors should provide the percentage of the surface Pt sites derived from the CO chemisorption measurements. As mentioned above, the formation of PtSn alloys will suppress the adsorption of CO on Pt sites, thus making it difficult to obtain an accurate number of exposed surface sites. Actually, it is mentioned by the author that Pt/ Al_2O_3 sample is much more active than the PtSn/ Al_2O_3 for transforming propane into other molecules. But, why the TOF of Pt/ Al_2O_3 is almost identical to PtSn/ Al_2O_3 ? It is not reasonable to the referee.”

We sincerely thank the review for his/her valuable comment. Although the formation of PtSn alloy weakened the adsorption of CO on Pt sites, the influence is acceptable for the quantification of surface atom *via* CO-pulse chemisorption methodology, which is widely adopted [Nat. Catal. 2020, 3, 628-638; Nat. Mater. 2019, 18, 866-873; Nat. Commun. 2022, 13, 5065; ...]. Moreover, the alloy of Sn significantly decreased the proportion of bridged CO species, enabling a more accurate estimation since the stoichiometry of Pt/CO is assumed to be 1 during the calculation.

As requested by the reviewer, we calculated the dispersion of Pt in PtSn/ $\gamma\text{-Al}_2\text{O}_3$. Using 50 mg of catalyst with a Pt mass loading of 2.71 wt% for CO pulse titration, the total moles of Pt atoms were calculated as 6.95 μmol . According to **Supplementary Fig. 11** (copied below), the total moles of adsorbed CO molecules on the catalyst were determined to be 0.90 μmol (**Supplementary Table 8**, copied below). Assuming that one Pt atom adsorbs one CO molecule, the 50 mg of catalyst contained 0.90 μmol of surface Pt atoms. Therefore, the dispersion of Pt was calculated as 12.9%.

Supplementary Figure 11 | CO pulse titration over PtSn/ $\gamma\text{-Al}_2\text{O}_3$ (a) and Pt/ $\gamma\text{-Al}_2\text{O}_3$ (b).

Supplementary Table 8 | Result of CO pulse titration over PtSn/ γ -Al₂O₃ and Pt/ γ -Al₂O₃.

CO pulse titration		Unsaturated peak			Saturated peak			
		1st	2nd	3rd	4th	5th	6th	7th
PtSn/ γ - Al ₂ O ₃	A _{Integral}	17.74	27.64	28.16	28.74	28.81	28.87	28.79
	A _{Adsorption}	11.07	1.16	0.64	/	/	/	/
	n _{Adsorption} (μ mol)	0.77	0.08	0.05	/	/	/	/
	Σ n _{Adsorption} (μ mol)	0.90						
Pt/ γ - Al ₂ O ₃	A _{Integral}	10.97	13.53	15.43	20.51	26.86	29.84	30.29
	A _{Adsorption}	19.32	16.75	14.86	9.78	3.43	0.45	/
	n _{Adsorption} (μ mol)	1.28	1.11	0.98	0.65	0.23	0.03	/
	Σ n _{Adsorption} (μ mol)	4.28						

We apologize for the improper calculation before, where we calculated the TOF (turnover frequency) number of PtSn/ γ -Al₂O₃ based on surface Pt atoms but calculated the TOF number of Pt/ γ -Al₂O₃ based on total Pt atoms. This led to an underestimation of the TOF number for Pt/ γ -Al₂O₃. In the revised manuscript, we have conducted CO titration over Pt/ γ -Al₂O₃ and calculated the amount of surface Pt atoms. Using 50 mg of catalyst with a Pt mass loading of 2.84 wt% for CO pulse titration, the total moles of Pt atoms were calculated as 7.28 μ mol. According to **Supplementary Fig. 11**, the total moles of adsorbed CO molecules on the catalyst were determined to be 4.28 μ mol (**Supplementary Table 8**). Therefore, the dispersion of Pt was calculated as 58.8%. Consequently, the TOF number of Pt/ γ -Al₂O₃, based on surface Pt atoms, was determined to be 70.7 h⁻¹, which is higher than that (41.3 h⁻¹) of PtSn/ γ -Al₂O₃. This finding is consistent with the higher activity of Pt/ γ -Al₂O₃ compared to PtSn/ γ -Al₂O₃.

We have added related discussions to the revised manuscript (p.6, lines 111-112, p. 16, lines 422-431, Supplementary Fig. 11, Supplementary Table 8).

“3. In Supplementary Table 9, the improved conversion in the test with a lower partial pressure of propane is associated with the shift of the reaction by Le Chatelier’s principle. However, the amount of the reactant is decreased by 25 times. In other words, the number of converted propane molecules are much less in the test under high-pressure conditions. The claim by the authors seem not make sense in terms of reaction kinetics. If the authors want to comment on this point, they should calculate the forward reaction rate based on the kinetic equations of the propane-to-acetone process.”

We sincerely thank the review for his/her critical comment. We are sorry for the misleading caused by improper expression and data presentation. As stated by the reviewer, it is true that increasing the amount of the reactants (here the partial pressure of propane) would promote the reaction rate and more products are expected under high-pressure conditions. We have verified this point by updating **Supplementary Table 9** (copied below). More propane molecules are converted at high-pressure conditions (37.13 μ mol for 5 bar, 32.57 μ mol for 3 bar, and 20.67 μ mol for 0.2 bar). However, the remained large amount of unreacted propane molecules make the conversion at high-pressure conditions significantly lower.

Supplementary Table 9. Comparison of propane conversion. Comparison between the propane conversion during wet reforming over PtSn/ γ -Al₂O₃ and the kinetic analysis.

Reaction conditions	Propane conversion	Moles of converted propane molecules	Apparent reaction rate constant (k)	Apparent reaction rate (R)
5 bar of C ₃ H ₈ and 1 bar of N ₂ , 350 °C	1.84% (2 h)	37.13 μ mol	0.007 bar ^{-0.12} h ⁻¹	0.141 bar h ⁻¹
3 bar of C ₃ H ₈ and 1 bar of N ₂ , 350 °C	2.69% (2 h)	32.57 μ mol	0.011 bar ^{-0.12} h ⁻¹	0.126 bar h ⁻¹
0.2 bar of C ₃ H ₈ and 0.8 bar of N ₂ , 350 °C	25.60% (2 h)	20.67 μ mol	0.161 bar ^{-0.12} h ⁻¹	0.087 bar h ⁻¹

As suggested by this reviewer, we have also calculated the forward reaction rate based on the kinetic equations of the propane-to-acetone process. **Figure 3c** indicates that the apparent reaction order for propane steam reforming to produce acetone is 1.12. From the apparent reaction order, the reaction kinetic rate equation can be written as:

$$R = -\frac{dp_{C_3H_8}}{dt} = k \cdot p_{C_3H_8}^{1.12} \quad (\text{eq. R7})$$

$$-\frac{dp_{C_3H_8}}{p_{C_3H_8}^{1.12}} = k dt \quad (\text{eq. R8})$$

$$\frac{1}{0.12 p_{C_3H_8}^{0.12}} = kt + M \quad (\text{eq. R9})$$

where R is the apparent reaction rate, $p_{C_3H_8}$ is the partial pressure of propane at the certain time, t is the reaction time, k is the apparent reaction rate constant, and M is the integral constant.

At $t = 0$, the initial pressures under three reaction conditions are 14.6 bar, 8.8 bar, and 0.58 bar, respectively, based on the Charles' law, $P = P_0(1 + T/273)$. By substituting these initial values into equation R9, the integral constant terms M under these three conditions are found to be 6.04, 6.42, and 8.89.

At $t = 2$ h, the final state corresponding to the conversion rate of C₃H₈ from **Supplementary Table 9** is calculated for the respective conditions. These values, along with the integral constant terms M mentioned above, are substituted into equation R9 to determine the corresponding apparent reaction rate constant, k , and apparent reaction rate, R , as shown in **Supplementary Table 9**.

As shown in **Supplementary Table 9**, the apparent reaction rates increased with the elevated partial pressure of propane, which follows the reaction kinetics. The previously calculated equilibrium conversions we present exhibited an increasing trend from high- to low-pressure conditions, which follows the *Le Chatelier's principle*. This result indicated the possibility of achieving a higher conversion at a lower pressure condition as we experimentally observed. We removed the statement about *Le Chatelier's principle* that is potentially misleading in the main text.

We have added related discussions to the revised manuscript (p. 6, lines 113-115, Supplementary Table 9).

“4. I suggest the authors to carry out TG analysis of the spent catalysts to determine the coke contents in the Pt/Al₂O₃ and PtSn/Al₂O₃ catalysts, because it can help the authors to close the mass balance of carbon species.”

We sincerely thank the review for his/her valuable suggestions. As suggested, we have carried out TG analysis of the spent catalysts. The weight losses of spent PtSn/γ-Al₂O₃ and spent Pt/γ-Al₂O₃ were 4.14% and 5.99%, respectively (**Supplementary Figure 18**, copied below).

Supplementary Figure 18 | TG profiles of spent PtSn/γ-Al₂O₃ and spent Pt/γ-Al₂O₃.

Assuming that the coke exists in the form of C₆H₆ (M_w=78), we can calculate the amount of consumed propane for coke formation as:

$$25 \text{ mg} \times 4.14\% \div 78 \text{ g/mol} \times 2 = 26.5 \text{ } \mu\text{mol} \text{ (For PtSn/}\gamma\text{-Al}_2\text{O}_3\text{)}$$

$$25 \text{ mg} \times 5.99\% \div 78 \text{ g/mol} \times 2 = 38.4 \text{ } \mu\text{mol} \text{ (For Pt/}\gamma\text{-Al}_2\text{O}_3\text{)}$$

Then, we calculated the carbon balance based on a modified equation (**eq. R10**) suggested for reactions with low conversions [Joule 2019, 3, 2876-2883]. The results were presented in **Supplementary Table 11** (copied below).

$$\text{Modified carbon balance} = \frac{\sum n_{\text{produced}}}{\sum n_{\text{consumed}}} \times 100\% \quad (\text{eq. R10})$$

Supplementary Table 11. Mass balance of carbon species for PtSn/ γ -Al₂O₃ and Pt/ γ -Al₂O₃.

Sample	Total Consumed propane derived from GC analysis before and after reaction	Consumed propane derived from produced molecules via GC, ¹ H NMR analysis	Consumed propane derived from produced Coke via TG analysis	Modified carbon balance
PtSn/ γ -Al ₂ O ₃	99.0 μ mol	37.1 μ mol	26.5 μ mol	64.3%
Pt/ γ -Al ₂ O ₃	256.9 μ mol	131.8 μ mol	38.4 μ mol	66.3%

Of note, the obtained carbon balance for PtSn/ γ -Al₂O₃ and Pt/ γ -Al₂O₃ are 64.3% and 66.3%, respectively, which are apparently smaller than the ideal 100% value. Since the quantification methodology including GC and ¹H NMR analysis in this study strictly calibrated by standard curves. We suspect that the main reason for the underestimated carbon balance to the exfoliation of coke species under the harsh hydrothermal conditions. As previously reported, aliphatic coke species are preferred to be formed in PtSn/Al₂O₃ while polyaromatics are preferred to be formed in Pt/Al₂O₃ [*Ind. Eng. Chem. Res.* 2018, 57, 8647-8654]. The dilution of the adjacent Pt atoms by Sn incorporation provides even less anchoring site for aliphatic coke species, enabling easier detachment process from the metal surface by hydration or hydrogenation. These results are consistent with the observed fewer coke amount in spent PtSn/ γ -Al₂O₃ than Pt/ γ -Al₂O₃. It also rationalized that the preferential formation of aliphatic coke species results in the negligible D and G band signals in the Raman spectrum of PtSn/ γ -Al₂O₃, while considerable coke species are quantified through TG analysis. Unfortunately, these detached carbon fragments are too low in productivity to be collected for analysis and some may adsorb on the surface of the reactor. We also want to emphasize that to reach a precise quantitative identification of the amount of coke is of difficulty, especially at a low conversion rate. Indeed, the TG analysis is more commonly used for qualitative comparison and the coke selectivity is generally calculated by subtraction in propane dehydrogenation reactions [*Nat. Commun.* 15, 6529 (2024); *Nat. Chem.* 16, 575-583 (2024); *Science* 381, 886-890 (2023)]. In propane dehydrogenation, co-feeding of H₂ is generally adopted for eliminating coke deposition. Developing of coke-resistant catalysts in propane-only systems is still a big challenge [*Nat. Chem.* 16, 575-583 (2024); *Nature* 585, 221-224 (2020)]. Nevertheless, we admit the production of a considerable amount of coke deposits in the present system, which is exactly one of the main problems to be solved in the future work. In this work, we would like to emphasize the design and realization of C₃H₈ + H₂O \rightarrow C₃H₆O + 2H₂ as an atom-economic process for alkane utilization and oxygenate production.

We have added related discussions to the revised manuscript (p. 7, lines 143-145, p.14, lines 366-371, Supplementary Table 18, Supplementary Table 11).

“5. The authors show the influence of the volume of water on the catalytic performance. The reviewer agrees with the authors that the amount of water will affect the yields of acetone. However, by varying the water volume in such a small range can cause marked differences is quite interesting. Could these phenomena be reproduced? In particular, it is unlikely that the presence of water can promote the propane dehydrogenation reaction. A plausible explanation for the experimental observation could be that, the addition of water can favour the hydration of propylene, which shifts the propane dehydrogenation reaction. I strongly suggest the authors to carefully re-considered

the discussion in this section.”

We sincerely thank this reviewer for his/her valuable comments. The phenomena are reproducible and we performed three independent measurements for product quantification. We agree with the reviewer’s proposal that the addition of water can favor the hydration of propylene, which shifts the propane dehydrogenation reaction. Of note, there are also several reports that suggest a co-feeding of water steam promotes propane dehydrogenation in increasing reaction rate and decreasing apparent activation energy [*Catal. Sci. Technol.* 2015, 5, 3991-4000; *Chem. Eng. J.* 2015, 278, 240-248; *J. Phys. Chem. C* 2021, 125, 5623-5634]. The different KIE values in isotopic labeling experiment also demonstrated that the conversion of both reactant, propane (KIE=4.5) and propene (KIE=3.0), are highly related to water molecules. Therefore, we think that the incorporation of water both promoted the dehydrogenation and hydration process. As suggested, we have revised the previous discussion (p.9, lines 226-228, p.10, lines 229-240, Refs.35 and 37).

“6. In Fig. 5b, the authors show that, a mixture of propene and water will be transformed into propane, which is surprising to the referee. How can this occur? What’s the mechanism? There are no hydrogen in the reaction feed. How does the reverse hydrogenation reaction occur?”

We sincerely thank this reviewer for his/her insightful questions. First, we would like to identify the specific roles of different active sites that are involved in this reaction. Pt sites are responsible for dehydrogenation, while Pt ensembles contribute to the cracking of C-C bonds. γ -Al₂O₃ is effective for the hydration process. Then, we speculate that following reactions would happen by mixing propene and water over the PtSn/ γ -Al₂O₃, as shown in **Supplementary Table 20** (copied below). H₂ or surface-active Pt-H species can be produced *via* **eqs. R12-15**, which serves as hydrogen source for propene hydrogenation to propane (**eq. R16**).

Supplementary Table 20. Possible reactions involved in propene-water system.

$C_3H_6 + H_2O \rightarrow i-C_3H_7OH$	hydration of propene	(eq. R11)
$i-C_3H_7OH \rightarrow (CH_3)_2CO + H_2$	dehydrogenation of isopropanol	(eq. R12)
$i-C_3H_7OH + 5H_2O \rightarrow 3CO_2 + 9H_2$	steam reforming of isopropanol	(eq. R13)
$C_3H_6 + 3H_2O \rightarrow 3CO + 6H_2$	steam reforming of propene	(eq. R14)
$CO + H_2O \rightarrow CO_2 + H_2$	water-gas shift reaction	(eq. R15)
$C_3H_6 + H_2 \rightarrow C_3H_8$	hydrogenation of propene	(eq. R16)

We have revised the previous discussion (p.12, lines 320-321, p.13, line 322, Supplementary Table 20).

“7. In Supplementary Fig. 30, the authors carry out operando XAS measurements. I suggest they move these data to the main text to replace the ex-situ XAS data shown in Fig. 2.”

We sincerely thank this reviewer for his/her constructive suggestions. As requested, we have

moved the operando XAS data to the main text in the revised Figure 2, d and e. Considering the order of description, we did not delete the *ex-situ* XAS data in Figure 2, d and e.

“8. The direct conversion of alkanes into ketones is an interesting process. However, when comparing this process with the mature industrial process for production of oxygenates, the authors need to consider the very low reaction rates of the direct process. In other words, an interesting reaction proceeds in a very low rate is not meaningful for chemical industry. The discussion and comparison of different processes should be made in a reasonable manner in order to avoid confusion/misleading to the readers.”

We sincerely thank this reviewer for his/her important comments. We recognize that our work currently demonstrates a relatively low reaction rate, yet comparable to some reported propane dehydrogenation catalysts at higher temperatures in terms of mass-specific rate, as shown in **Table R1**.

Table R1 | Comparison of the reaction rates for propane conversion.

Catalyst	Temperature	Production rate	Reference
PtSn/ γ -Al ₂ O ₃	350 °C	0.487 mmol _{acetone} g _{catalyst} ⁻¹ h ⁻¹	This work
MoO ₃ /K-SiO ₂ -TiO ₂	550 °C	0.48 mmol _{propene} g _{catalyst} ⁻¹ h ⁻¹	J. Catal. 191, 12-29 (2000)
h-BN	480 °C	0.6 mmol _{propene} g _{catalyst} ⁻¹ h ⁻¹	Science 372, 76-80 (2021)
V-MCM-41	550 °C	0.714 mmol _{propene} g _{catalyst} ⁻¹ h ⁻¹	Appl. Catal. A: Gen. 209, 155-164 (2001)
0.01Pt-5GaN/STO	300 °C	0.396 mmol _{propene} g _{catalyst} ⁻¹ h ⁻¹	Appl. Catal. B: Environ. Energy 356, 124246 (2024)
VO _x /Al ₂ O ₃	480 °C	0.9 mmol _{propene} g _{catalyst} ⁻¹ h ⁻¹	Appl. Catal. B: Environ. Energy 325, 122337 (2023)

It is important to contextualize our work within the broader scope of chemical research and industry practices. Although the current reaction rates are not yet suitable for industrial application, this foundational research is critical for exploring alternative processes and materials for alkane utilization and oxygenate production. We also elucidated the underlying mechanisms and identify key factors influencing the reaction.

We also want to emphasize that the direct conversion of alkanes to oxygenates are of particular challenge to achieve a high reaction rate involving water as oxygen source, yet an exciting and interesting field to be explored [*Science* 2017, 356, 523-527; *Science* 2020 368, 513-517]. Based on the helpful discussions with the reviewers, we intended to devote our future efforts to catalyst design by further enhancing the intrinsic reaction rate, avoiding coke formation, and facilitating mass transport process via microenvironment tuning.

We have added the relevant discussions in the revised **Conclusion** section (p.14, lines 366-371).

REVIEWERS' COMMENTS

Reviewer #3 (Remarks to the Author):

The authors have now responded to my comments in a satisfactory manner. The manuscript is greatly improved.

Reviewer #4 (Remarks to the Author):

The reviewer is very grateful for the additional experiments and the response to the reviewer's previous comments. The reviewer thinks that this paper is now worthy of publication in Nature Communications.

Reviewer #5 (Remarks to the Author):

The authors have addressed my comments. The paper can be accepted in its present form.

Point-by-point response to reviewers' comments

Manuscript ID: NCOMMS-23-30241C

Title: Propane wet reforming over PtSn nanoparticles on γ -Al₂O₃ for acetone synthesis

Reviewer #3 (Remarks to the Author)

Comment

The authors have now responded to my comments in a satisfactory manner. The manuscript is greatly improved.

Response

We sincerely thank the reviewer for his/her positive comments. We highly appreciate the constructive suggestions and helpful discussions raised by the reviewer for improving our work.

Reviewer #4 (Remarks to the Author)

Comment

The reviewer is very grateful for the additional experiments and the response to the reviewer's previous comments. The reviewer thinks that this paper is now worthy of publication in Nature Communications.

Response

We sincerely thank the reviewer for his/her valuable comments and meaningful suggestions.

Reviewer #5 (Remarks to the Author)

Comment

The authors have addressed my comments. The paper can be accepted in its present form.

Response

We sincerely thank the reviewer for his/her strong support and critical comments which help us improve the quality of this work.